# Shielding Regular Safety Properties in Reinforcement Learning

## Abstract

To deploy reinforcement learning (RL) systems in real-world scenarios we need to consider requirements such as safety and constraint compliance, rather than blindly maximizing for reward. In this paper we study RL with regular safety properties. We present a constrained problem based on the satisfaction of regular safety properties with high probability and we compare our setup to the some common constrained Markov decision processes (CMDP) settings. We also present a meta-algorithm with provable safety-guarantees, that can be used to shield the agent from violating the regular safety property during training and deployment. We demonstrate the effectiveness and scalability of our framework by evaluating our meta-algorithm in both the tabular and deep RL setting.

## 1 Introduction

The field of safe reinforcement learning (RL) [6, 28] has gained increasing interest, as practitioners begin to understand the challenges of applying RL in the real world [26]. There exist several distinct paradigms in the literature, including constrained optimization [2, 20, 49, 58, 62, 74], logical constraint satisfaction [17, 24, 36–38, 66], safety-critical control [15, 19, 53], all of which are unified by prioritizing safety- and risk-awareness during the decision making process.

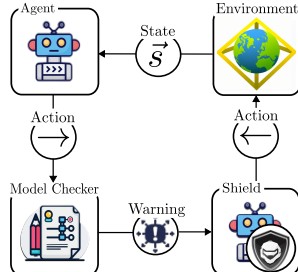

Constrained Markov decision processes (CMDP) [4] have emerged as a popular framework for modelling safe RL, or RL with constraints. Typically, the goal is to obtain a policy that maximizes

Figure 1: Diagrammatic representation of runtime verification and shielding.

reward while simultaneously ensuring that the expected cumulative cost remains below a pre-defined threshold. A key limitation of this setting is that constraint violations are enforced in expectation rather than with high probability, the constraint thresholds also have limited semantic meaning, can be very challenging to tune and in some cases inappropriate for highly safety-critical scenarios [66]. Furthermore, the cost function in the CMDP is typically Markovian and thus fails to capture a significantly expressive class of safety properties and constraints.

Regular safety properties [9] are interesting because for all but the simplest properties the corresponding cost function is non-Markovian. Our problem setup consists of the standard RL objective with regular safety properties as constraints, we note that there has been a significant body of work that combines temporal logic constraints with RL [17, 24, 36–38, 66], although many of these do not explicitly separate reward and safety in the same way that we do.

Our approach relies on shielding [3], which is a safe exploration strategy that ensures the satisfaction of temporal logic constraints by deploying the learned policy in conjunction with a reactive system

that overrides any *unsafe* actions. Most shielding approaches typically make highly restrictive assumptions, such as full knowledge of the environment dynamics [3], or access to a simulator [29], although there has been recent work to deal with these restrictions [30, 39, 73]. In this paper, we opt for the most permissive setting, where the dynamics of the environment are unknown, runtime verification of the agent is realized by finite horizon model checking with a learned approximation of the environment dynamics. However, in principle our framework is flexible enough to accommodate more standard model checking procedures as long as certain assumptions are met.

Our approach can be summarised as an online shielding approach (see Fig. 1), that dynamically identifies unsafe actions during training and deployment, and deploys a safe 'backup policy' when necessary. We summarise the main contributions of our paper as follows:

(1) We state a constrained RL problem based on the satisfaction of regular safety properties with high probability, and we identify the conditions whereby our setup generalizes several CMDP settings, including *expected* and *probabilistic cumulative cost* constraints.

(2) We present several model checking algorithms that can verify the finite-horizon satisfaction probability of regular safety properties, this includes statistical model checking procedures that can be used if either the transition probabilities are unavailable or if the state space is too large.

(3) We develop a set of sample complexity results for the statistical model checking procedures introduced in point (2), which are then used to develop a shielding meta-algorithm with provable safety guarantees, even in the most permissive setting (i.e., no access to the transition probabilities).

(4) We empirically demonstrate the effectiveness of our framework on a variety of regular safety properties in both a tabular and deep RL settings.

## 2   Related Work

**Safety Paradigms in Reinforcement Learning.** There exist many safety paradigms in RL, the most popular being constrained MDPs. For CMDPs several constrained optimization algorithms have been developed, most are gradient-based methods built upon Lagrange relaxations of the constrained problem [20, 49, 58, 62] or projection-based local policy search [2, 74]. Model-based approaches to CMDP [7, 11, 41, 64] have also gathered recent interest as they enjoy better sample complexity than their model-free counterparts, which can be imperative for safe learning [44].

Linear Temporal Logic (LTL) constraints [17, 24, 36–38, 66] for RL have been developed as an alternative to CMDPs to specify stricter and more expressive constraints. The LTL formula is typically treated as the entire task specification, although some works have aimed to separate LTL satisfaction and reward into two distinct objectives [66]. The typical procedure in this setting is to identify end components of the MDP that satisfy the LTL constraint and construct a corresponding reward function such that the optimal policy satisfies the LTL constraint with maximal probability. Formal PAC-style guarantees have been developed for this setting [27, 36, 66, 71] although they typically rely on non-trivial assumptions. We note that LTL constraints can capture regular safety properties, although we explicitly separate reward and safety, making the work in this paper distinct from previous work.

More rigorous safety-guarantees can be obtained by using *safety filters* [3], *control barrier functions* (CBF) [5], and *model predictive safety certification* (MPSC) [67, 68]. To achieve zero-violation training these methods typically assume that the dynamics of the system are known and thus they are typically restricted to low-dimensional systems. While these methods come from safety-critical control, they are closely related to safe reinforcement learning [15].

**Learning Over Regular Structures.** RL and regular properties have been studied in conjunction before, perhaps most famously as 'Reward Machines' [42, 43] – a type of finite state automaton that specifies a different reward function at each automaton state. Reward machines do not explicitly deal with safety, rather non-Markovian reward functions that depend on histories distinguished by regular languages. Several methods have been developed to exploit the structure of these automata and dramatically speed up learning [42, 43, 55, 61], e.g., *counter factual experiences*.

Regular decision processes (RDP) [13] are a specific class non-Markovian DPs [8] that have also been studied in several works [13, 22, 51, 59, 65]. Most of these works are theoretical and slightly out-of-scope for this paper, as the RDP setting does not explicitly handle safety and encompasses both non-Markovian rewards and transition probabilities.

**Shielding.** From formal methods, shielding for safe RL [3] forces hard constraints on policies, using a reactive system that 'shields' the agent from taking unsafe actions. Synthesising a *correct-by-construction* reactive 'shield' typically requires access to the environment dynamics and can be computationally demanding when the state or action space is large. Several recent works have aimed to scale the concept of shielding to more general settings, relaxing the prerequisite assumptions for shielding, by either only assuming access to a 'black box' model for planning [29], or learning a world model from scratch [30, 39, 73]. Other notable works that can be viewed as shielding include, MASE [69] – a safe exploration algorithm with access to an 'emergency reset button', and Recovery-RL [63] – which has access to a 'recovery policy' that is activated when the probability of reaching an unsafe state is too high. A simple form of shielding with LTL specifications has also been considered [37, 54], but experimentally these methods have only been tested in quite simple settings.

## 3  Preliminaries

For a finite set $\mathcal{S}$, let $Pow(\mathcal{S})$ denote the power set of $\mathcal{S}$. Also, let $Dist(\mathcal{S})$ denote the set of distributions over $\mathcal{S}$, where a distribution $\mu : \mathcal{S} \to [0, 1]$ is a function such that $\sum_{s \in \mathcal{S}} \mu(s) = 1$. Let $\mathcal{S}^*$ and $\mathcal{S}^\omega$ denote the set of finite and infinite sequences over $\mathcal{S}$ respectively. The set of all finite and infinite sequences is denoted $\mathcal{S}^\infty = \mathcal{S}^* \cup \mathcal{S}^\omega$. We denote as $|\rho|$ the length of a sequence $\rho \in \mathcal{S}^\infty$, where $|\rho| = \infty$ if $\rho \in \mathcal{S}^\omega$. We also denote as $\rho[i]$ the $i + 1$-th element of a sequence, when $i < |\rho|$, and we denote as $\rho{\downarrow} = \rho[|\rho| - 1]$ the last element of a sequence, when $\rho \in \mathcal{S}^*$. A sequence $\rho_1$ is a prefix of $\rho_2$, denoted $\rho_1 \preceq \rho_2$, if $|\rho_1| \leq |\rho_2|$ and $\rho_1[i] = \rho_2[i]$ for all $0 \leq i \leq |\rho_1|$. A sequence $\rho_1$ is a proper prefix of $\rho_2$, denoted $\rho_1 \prec \rho_2$, if $\rho_1 \preceq \rho_2$ and $\rho_1 \neq \rho_2$.

**Labelled MDPs and Markov Chains.** An MDP is a tuple $\mathcal{M} = (\mathcal{S}, \mathcal{A}, \mathcal{P}, \mathcal{P}_0, \mathcal{R}, AP, L)$, where $\mathcal{S}$ and $\mathcal{A}$ are finite sets of states and actions resp.; $\mathcal{P} : \mathcal{S} \times \mathcal{A} \to Dist(\mathcal{S})$ is the *transition function*; $\mathcal{P}_0 \in Dist(\mathcal{S})$ is the *initial state distribution*; $\mathcal{R} : \mathcal{S} \times \mathcal{A} \to [0, 1]$ is the *reward function*; $AP$ is a set of *atomic propositions*, where $\Sigma = Pow(AP)$ is the *alphabet* over $AP$; and $L : \mathcal{S} \to \Sigma$ is a *labelling function*, where $L(s)$ denotes the set of atoms that hold in a given state $s \in \mathcal{S}$. A memory-less (stochastic) *policy* is a function $\pi : \mathcal{S} \to Dist(\mathcal{A})$ and its *value function*, denoted $V_\pi : \mathcal{S} \to \mathbb{R}$ is defined as the *expected reward* from a given state under policy $\pi$, i.e., $V_\pi(s) = \mathbb{E}_\pi[\sum_{t=0}^{T} \mathcal{R}(s_t, a_t)|s_0 = s]$, where $T$ is a fixed episode length. Furthermore, denote as $\mathcal{M}_\pi = (\mathcal{S}, \mathcal{P}_\pi, \mathcal{P}_0, AP, L)$ the *Markov chain* induced by a fixed policy $\pi$, where the transition function is such that $\mathcal{P}_\pi(s'|s) = \sum_{a \in A} \mathcal{P}(s'|s, a)\pi(a|s)$. A path $\rho \in \mathcal{S}^\infty$ through $\mathcal{M}_\pi$ is a finite (or infinite) sequence of states. Using standard results from measure theory it can be shown that the set of all paths $\{\rho \in \mathcal{S}^\omega \mid \rho_{pref} \preceq \rho\}$ with a common prefix $\rho_{pref}$ is measurable [9].

**Probabilistic CTL.** (PCTL) [9] is a branching-time temporal logic for specifying properties of stochastic systems. A well-formed PCTL property can be constructed with the following grammar,

$$\Phi ::= \text{true} \mid a \mid \neg\Phi \mid \Phi \wedge \Phi \mid \mathbb{P}_{\bowtie p}[\varphi]$$
$$\varphi ::= X\Phi \mid \Phi U \Phi \mid \Phi U^{\leq n} \Phi$$

where $a \in AP$, $\bowtie \in \{<, >, \leq, \geq\}$ is a binary comparison operator, and $p \in [0, 1]$ is a probability. Negation $\neg$ and conjunction $\wedge$ are the familiar logical operators from propositional logic, and next $X$, until $U$ and bounded until $U^{\leq n}$ are the temporal operators from CTL [9]. We make the distinction here between state formula $\Phi$ and path formula $\varphi$. The satisfaction relation for state formula $\Phi$ is defined in the standard way for Boolean connectives. For probabilistic quantification we say that $s \models \mathbb{P}_{\bowtie p}[\varphi]$ iff $\Pr(s \models \varphi) := \Pr(\rho \in S^\omega \mid \rho[0] = s, \rho \models \varphi) \bowtie p$. Let $\Pr^{\mathcal{M}}(s \models \varphi)$ be the probability w.r.t. the Markov chain $\mathcal{M}$. For path formula $\varphi$ the satisfaction relation is as follows,

$$
\begin{array}{lll}
\rho \models X\Phi & \text{iff} & \rho[1] \models \Phi \\
\rho \models \Phi_1 U \Phi_2 & \text{iff} & \exists j \geq 0 \, s.t. \, (\rho[j] \models \Phi_2 \wedge \forall 0 \leq i < j, \rho[i] \models \Phi_1) \\
\rho \models \Phi_1 U^{\leq n} \Phi_2 & \text{iff} & \exists 0 \leq j \leq n \, s.t. \, (\rho[j] \models \Phi_2 \wedge \forall 0 \leq i < j, \rho[i] \models \Phi_1)
\end{array}
$$

From the standard operators of propositional logic we may derive disjunction $\vee$, implication $\to$ and coimplication $\leftrightarrow$. We also note that the common temporal operators 'eventually' $\Diamond$ and 'always' $\Box$, and their bounded counterparts $\Diamond^{\leq n}$ and $\Box^{\leq n}$ can be derived in a familiar way, i.e., $\Diamond\Phi ::= \text{true} U \Phi$, $\Box\Phi ::= \neg\Diamond\neg\Phi$, resp. $\Diamond^{\leq n}\Phi ::= \text{true} U^{\leq n}\Phi$, $\Box^{\leq n}\Phi ::= \neg\Diamond^{\leq n}\neg\Phi$.

**Regular Safety Property.** A linear time property $P_{safe} \subseteq \Sigma^\omega$ over the alphabet $\Sigma$ is a safety property if for all words $w \in \Sigma^\omega \setminus P_{safe}$, there exists a finite prefix $w_{pref}$ of $w$ such that $P_{safe} \cap \{w' \in \Sigma^\omega \mid$

$w_{pref} \preceq w'\} = \varnothing$. Any such sequence $w_{pref}$ is called a *bad prefix* for $P_{safe}$, a bad prefix $w_{pref}$ is called *minimal* iff there does not exist $w'' \prec w_{pref}$ such that $w''$ is a bad prefix for $P_{safe}$. Let $BadPref(P_{safe})$ and $MinBadPref(P_{safe})$ denote the set of of bad and minimal bad prefixes resp.

A safety property $P_{safe} \in \Sigma^\omega$ is *regular* if the set $BadPref(P_{safe})$ constitutes a regular language. That is, there exists some *deterministic finite automata* (DFA) that accepts the bad prefixes for $P_{safe}$ [9], that is, a path $\rho \in \mathcal{S}^\omega$ is 'unsafe' if the trace $trace(\rho) = L(\rho[0]), L(\rho[1]), \ldots \in \Sigma^\omega$ is accepted by the corresponding DFA.

**Definition 3.1** (DFA). *A deterministic finite automata is a tuple $\mathcal{D} = (\mathcal{Q}, \Sigma, \Delta, \mathcal{Q}_0, \mathcal{F})$, where $\mathcal{Q}$ is a finite set of states, $\Sigma$ is a finite alphabet, $\Delta : \mathcal{Q} \times \Sigma \to \mathcal{Q}$ is the transition function, $\mathcal{Q}_0$ is the initial state, and $\mathcal{F} \subseteq \mathcal{Q}$ is the set of accepting states. The extended transition function $\Delta^*$ is the total function $\Delta^* : \mathcal{Q} \times \Sigma^* \to \mathcal{Q}$ defined recursively as $\Delta^*(q, w) = \Delta(\Delta^*(q, w \setminus w{\downarrow}), w{\downarrow})$. The language accepted by DFA $\mathcal{D}$ is denoted $\mathcal{L}(\mathcal{D}) = \{w \in \Sigma^* \mid \Delta^*(\mathcal{Q}_0, w) \in \mathcal{F}\}$.*

Furthermore, we denote as $P_{safe}^H \subseteq \Sigma^\omega$ the corresponding finite-horizon safety property for $H \in \mathbb{Z}_+$, where for all words $w \in \Sigma^\omega \setminus P_{safe}^H$ there exists $w_{pref} \preceq w$ such that $|w_{pref}| \leq H$ and $w_{pref} \in BadPref(P_{safe})$. We model check regular safety properties by synchronizing the DFA and Markov chain in a standard way – by computing the product Markov chain.

**Definition 3.2** (Product Markov Chain). *Let $\mathcal{M} = (\mathcal{S}, \mathcal{P}, \mathcal{P}_0, AP, L)$ be a Markov chain and $\mathcal{D} = (\mathcal{Q}, \Sigma, \Delta, \mathcal{Q}_0, \mathcal{F})$ be a DFA. The product Markov chain is $\mathcal{M} \otimes \mathcal{D} = (\mathcal{S} \times \mathcal{Q}, \mathcal{P}', \mathcal{P}_0', \{accept\}, L')$, where $L'(\langle s, q \rangle) = \{accept\}$ if $q \in \mathcal{F}$ and $L'(\langle s, q \rangle) = \varnothing$ o/w, $\mathcal{P}_0'(\langle s, q \rangle) = \mathcal{P}_0(s)$ if $q = \Delta(\mathcal{Q}_0, L(s))$ and 0 o/w, and $\mathcal{P}'(\langle s', q' \rangle | \langle s, q \rangle) = \mathcal{P}(s'|s)$ if $q' = \Delta(q, L(s'))$ and 0 o/w.*

To compute the satisfaction probability of $P_{safe}$ for a given state $s \in \mathcal{S}$ we consider the set of paths $\rho \in \mathcal{S}^\omega$ from $s$ and the corresponding trace in the DFA. We provide the following definition.

**Definition 3.3** (Satisfaction probability for $P_{safe}$). *Let $\mathcal{M} = (\mathcal{S}, \mathcal{P}, \mathcal{P}_0, AP, L)$ be a Markov chain and let $\mathcal{D} = (\mathcal{Q}, \Sigma, \Delta, \mathcal{Q}_0, \mathcal{F})$ be the DFA such that $\mathcal{L}(\mathcal{D}) = BadPref(P_{safe})$. For a path $\rho \in \mathcal{S}^\omega$ in the Markov chain, let $trace(\rho) = L(\rho[0]), L(\rho[1]), \ldots \in \Sigma^\omega$ be the corresponding word over $\Sigma = Pow(AP)$. From a given state $s \in \mathcal{S}$ the satisfaction probability for $P_{safe}$ is defined as follows,*

$$\mathrm{Pr}^\mathcal{M}(s \models P_{safe}) := \mathrm{Pr}^\mathcal{M}(\rho \in \mathcal{S}^\omega \mid \rho[0] = s, trace(\rho) \notin \mathcal{L}(\mathcal{D}))$$

*Perhaps more importantly, we note that this satisfaction probability can be written as the following reachability probability in the product Markov chain,*

$$\mathrm{Pr}^\mathcal{M}(s \models P_{safe}) = \mathrm{Pr}^{\mathcal{M} \otimes \mathcal{D}}(\langle s, q_s \rangle \not\models \Diamond accept)$$

*where $q_s = \Delta(\mathcal{Q}_0, L(s))$ and $\Diamond accept$ is a PCTL path formula that reads, 'eventually accept' [9].*

For the corresponding finite-horizon safety property $P_{safe}^H$ we state the following result.

**Proposition 3.4** (Satisfaction probability for $P_{safe}^H$). *Let $\mathcal{M}$ and $\mathcal{D}$ be the MDP and DFA in Defn. 3.3. For a path $\rho \in \mathcal{S}^\omega$ in the Markov chain, let $trace_H(\rho) = L(\rho[0]), L(\rho[1]) \ldots, L(\rho[H])$ be the corresponding finite word over $\Sigma = Pow(AP)$. For a given state $s \in \mathcal{S}$ the finite horizon satisfaction probability for $P_{safe}$ is defined as follows,*

$$\mathrm{Pr}^\mathcal{M}(s \models P_{safe}^H) := \mathrm{Pr}^\mathcal{M}(\rho \in \mathcal{S}^\omega \mid \rho[0] = s, trace_H(\rho) \notin \mathcal{L}(\mathcal{D}))$$

*where $H \in \mathbb{Z}_+$ is some fixed model checking horizon. Similar to before, we show that the finite horizon satisfaction probability can be written as the following bounded reachability probability,*

$$\mathrm{Pr}^\mathcal{M}(s \models P_{safe}^H) = \mathrm{Pr}^{\mathcal{M} \otimes \mathcal{D}}(\langle s, q_s \rangle \not\models \Diamond^{\leq H} accept)$$

*where $q_s = \Delta(\mathcal{Q}_0, L(s))$ is as before and $\Diamond^{\leq H} accept$ is the corresponding step-bounded PCTL path formula that reads, 'eventually accept in H timesteps'.*

The unbounded reachability probability can be computed by solving a system of linear equations, the bounded reachability probability can be computed with $\mathcal{O}(H)$ matrix multiplications, in both cases the time complexity of the procedure is a polynomial in the size of the product Markov chain [9].

## 4 Problem Setup

In this paper, we are interested in the quantitative model checking of regular safety properties for a fixed finite horizon $H$ and in the context of episodic RL, i.e., where the length of the episode $T$ is fixed. In particular, at every timestep we constrain the (step-bounded) reachability probability $\Pr(\langle s, q\rangle \not\models \Diamond^{\leq H} accept)$ in the product Markov chain $\mathcal{M}_\pi \otimes \mathcal{D}$. We assume that $H$ is chosen so as to avoid any irrecoverable states [35, 64], i.e., those that lead to a violation of the safety property no matter the sequence of actions taken, the precise details of this notion are presented in Section 6. We specify the following constrained problem,

**Problem 4.1** (Step-wise bounded regular safety property constraint). *Let $P_{safe}$ be a regular safety property, $\mathcal{D}$ be the DFA such that $\mathcal{L}(\mathcal{D}) = BadPref(P_{safe})$ and $\mathcal{M}$ be the MDP;*

$$\max_\pi V_\pi \quad subject\ to \quad \Pr\left(\langle s_t, q_t\rangle \models \Diamond^{\leq H} accept\right) \leq p_1 \quad \forall t \in [0, T]$$

*where all probability is taken under the product Markov Chain $\mathcal{M}_\pi \otimes \mathcal{D}$, $p_1 \in [0, 1]$ is a probability threshold, $H$ is the model checking horizon and $T$ is the fixed episode length.*

The hyperparameter $p_1$ is be directly used to trade-off safety and exploration in a semantically meaningful way; $p_1$ prescribes the probability of satisfying the finite-horizon safety property $P_{safe}^H$ at each timestep. In particular, if $p_1$ is sufficiently small then we can guarantee (with high-probability) that the regular safety property $P_{safe}$ is satisfied for the entire episode length $T$.

**Proposition 4.2.** *Let $P_{safe}^T$ denote the (episodic) regular safety property for a fixed episode length $T$. Then satisfying $\Pr\left(\langle s_t, q_t\rangle \models \Diamond^{\leq H} accept\right) \leq p_1$ for all $t \in [0, T]$ guarantees that $\Pr(s_0 \models P_{safe}^T) \geq 1 - p_1 \cdot \lceil T/H \rceil$, where $s_0 \sim \mathcal{P}_0$ is the initial state.*

**Comparison to CMDP.** In the remainder of this section, we compare our problem setup to various CMDP settings [4], with the aim of unifying different perspectives from safe RL. The purpose of this is to show that our proposed method for solving Problem 4.1 can also be used to satisfy other more common CMDP constraints. First, we define the following cost function that prescribes a scalar cost $C > 0$ when the regular safety property $P_{safe}$ is violated and 0 otherwise.

**Definition 4.3** (Cost function). *Let $P_{safe}$ be a regular safety property and let $\mathcal{D}$ be the DFA such that $\mathcal{L}(\mathcal{D}) = BadPref(P_{safe})$, modified such that for all $q \in \mathcal{F}$, $q \to \mathcal{Q}_0$. The cost function is then defined as,*

$$\mathcal{C}(\langle s, q\rangle) = \begin{cases} C & if\ accept \in L'(\langle s, q\rangle) \\ 0 & otherwise \end{cases}$$

*where $C > 0$ is some generic scalar cost and $L'$ is the labelling function defined in Def. 3.2.*

*Resetting the DFA.* Rather than reset the environment, the DFA is reset once it reaches an accepting state, so as to measure the rate of constraint satisfaction over a fixed episode length $T$. This can easily be realized by replacing any outgoing transitions from the accepting states with transitions back to the initial state, i.e., for all $q \in \mathcal{F}$, $q \to \mathcal{Q}_0$.

*Non-Markovian costs.* The cost function is Markov on the product states $\langle s, q\rangle \in \mathcal{S} \times \mathcal{Q}$. However, in most cases the cost function is non-Markovian in the original state space $\mathcal{S}$, since the automaton state $q \in \mathcal{Q}$ could depend on some arbitrary history of states. Thus our problem setup generalizes the standard CMDP framework with non-Markovian safety constraints.

*Invariant properties.* Invariant properties $P_{inv}(\Phi)$, also written $\Box \Phi$ ('always $\Phi$'), where $\Phi$ is a propositional state formula, are the simplest type of safety properties where the cost function is still Markov in the original state space. In this case we are operating in the standard CMDP framework, we also note that checking invariant properties with a fixed model checking horizon has been studied in previous works, as *bounded safety* [29, 30] and *safety for a finite horizon* [45].

The most common type of CMDP constraints are *expected cumulative (cost) constraints*, which constrain the expected cost below a given threshold.

**Problem 4.4** (Expected cumulative constraint [4, 58]).

$$\max_\pi V_\pi \quad subject\ to \quad \mathbb{E}_{\langle s_t, q_t\rangle \sim \mathcal{M}_\pi \otimes \mathcal{D}}\left[\sum_{t=0}^{T} \mathcal{C}(\langle s_t, q_t\rangle)\right] \leq d_1$$

*where $d_1 \in \mathbb{R}_+$ is the cost threshold and $T$ is the fixed episode length.*

*Probabilistic cumulative (cost) constraints*, are a stricter class of constraints that constrain the cumulative cost with high probability, rather than in expectation.

**Problem 4.5** (Probabilistic cumulative constraint [18, 56])**.**

$$\max_{\pi} V_\pi \quad subject\ to \quad \mathbb{P}_{\langle s_t, q_t \rangle \sim \mathcal{M}_\pi \otimes \mathcal{D}} \left[ \sum_{t=0}^{T} \mathcal{C}(\langle s_t, q_t \rangle) \leq d_2 \right] \geq 1 - \delta_2$$

*where $d_2 \in \mathbb{R}_+$ is the cost threshold, $\delta_2$ is a tolerance parameter, and $T$ is the fixed episode length.*

We also consider *instantaneous constraints*, which bound the cost 'almost surely' at each timestep $t \in [0, T]$. These are an even stricter type of constraint for highly safety-critical applications.

**Problem 4.6** (Instantaneous constraint [23, 60, 69])**.**

$$\max_{\pi} V_\pi \quad subject\ to \quad \mathbb{P}_{\langle s_t, q_t \rangle \sim \mathcal{M}_\pi \otimes \mathcal{D}} \left[ \mathcal{C}(\langle s_t, q_t \rangle) \leq d_3 \right] = 1 \quad \forall t \in [0, T]$$

*where $d_3 \in \mathbb{R}_+$ is the cost threshold and $T$ is the fixed episode length.*

In particular, these problems define a constrained set of feasible policies $\Pi$. We make the distinction here between a feasible policy and a solution to the problem, the former being any policy satisfying the constraints of the problem and the later being the optimal policy within the feasible set $\Pi$.

**Theorem 4.7.** *A feasible policy for Problem 4.1 is also a feasible policy for Problems 4.4, 4.5 and 4.6 under specific parameter settings for $p_1$, $d_1$, $d_2$ and $\delta_2$, and $d_3$.*

In Appendix G we provide a full set of statements that outline the relationships between the constrained problems presented in this section. The significance of these results is that they demonstrate by solving Problem 4.1 with our proposed method we can obtain feasible policies for Problems 4.4, 4.5 and 4.6, although for most of these problems there is no direct relationship between our problem setup, in particular we can say little about whether the optimal policy for one problem is necessarily optimal for another. Nevertheless, we find it interesting to explore the relationships between our setup and other perhaps more common constrained RL problems.

# 5 Model checking

In this section we outline several procedures for checking the finite-horizon satisfaction probability of regular safety properties and we summarise the settings in which they can be used.

**Assumption 5.1.** *We are given access to the 'true' transition probabilities $\mathcal{P}$.*

**Assumption 5.2.** *We are given access to a 'black box' model that perfectly simulates the 'true' transition probabilities $\mathcal{P}$.*

**Assumption 5.3.** *We are given access to an approximate dynamic model $\widehat{\mathcal{P}} \approx \mathcal{P}$, where the total variation (TV) distance $D_{TV}(\mathcal{P}_\pi(\cdot \mid s), \widehat{\mathcal{P}}_\pi(\cdot \mid s)) \leq \epsilon/H$, for all $s \in \mathcal{S}$.*[1]

**Exact model checking.** Under Assumption 5.1 we can precisely compute the (finite horizon) satisfaction probability of $P_{safe}$, in the Markov chain $\mathcal{M}_\pi$ induced by the fixed policy $\pi$ in time $\mathcal{O}(\text{poly}(\text{size}(\mathcal{M}_\pi \otimes \mathcal{D})) \cdot H)$ [9], where $\mathcal{D}$ is the DFA such that $\mathcal{L}(\mathcal{D}) = BadPref(P_{safe})$ and $H$ is the model checking horizon. $H$ should not be too large and so the complexity of exact model checking ultimately depends on the size of the product $\mathcal{M}_\pi \otimes \mathcal{D}$, and so if the size of either the MDP or DFA is too large then exact model checking may be infeasible.

**Monte-Carlo model checking.** To address the limitations of exact model checking, we can drop Assumption 5.1. Rather, under Assumption 5.2, we can sample sufficiently many paths from a 'black box' model of the environment dynamics and estimate the reachability probability $\Pr(\langle s, q \rangle \models \Diamond^{\leq H} accept)$ in the product Markov chain $\mathcal{M}_\pi \otimes \mathcal{D}$, by computing the proportion of accepting paths. Using statistical bounds, such as Hoeffding's inequality [40] or Bernstein-type bounds [52], we can bound the error of this estimate, with high probability.

**Proposition 5.4.** *Let $\epsilon > 0$, $\delta > 0$, $s \in \mathcal{S}$ and $H \geq 1$ be given. Under Assumption 5.2, we can obtain an $\epsilon$-approximate estimate for the probability $\Pr(\langle s, q \rangle \models \Diamond^{\leq H} accept)$ with probability at least $1 - \delta$, by sampling $m \geq \frac{1}{2\epsilon^2} \log\left(\frac{2}{\delta}\right)$ paths from the 'black box' model.*

---

[1]For two discrete probability distributions $\mu_1$ and $\mu_2$ over the same space $\mathcal{X}$ the TV distance is defined as: $D_{TV}(\mu_1(\cdot), \mu_2(\cdot)) = \frac{1}{2} \sum_{x \in X} |\mu_1(x) - \mu_2(x)|$

We note that the time complexity of these statistical methods does not depend in the size of the product MDP or DFA, since the product states $\langle s, q \rangle \in \mathcal{S} \times \mathcal{Q}$ can be computed *on-the-fly*, rather the time complexity depends on the horizon $H$, the desired level of accuracy $\epsilon$, failure probability $\delta$.

**Model checking with approximate models.** In most realistic cases neither the 'true' transition probabilities nor a perfect 'black box' model is available to us before-hand. Under Assumption 5.3 we can model check with an 'approximate' model of the MDP dynamics, which can either be constructed ahead of time (offline) or learned from experience, with maximum likelihood (or similar). We can then either exact model check in with the 'approximate' probabilities, or if the MDP is too large, we can leverage statistical model checking by sampling paths from the 'approximate' model.

**Proposition 5.5.** *Let $\epsilon > 0$, $\delta > 0$, $s \in \mathcal{S}$ and $H \geq 1$ be given. Under Assumption 5.3 we can make the following two statements:*

*(1) We can obtain an $\epsilon$-approximate estimate for $\Pr(\langle s, q \rangle \models \Diamond^{\leq H} accept)$ with probability $1$ by exact model checking with the transition probabilities of $\widehat{\mathcal{P}}_\pi$ in time $\mathcal{O}(poly(size(\mathcal{M}_\pi \otimes \mathcal{D})) \cdot H)$.*

*(2) We can obtain an $\epsilon$-approximate estimate for $\Pr(\langle s, q \rangle \models \Diamond^{\leq H} accept)$ with probability at least $1 - \delta$, by sampling $m \geq \frac{2}{\epsilon^2} \log\left(\frac{2}{\delta}\right)$ paths from the 'approximate' dynamics model $\widehat{\mathcal{P}}_\pi$.*

# 6 Shielding the policy

At a high-level, the shielding meta-algorithm works by switching between the 'task policy' trained with RL to maximize rewards and a 'backup policy', which typically constitutes a low-reward, possibly rule-based policy that is guaranteed to be safe. In some cases this 'backup policy' may be available to us before training, although in most realistic cases it will need to be learned. In our case we switch from the 'task policy' to the 'backup policy' when the reachability probability $\Pr(\langle s, q \rangle \models \Diamond^{\leq H} accept)$ exceeds the probability threshold $p_1$. To check this we can use any of the model checking procedures presented earlier. The 'backup policy' is used when the reachability probability exceeds $p_1$. Intuitively if the 'backup policy' is guaranteed to be safe, then our system should satisfy the constraints of Problem 4.1, independent of the 'task policy'.

**Backup policy.** In general we assume no knowledge of the safety dynamics before training and so the 'backup policy' needs to be learned. In particular, we can use the cost function defined in Defn. 4.3 and train the 'backup policy' with RL to minimize the *expected discounted cost*

---

**Algorithm 1** Shielding (with runtime verification of regular safety properties)

---
**Input:** model checking parameters ($\epsilon$, $\delta$, $p$, $H$), labelling function $L$, DFA $\mathcal{D} = (\mathcal{Q}, \Sigma, \Delta, \mathcal{Q}_0, \mathcal{F})$.
*Optional:* probabilities $\mathcal{P}$, 'backup policy' $\pi_{safe}$.
**Initialize:** 'task policy' $\pi_{task}$, 'backup policy' $\pi_{safe}$ and (approximate) probabilities $\widehat{\mathcal{P}}$.
**for each episode do**
    Observe $s_0$, $L(s_0)$ and $q_0 \leftarrow \Delta(\mathcal{Q}_0, L(s_0))$
    **for** $t = 0, \ldots, T$ **do**    ▷ Fixed episode length
        Sample action $a \sim \pi_{task}(\cdot \mid s_t)$
        **if** $\Pr(\langle s, q \rangle \models \Diamond^{\leq H} accept) \leq p_1$ **then**
            *// Use the proposed action*
            $a_t \leftarrow a$
        **else**
            *// Override the action*
            $a_t \sim \pi_{safe}(\cdot \mid s_t, q_t)$
        Play $a_t$ and observe $s_{t+1}$, $L(s_{t+1})$, $r_t$
        $q_{t+1} \leftarrow \Delta(q_t, L(s_{t+1}))$,
        $c_t \leftarrow 1[q_{t+1} \in \mathcal{F}]$
        Update $\pi_{task}$ with $(s_t, a_t, s_{t+1}, r_t)$
        Update $\pi_{safe}$ with $(s_t, q_t, a_t, s_{t+1}, q_{t+1}, c_t)$
        Update $\widehat{\mathcal{P}}$ with $(s_t, a_t, s_{t+1})$

---

$(\mathbb{E}_\pi[\sum_{t=0}^{T} \gamma^t \mathcal{C}(s_t, q_t)])$. Importantly, we note that the cost function is defined on the product state space $\mathcal{S} \times \mathcal{Q}$ and so the 'backup policy' must also operate on this state space, possibly leading to slower convergence. However, we can eliminate this issue entirely by training the 'backup policy' with *counterfactual experiences* [42, 43] – a method originally used for reward machines that generates additional synthetic data for the policy, by simulating experience from each automaton state.

**Meta Algorithm.** We now present the structure of the shielding meta-algorithm (see Algorithm 1). The precise realization of this algorithm can vary depending on problem setting, tabular, deep RL, etc., however the main structure of the algorithm remains the same. In particular, during interaction with the environment we shield the agent by checking that the reachability probability $\Pr(\langle s, q \rangle \models \Diamond^{\leq H} accept)$ does not exceed threshold $p_1$. Then, with the new accumulated experience we update the 'task policy' denoted $\pi_{task}$ and the 'backup policy' denoted $\pi_{safe}$ with RL, and if need be

we update our (approximate) dynamics model accordingly. In principle, the underlying RL algorithm used to train either 'task policy' or 'backup policy' can differ, and the dynamics model can be a simple maximum likelihood estimate or something more complex, e.g., Gaussian Process model [25, 70], ensemble of parametric neural networks [21, 44] or a world model [32, 33].

**Global Safety Guarantees.** In the tabular setting we can guarantee the safety of the system described in Algorithm 1 under various assumptions, even when doing Monte-Carlo model checking on an 'approximate' model of the environment dynamics. First, we provide the following definitions.

**Definition 6.1** (Non-critical state). *A product state $\langle s, q \rangle \in \mathcal{S} \times \mathcal{Q}$ is said to be non-critical for a given model checking horizon $H$ if for all policies $\pi$ we have $\Pr(\langle s, q \rangle \models \Diamond^{\leq H} accept) = 0$.*

**Definition 6.2** (Irrecoverable). *A critical state $\langle s, q \rangle \in \mathcal{S} \times \mathcal{Q}$ is said to be irrecoverable with probability $p_1$ if for all policies $\pi$ we have $\Pr(\langle s, q \rangle \models \Diamond accept) \geq p_1$. In other words, for any sequence of actions $a_0, a_1, \ldots$ the minimum probability $\Pr^{min}(\langle s, q \rangle \models \Diamond accept)$ of reaching an accepting state is $p_1$, where $\Pr^{min}(\langle s, q \rangle \models \Diamond accept) = \inf_\pi \Pr^{\mathcal{M}_\pi \otimes \mathcal{D}}(\langle s, q \rangle \models \Diamond accept)$*

The safety-guarantees for Algorithm 1 rely on the following assumptions.

**Assumption 6.3.** *We assume $H$ is sufficiently large so that it is not possible to transition from any non-critical state to an irrecoverable state. Furthermore we assume that there exists some $H^* < H$ such that if $\Pr^{min}(\langle s, q \rangle \models \Diamond accept) = p_1$ then $\Pr^{min}(\langle s, q \rangle \models \Diamond^{\leq H^*} accept) = p_1$.*

**Assumption 6.4.** *The initial state $\langle s_0, L(s_0) \rangle$ is non-critical and for any state $\langle s, q \rangle \in \mathcal{S} \times \mathcal{Q}$ that is not irrecoverable, the 'backup policy' $\pi_{safe}$ is satisfies $\Pr^{\mathcal{M}_{\pi_C} \otimes \mathcal{D}}(\langle s, q \rangle \models \Diamond^{\leq H} accept) \leq p_1$*

**Theorem 6.5.** *Under Assumption 6.3 and 6.4, and provided that every state action pair $(s, a) \in \mathcal{S} \times \mathcal{A}$ has been visited at least $\mathcal{O}\left(\frac{H^2 |\mathcal{S}|^2}{\epsilon^2} \log\left(\frac{|\mathcal{A}||\mathcal{S}|^2}{\delta}\right)\right)$ times. Then with probability $1 - \delta$ the system satisfies the constraints of Problem 4.1, independent of the 'task policy'.*

The theory is quite conservative here due to the strong dependence on $|S|$, in practice we can replace the outer $|S|^2$ by the maximum number of successor states from any given state. With regards to our assumptions, both are not overly restrictive. Assumption 6.3 essentially states that any irrecoverable states, will reach the accepting state with some probability $> g$ within a fixed horizon $H^*$. Similar statements have been considered in prior work [35, 64]. Assumption 6.4 states that the 'backup policy' satisfies $\Pr(\langle s, q \rangle \models \Diamond^{\leq H} accept) \leq p_1$ if possible, we would expect this to be the case when training the 'backup policy' with RL to minimize cost. The analysis for Theorem 6.5 then follows by showing that the system can be recovered to a non-critical state after entering a critical but not irrecoverable state.

# 7   Empirical Evaluation

We implemented two separate realizations of Algorithm 1, the first adapted to tabular environments which implements both exact or statistical model checking over the learned transition probabilities, the second is adapted to (visual) deep RL, making use of *world models* [32, 33], specifically DreamerV3 [34], to learn a latent dynamics model for model checking and policy optimization.

**Tabular RL.** We conduct experiments on a simple 'colour' grid-world environment, with regular safety properties of increasing difficulty. In short, the goal is to navigate from a starting state to a goal position as frequently as possible, while respecting a given regular safety property during training. The environment is stochastic – with some probability $p$ the agent's action is ignored and another action is chosen uniformly instead. For smaller $p$ val-

Table 1: Safety properties

| property $P_{safe}$ |
| --- |
| (1) $\Box \neg green$ |
| (2) $\Box goal \rightarrow \Diamond^{\leq 10} blue$ |
| (3) $\Box goal \rightarrow \Diamond^{\leq 10} \Box^{\leq 5} purple$ |

ues the environment becomes more deterministic and the safety property typically becomes easier to satisfy with higher probability, we refer the reader to Appendix D.1 for more details. Table 1 outlines the three safety properties used for our environments. We use PCTL-like notation to describe the safety properties, although strictly speaking (2) and (3) are actually PCTL$^*$ path formula. Regardless of this slight technical detail, properties (1)-(3) are valid regular safety properties, as we can come up with a DFA that accepts the bad prefixes for them.

We compare our approach to Q-learning (without any penalties), and Q-learning on the product state space, with penalties provided by the cost function (Defn. 4.3) and trained with counterfactual

experiences [43]. In all cases, by separating reward and safety into two distinct policies, we are able to effectively trade-off the two objectives. Q-learning simply finds the best policy ignoring the costs, and Q-learning with penalties is able to find a safe policy, but struggles to meaningfully balance both objectives (see Fig. 2). Hyperparameter settings for all experiments are detailed in Appendix E. In addition, we provide an extensive series of ablation studies in Appendix F for these experiments. For example, we show that we don't loose much by using Monte Carlo model checking as opposed to exact model checking with the 'true' probabilities. We also show that tuning the cost coefficient $C$ offers no meaningful way to trade-off reward and the probability of constraint satisfaction.

**Deep RL.** We deploy our version of Algorithm 1 built on DreamerV3 [34] on Atari Seaquest, provided as part of the Arcade Learning Environment (ALE)[10, 50]. We experiment with two different regular safety properties: (1) ($\Box \neg surface \rightarrow \Box(surface \rightarrow diver)$) $\wedge$ ($\Box \neg out\text{-}of\text{-}oxygen$) $\wedge$ ($\Box \neg hit$) and (2) $\Box diver \wedge \neg surface \rightarrow \Diamond^{\leq 30} surface$. We compare our approach to the base DreamerV3 algorithm and a version of DreamerV3 that implements the augmented Lagrangian penalty framework, similarly to [7, 41], for additional details see Appendix B.1.

Again our approach is able to effectively trade-off both objectives, while (base) DreamerV3 ignores the cost, the Lagrangian approach appears to learn a safe policy that is not always efficient in terms of reward (see Fig. 3). We refer the reader to Appendix D.2 for more details of the environment and an extended discussion.

**Separating Reward and Safety.** The separation of reward and safety objectives into two distinct policies has been demonstrated as an effective strategy towards safety-aware decision making [3, 30, 46, 63], in many cases the safety objective is simpler and can be more quickly learnt [46]. In our experiments it is clear that when the system enters a critical state, the 'backup policy' is able to efficiently guide the system back to a non-critical state where the task policy can continue collecting reward. However, there is evidence that the complete separation of policies is not always appropriate [31] and penalties or a slight coupling of the policies is required to stop the 'task' and 'backup policy' fighting for control of the system. Furthermore, by separating reward and safety, we typically loose any asymptotic convergence guarantees, similar to the situation faced for hierarchical RL [61], although there has been recent work to develop convergence guarantees for shielding [75].

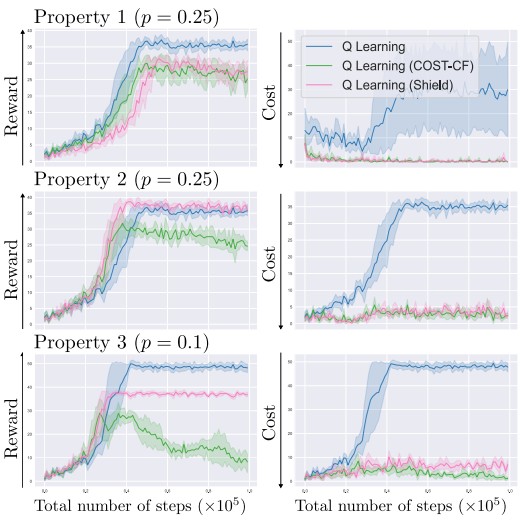

Figure 2: Episode reward and cost for tabular RL 'colour' gridworld environment.

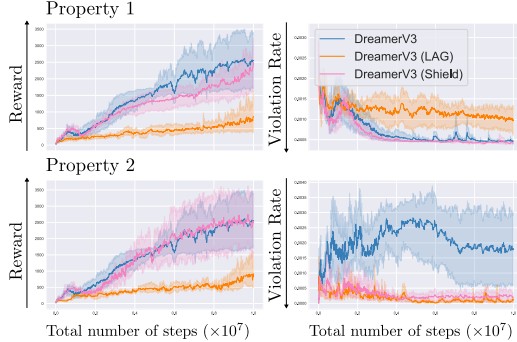

Figure 3: Episode reward and violation rate for deep RL Atari Seaquest.

# 8 Conclusion

In this paper we propose a shielding meta-algorithm for the runtime verification of regular safety properties, given as a probabilistic constraint on the system. We provide a thorough theoretical examination of the problem and develop probabilistic safety guarantees for the meta-algorithm, which hold under reasonable assumptions. Empirically, we demonstrate that shielding is able to effectively balance both reward and safety, in both the tabular and deep RL setting. A more thorough theoretical and empirical examinations of the conditions for when shielding is appropriate would be an interesting direction for future work.

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

 # A  Algorithms

---

**Algorithm 2** Exact Model Checking [9]

---

**Input:** model checking parameters $(p, H)$, current state $\langle s, q \rangle$, current action $a$, product MC
$\mathcal{M}_\pi \otimes \mathcal{D} = (\mathcal{S} \times \mathcal{Q}, \mathcal{P}', \mathcal{P}'_0, \{accept\}, L')$
**Output:** *true* if $\Pr(\langle s, q \rangle \models \Diamond^{\leq H} accept) \leq p_1$

Initialize zero vector $\mathbf{x}^{(0)} \leftarrow \mathbf{0}$ with size $|\mathcal{S}| \times |\mathcal{Q}|$
Initialize probability matrix $\mathbf{A} \leftarrow (\mathcal{P}'(s, t))_{s, t \notin accept}$ (ignoring accepting states)
Initialize probability vector $\mathbf{b} \leftarrow (\mathcal{P}'(s, accept))_{s \notin accept}$ (going to accepting states)
*// Iterate over the model checking horizon*
**for** $i = 1, \ldots, H$ **do**
    Compute $\mathbf{x}^{(i)} = \mathbf{A}\mathbf{x}^{(i-1)} + \mathbf{b}$
*// Get the corresponding probability*
Let $X \leftarrow \mathbf{x}_{\langle s, q \rangle}$
**If** $X < p$ **return** *true* **else return** *false*

---

**Algorithm 3** Monte-Carlo Model Checking

---

**Input:** model checking parameters $(\epsilon, \delta, p, H)$, current state $\langle s, q \rangle$, current action $a$, policy $\pi$,
labelling function $L$, DFA $\mathcal{D} = (\mathcal{Q}, \Sigma, \Delta, \mathcal{Q}_0, \mathcal{F})$ and (approximate) transition probabilities $\mathcal{P}$
**Output:** *true* if $\Pr(\langle s, q \rangle \models \Diamond^{\leq H} accept) \leq p_1$

Choose $m \geq 2/(\epsilon^2) \log(2/\delta)$
**for** $i = 1, \ldots, m$ **do**
    Set $s_0 \leftarrow s$, $q_0 \leftarrow q$ and $a_0 \leftarrow a$
    *// Sample a path through the model*
    **for** $j = 1, \ldots, H$ **do**
        Sample next state $s_j \sim \mathcal{P}(\cdot \mid s_{j-1}, a_{j-1})$,
        Compute $q_j \leftarrow \Delta(q_{j-1}, L(s_j))$,
        Sample action $a_j \sim \pi(\cdot \mid s_j)$
    *// Check if the path is accepting*
    Let $X_i \leftarrow \mathbb{1}[q_H \in \mathcal{F}]$
*// Construct probability estimate*
Let $\widetilde{X} \leftarrow \frac{1}{m} \sum_{i=1}^{m} X_i$
**If** $\widetilde{X} < p - \epsilon$ **return** *true* **else return** *false*

---

**Algorithm 4** Tabular Q-learning (Regular Safety Property) with Counter Factual Experiences [65]

---

**Input:** MDP $\mathcal{M} = (\mathcal{S}, \mathcal{A}, \mathcal{P}, \mathcal{P}_0, \mathcal{R}, AP, L)$, DFA $\mathcal{D} = (\mathcal{Q}, \Sigma, \Delta, \mathcal{Q}_0, \mathcal{F})$, discount factor $\gamma \in (0, 1]$, learning rate $\alpha \in (0, 1]$, temperature $\tau > 0$, cost coefficient $C$ and fixed episode length $T$
**Initialize:** (Q-table) $\hat{Q}(s, q, a) \leftarrow 0 \,\forall s \in \mathcal{S}, q \in \mathcal{Q}, a \in \mathcal{A}$

**for each episode do**
    Observe $s_0$, $L(s_0)$ and $q_0 \leftarrow \Delta(\mathcal{Q}_0, L(s_0))$
    **for** $t = 0, \ldots, T$ **do**
        Sample action $a_t$ from $\langle s_t, q_t \rangle$ using the Boltzmann policy derived from $\hat{Q}$ with temp. $\tau$
        Play action $a_t$ and observe $s_{t+1}$, $L(s_{t+1})$ and $r_t$ (reward is optional).
        *// Generate synthetic data by simulating all automaton transitions*
        **for** $\bar{q} \in \mathcal{Q}$ **do**
            Compute $\bar{q}' \leftarrow \Delta(q', L(s_{t+1}))$
            Compute cost $\bar{c}' \leftarrow C \cdot \mathbb{1}[\bar{q}' \in \mathcal{F}]$
            Compute *done* $\leftarrow \mathbb{1}[\bar{q}' \in \mathcal{F}]$
            *// Q-learning step*
            $\hat{Q}(s_t, \bar{q}, a_t) \leftarrow (1 - \alpha) \cdot \hat{Q}(s_t, \bar{q}, a_t) + \alpha \cdot (r_t + \bar{c}' + \gamma \cdot done \cdot \max_{a' \in \mathcal{A}} \hat{Q}(s_{t+1}, \bar{q}', a')$
        Compute $q_{t+1} \leftarrow \Delta(q_t, L(s_{t+1}))$ and continue

---

**Algorithm 5** DreamerV3 [34] with Shielding (Regular Safety Property)

---

**Initialize:** replay buffer $D$ with $S$ random episodes, world model parameters $\theta$, 'task policy' $\pi_{task}$ and 'backup policy' $\pi_{safe}$ randomly.

**for each episode do**
    Observe $o_0$, $L(s_0)$ and $q_0 \leftarrow \Delta(\mathcal{Q}_0, L(s_0))$
    **for** t = 1, . . . , T **do**
        Sample action $a \sim \pi_{task}$ from the task policy
        *// Estimate the reachability probability using the world model $p_\theta$*
        **if** $\Pr(\langle s, q \rangle \models \Diamond^{\leq H} accept) \leq p_1$ **then**
            *Use proposed action*
            $a_t \leftarrow a$
        **else**
            *// Override action*
            $a_t \sim \pi_{safe}$
        Play action $a_t$ and observe $o_{t+1}$, $L(s_{t+1})$ and $r_t$
        Compute $q_{t+1} \leftarrow \Delta(q_t, L(s_{t+1}))$,
        Compute cost $c_t \leftarrow 1[q_{t+1} \in \mathcal{F}]$
        Append $(o_t, a_t, r_t, c_t, o_{t+1})$ to the replay buffer $D$
        **if update then**
            *// World model learning*
            Sample a batch $B$ of transition sequences $\{(o_{t'}, a_{t'}, r_{t'}, c_{t'}, o_{t'+1})\} \sim \mathcal{D}$.
            Update the world model parameters $\theta$ with maximum likelihood.
            *// Task policy optimization*
            'Imagine' sequences $\{\hat{o}_{t':t'+H}, \hat{r}_{t':t'+H}, \hat{c}_{t':t'+H}\}$ with the 'task policy' $\pi_{task}$
            Update the 'task policy' $\pi_{task}$ with RL (to maximize reward)
            Update the corresponding value critics with maximum likelihood
            *// Backup policy optimization*
            'Imagine' sequences $\{\hat{o}_{t':t'+H}, \hat{r}_{t':t'+H}, \hat{c}_{t':t'+H}\}$ with the 'backup policy' $\pi_{safe}$
            Update the 'backup policy' $\pi_{safe}$ with RL (to minimize cost)
            Update the corresponding value critics with maximum likelihood

---

# B Technical Details

## B.1 Augmented Lagrangian

We first define the following objective functions,

$$J_{\mathcal{R}}(\pi) \;=\; \mathbb{E}_\pi \left[ \sum_{t=0}^{T} \mathcal{R}(s_t, a_t) \right] \tag{1}$$

$$J_{\mathcal{C}}(\pi) \;=\; \mathbb{E}_\pi \left[ \sum_{t=0}^{T} \mathcal{C}(s_t, a_t) \right] \tag{2}$$

$$\tag{3}$$

The augmented Lagrangian [72] is an adaptive penalty-based technique for the following constrained optimization problem,

$$\max_{\pi} J_{\mathcal{R}}(\pi) \quad \text{subject to} \quad J_{\mathcal{C}}(\pi) \leq d \tag{4}$$

where $d$ is some cost threshold. The corresponding Lagrangian is given by,

$$\max_{\pi} \min_{\lambda \geq 0} \left[ J_{\mathcal{R}}(\pi) - \lambda \left( J_{\mathcal{C}}(\pi) - d \right) \right] = \max_{\pi} \begin{cases} J_{\mathcal{R}}(\pi) & \text{if } J_{\mathcal{C}}(\pi) < d \\ -\infty & \text{otherwise} \end{cases} \tag{5}$$

The LHS is an equivalent form for the constrained optimization problem (RHS), since if $\pi$ is feasible, i.e. $J_{\mathcal{C}}(\pi) < d$ then the maximum value for $\lambda$ is $\lambda = 0$. If $\pi$ is not feasible then $\lambda$ can be arbitrarily large to solve this equation. Unfortunately this form of the objective function is non-smooth when moving from feasible to infeasible policies, thus we introduce a proximal relaxation of the augmented

Lagrangian [72],

$$\max_{\pi} \min_{\lambda \geq 0} \left[ J_{\mathcal{R}}(\pi) - \lambda \left( J_{\mathcal{C}}(\pi) - d \right) + \frac{1}{\mu_k} (\lambda - \lambda_k)^2 \right] \tag{6}$$

where $\mu_k$ is a non-decreasing penalty multiplier dependent on the gradient step $k$. The new term that has been introduced here encourages the $\lambda$ to stay close to the previous value $\lambda_k$, resulting in a smooth and differentiable function. The derivative w.r.t $\lambda$ gives us the following gradient update step,

$$\lambda_{k+1} = \begin{cases} \lambda_k + \mu_k(J_{\mathcal{C}}(\pi) - d) & \text{if } \lambda_k + \mu_k(J_{\mathcal{C}}(\pi) - d) \geq 0 \\ 0 & \text{otherwise} \end{cases} \tag{7}$$

At each gradient step, the penalty multiplier $\mu_k$ is updated in a non-decreasing way by using some small fixed (power) parameter $\sigma$,

$$\mu_{k+1} = \max\{(\mu_k)^{1+\sigma}, 1\} \tag{8}$$

The policy $\pi$ is then updated by taking gradient steps of the following unconstrained objective,

$$\tilde{J}(\pi, \lambda_k, \mu_k) = J_{\mathcal{R}}(\pi) - \Psi_{\mathcal{C}}(\pi, \lambda_k, \mu_k)$$

where,

$$\Psi_{\mathcal{C}}(\pi, \lambda_k, \mu_k) = \begin{cases} \lambda_k(J_{\mathcal{C}}(\pi) - d) + \frac{\mu_k}{2}(J_{\mathcal{C}}(\pi) - d)^2 & \text{if } \lambda_k + \mu_k(J_{\mathcal{C}}(\pi) - d) \geq 0 \\ -\frac{(\lambda_k)^2}{2\mu_k} & \text{otherwise} \end{cases}$$

# C  Technical Proofs

## C.1  Proof of Proposition 3.4

**Proposition 3.4 (restated)** (Satisfaction probability for $P_{safe}^H$). *Let $\mathcal{M}$ and $\mathcal{D}$ be the MDP and DFA from before (Defn. 3.3). For a path $\rho \in \mathcal{S}^\omega$ in the Markov chain, let $trace_H(\rho) = L(\rho[0]), L(\rho[1]) \ldots, L(\rho[H])$ be the corresponding finite word over $\Sigma = Pow(AP)$. For a given state $s \in \mathcal{S}$ the finite horizon satisfaction probability for $P_{safe}$ is defined as follows,*

$$\mathrm{Pr}^{\mathcal{M}}(s \models P_{safe}^H) := \mathrm{Pr}^{\mathcal{M}}(\rho \in \mathcal{S}^\omega \mid \rho[0] = s, trace_H(\rho) \notin \mathcal{L}(\mathcal{D}))$$

*where $H \in \mathbb{Z}_+$ is some fixed model checking horizon. Similar to before, we show that the finite horizon satisfaction probability can be written as the following bounded reachability probability,*

$$\mathrm{Pr}^{\mathcal{M}}(s \models P_{safe}^H) = \mathrm{Pr}^{\mathcal{M} \otimes \mathcal{D}}(\langle s, q_s \rangle \not\models \Diamond^{\leq H} accept)$$

*where $q_s = \Delta(\mathcal{Q}_0, L(s))$ is as before and $\Diamond^{\leq H} accept$ is the corresponding step-bounded PCTL path formula that reads, 'eventually accept in H timesteps'.*

*Proof.* Let $P_{safe}$ be a regular safety property and let $\mathcal{D} = (\mathcal{Q}, \Sigma, \Delta, \mathcal{Q}_0, \mathcal{F})$ be the DFA such that $\mathcal{L}(\mathcal{D}) = BadPref(P_{safe})$. We provide a formal definition for $P_{safe}$ and the corresponding finite horizon property $P_{safe}^H$, respectively:

$$P_{safe} = \{w \in \Sigma^\omega \mid \forall w_{pref} \in \Sigma^\omega s.t. \ w_{pref} \preceq w, w_{pref} \notin \mathcal{L}(\mathcal{D})\} \tag{9}$$

$$P_{safe}^H = \{w \in \Sigma^\omega \mid \forall w_{pref} \in \Sigma^\omega s.t. \ w_{pref} \preceq w \wedge |w_{pref}| \leq H + 1, w_{pref} \notin \mathcal{L}(\mathcal{D})\} \tag{10}$$

Let $\mathcal{M} = (\mathcal{S}, \mathcal{P}, \mathcal{P}_0, AP, L)$ be a Markov chain and consider the product Markov chain $\mathcal{M} \otimes \mathcal{D}$ from Defn. 3.2. For any path $\rho = s_0, s_1, s_2, \ldots$, there exists a unique run $q_0, q_1, q_2, \ldots$ for the trace $trace(\rho) = L(s_0), L(s_1), L(s_2) \ldots$, and denote,

$$\rho^+ = \langle s_0, q_0 \rangle, \langle s_1, q_1 \rangle, \langle s_2, q_2 \rangle \ldots \tag{11}$$

where start state is $\langle s_0, \Delta(\mathcal{Q}_0, L(s_0)) \rangle$. Before we deal with probabilities let's just consider a fixed path $\rho \in \mathcal{S}^\omega$, the finite trace $trace_H(\rho) = L(\rho[0]), L(\rho[1]) \ldots, L(\rho[H])$, the unique run $q_0, q_1, q_2, \ldots, q_H$ and the path $\rho^+ \in \Sigma^\omega \times \mathcal{Q}^\omega$ in the product Markov chain. We prove the following statement,

$$\rho \not\models P_{safe}^H \quad \text{if and only if} \quad \rho^+ \models \Diamond accept^{\leq H} \tag{12}$$

654 We start with the ($\rightarrow$) direction, in particular, $\rho \not\models P_{safe}^H$ if and only if $trace_H(\rho) \in \mathcal{L}(\mathcal{D})$. Recall
655 that by definition $\mathcal{L}(\mathcal{D}) = \{w \in \Sigma^* \mid \Delta^*(\mathcal{Q}_0, w) \in \mathcal{F}\}$, and so $trace_H(\rho) \in \mathcal{L}(\mathcal{D})$ implies that
656 $q_H = \Delta^*(\mathcal{Q}_0, trace_H(\rho)) \in \mathcal{F}$, which by construction implies that $\rho^+ \models \Diamond accept^{\leq H}$.

657 The opposite direction ($\leftarrow$) is a little more involved, in particular, $\rho^+ \models \Diamond accept^{\leq H}$ implies that
658 for the unique run $q_0, q_1, q_2, \ldots, q_H$ there exists $t \leq H$ such that $q_t \in \mathcal{F}$. We notice that since
659 $\mathcal{L}(\mathcal{D}) = BadPref(P_{safe})$ then once the DFA reaches an accepting state it will remain in an accepting
660 state for the rest of the run. Therefore, $q_t \in \mathcal{F}$ for $t \leq H$ implies that $q_H \in \mathcal{F}$. Then by definition
661 the trace $trace_H(\rho)$ that determined the unique run $q_0, q_1, q_2, \ldots, q_H$ must be in the language $\mathcal{L}(\mathcal{D})$,
662 which again by definition implies that $\rho \not\models P_{safe}^H$.

663 We now deal with the probabilities. First we note that the DFA $\mathcal{D}$ does not affect the probabilities of
664 the product Markov chain – it can be shown that for every measurable set $P$ of paths in $\mathcal{M}$,

$$\text{Pr}^{\mathcal{M}}(P) = \text{Pr}^{\mathcal{M} \otimes \mathcal{A}}(\rho^+ \mid \rho \in P) \tag{13}$$

665 see [9]. It now remains to construct this set $P$ in the proper way. In particular, if $P$ is the set of paths
666 starting in some state $s \in \mathcal{S}$ and that refute $P_{safe}$ in the next $H$ timesteps, i.e.,

$$P = \{\rho \in \mathcal{S}^\omega \mid \rho[0] = s, \{w' \in \Sigma^* \mid w_{pref} \preceq trace(\rho) \wedge |w_{pref}| \leq H + 1\} \cap \mathcal{L}(\mathcal{D}) \neq \varnothing\} \tag{14}$$

667 and $P^+$ is defined as the set of paths starting from the corresponding state $\langle s, q_s \rangle$ (where $q_s = $
668 $\Delta(\mathcal{Q}_0, L(s))$) in $\mathcal{M} \otimes \mathcal{D}$ that eventually reach an accepting state of $\mathcal{D}$ in the next $H$ steps, i.e.

$$P^+ = \{\rho^+ \in (\mathcal{S} \times \mathcal{Q})^\omega \mid \rho^+[0] = \langle s, q_s \rangle \wedge \rho^+ \models \Diamond^{\leq H} accept\} \tag{15}$$

669 Then by construction we have,

$$\text{Pr}^{\mathcal{M}}(P) = \text{Pr}^{\mathcal{M} \otimes \mathcal{D}}(\rho^+ \mid \rho[0] = s, \rho \in P) = \text{Pr}^{\mathcal{M} \otimes \mathcal{D}}(P^+) \tag{16}$$

670 Finally the probability $\text{Pr}^{\mathcal{M}}(P)$ and $\text{Pr}^{\mathcal{M}}(s \models P_{safe}^H)$ are related as follows,

$$\text{Pr}^{\mathcal{M}}(s \models P_{safe}^H) = 1 - \text{Pr}^{\mathcal{M}}(P) \tag{17}$$

$$= 1 - \text{Pr}^{\mathcal{M} \otimes \mathcal{D}}(P^+) \tag{18}$$

$$= 1 - \text{Pr}^{\mathcal{M} \otimes \mathcal{D}}(\langle s, q_s \rangle \models \Diamond^{\leq H} accept) \tag{19}$$

$$= \text{Pr}^{\mathcal{M} \otimes \mathcal{D}}(\langle s, q_s \rangle \not\models \Diamond^{\leq H} accept) \tag{20}$$

671 □

## C.2  Proof of Proposition 4.2

673 **Proposition 4.2 (restated).** *Let $P_{safe}^T$ denote the (episodic) regular safety property for a fixed episode*
674 *length $T$. Then satisfying $\text{Pr}\left(\langle s_t, q_t \rangle \models \Diamond^{\leq H} accept\right) \leq p_1$ for all $t \in [0, T]$ guarantees that*
675 $\text{Pr}(s_0 \models P_{safe}^T) \geq 1 - p_1 \cdot \lceil T/H \rceil$, *where $s_0 \sim \mathcal{P}_0$ is the initial state.*

676 *Proof.* Consider splitting up the episode in to $\lceil T/H \rceil$ chunks with length at most $H$. Let
677 $X_0, X_1, \ldots X_{\lceil T/H \rceil - 1}$ be the indicator random variables defined as follows,

$$X_i = \begin{cases} 1 & \text{if } \langle s_{i \cdot H}, q_{i \cdot H} \rangle \models \Diamond^{\leq H} accept \\ 0 & \text{otherwise} \end{cases} \tag{21}$$

678 Since $\text{Pr}(\langle s_t, q_t \rangle \models \Diamond^{\leq H} accept) \leq p_1$ for all $t \in [0, T]$ then the probability $\text{Pr}(X_i = 1) \leq p_1$. By
679 construction we have,

$$\text{if} \quad \bigcap_{i=0}^{\lceil T/H \rceil - 1} X_i = 0 \quad \text{then} \quad s_0 \models P_{safe}^T \tag{22}$$

Intuitively we satisfy $P_{safe}$ for the entire episode length if we never enter an accepting state in each of the $\lceil T/H \rceil$ chunks. The final result is then obtained by taking a union bound as follows,

$$\Pr(s_0 \models P_{safe}^T) \geq \Pr\left( \bigcap_{i=0}^{\lceil T/H \rceil - 1} X_i = 0 \right) \tag{23}$$

$$= 1 - \Pr\left( \bigcup_{i=0}^{\lceil T/H \rceil - 1} X_i = 1 \right) \tag{24}$$

$$\geq 1 - \sum_{i=0}^{\lceil T/H \rceil - 1} \Pr(X_i = 1) \tag{25}$$

$$\geq 1 - p_1 \cdot \lceil T/H \rceil \tag{26}$$

$$\tag{27}$$

$\square$

## C.3 Proof of Proposition 5.4

**Proposition 5.4 (restated).** *Let $\epsilon > 0$, $\delta > 0$, $s \in \mathcal{S}$ be given. Under Assumption 5.2, we can obtain an $\epsilon$-approximate estimate for $\Pr(\langle s, q \rangle \models \Diamond^{\leq H} accept)$ with probability at least $1 - \delta$, by sampling $m \geq \frac{1}{2\epsilon^2} \log\left(\frac{2}{\delta}\right)$ paths from the 'black box' model.*

*Proof.* In words, we estimate $\Pr(\langle s, q \rangle \models \Diamond^{\leq H} accept)$ by sampling $m$ paths from a 'black box' model of the environment dynamics. We label each path as satisfying or not and return the proportion of satisfying traces as an estimate for $\Pr(\langle s, q \rangle \models \Diamond^{\leq H} accept)$. We proceed as follows, let $\rho_1, \ldots \rho_m$ be a sequence of paths sampled from the 'black box' model and let $trace(\rho_1), \ldots trace(\rho_m)$ be the corresponding traces. Furthermore, let $X_1, \ldots, X_m$ be indicator r.v.s such that,

$$X_i = \begin{cases} 1 & \text{if } trace(\rho_1) \models \Diamond^{\leq H} accept, \\ 0 & \text{otherwise} \end{cases} \tag{28}$$

Recall that $trace(\rho_1) \models \Diamond^{\leq H} accept$ can be checked in time $O(\text{poly}(H))$. Now let,

$$\overline{X} = \frac{1}{m} \sum_{i=1}^{m} X_i \text{ where } \mathbb{E}[\overline{X}] = \Pr(\langle s, q \rangle \models \Diamond^{\leq H} accept) \tag{29}$$

then by Hoeffding's inequality [40],

$$\mathbb{P}\left[ |\overline{X} - \mathbb{E}[\overline{X}]| \geq \epsilon \right] \leq 2 \exp\left( -2m\epsilon^2 \right) \tag{30}$$

Bounding the RHS from above by $\delta$ and rearranging gives the desired result. $\square$

## C.4 Proof of Proposition 5.5

We start by introducing the following lemma.

**Lemma C.1** (Error amplification for trace distributions)**.** *Let $\widehat{\mathcal{P}} \approx \mathcal{P}$ be such that,*

$$D_{TV}\left( \mathcal{P}(\cdot \mid s), \widehat{\mathcal{P}}(\cdot \mid s) \right) \leq \alpha \; \forall s \in S \tag{31}$$

*Let the start state $s_0 \in \mathcal{S}$ be given, and let $\mathcal{P}_t(\cdot)$ and $\widehat{\mathcal{P}}_t(\cdot)$ denote the path distribution (at time $t$) for the two transition probabilities $\mathcal{P}$ and $\widehat{\mathcal{P}}$ respectively. Then the total variation distance between the two path distributions (at time $t$) are bounded as follows,*

$$D_{TV}\left( \mathcal{P}_t(\cdot), \widehat{\mathcal{P}}_t(\cdot) \right) \leq \alpha t \; \forall t \tag{32}$$

*Proof.* We will prove this fact by doing an induction on $t$. We recall that $\mathcal{P}_t(\cdot)$ and $\widehat{\mathcal{P}}_t(\cdot)$ denote the path distribution (at time $t$) for the two transition probabilities $\mathcal{P}$ and $\widehat{\mathcal{P}}$ respectively. Formally we define them as follows,

$$\mathcal{P}_t(\rho) = \Pr(s_0, \ldots, s_t \preceq \rho \mid s_0 = s, \mathcal{P}) \tag{33}$$

$$\widehat{\mathcal{P}}_t(\rho) = \Pr(s_0, \ldots, s_t \preceq \rho \mid s_0 = s, \widehat{\mathcal{P}}) \tag{34}$$

These probabilities read as follows, 'the probability of the sequence $s_0, \ldots, s_t \preceq \rho$ at time $t$', or similarly 'the probability that the sequence $s_0, \ldots, s_t$ is a prefix of $\rho$ at time $t$' Since the start state $s_0 \in \mathcal{S}$ is given we note that,

$$\mathcal{P}_0(\cdot) = \widehat{\mathcal{P}}_0(\cdot) \tag{35}$$

Before we continue with the induction on $t$ we make the following observation, for any path $\rho \in \mathcal{S}^\omega$ we have by the triangle inequality,

$$\left| \mathcal{P}_t(\rho) - \widehat{\mathcal{P}}_t(\rho) \right| = \left| \mathcal{P}(s_t \mid s_{t-1}) \mathcal{P}_{t-1}(\rho) - \widehat{\mathcal{P}}(s_t \mid s_{t-1}) \widehat{\mathcal{P}}_{t-1}(\rho) \right| \tag{36}$$

$$\leq \mathcal{P}_{t-1}(\rho) \left| \mathcal{P}(s_t \mid s_{t-1}) - \widehat{\mathcal{P}}(s_t \mid s_{t-1}) \right| + \widehat{\mathcal{P}}(s_t \mid s_{t-1}) \left| \mathcal{P}_{t-1}(\rho) - \widehat{\mathcal{P}}_{t-1}(\rho) \right| \tag{37}$$

Now we continue with the induction on $t$,

$$2 D_{TV}(\mathcal{P}_t(\cdot), \widehat{\mathcal{P}}_t(\cdot)) = \sum_{\rho \in \mathcal{S}^\omega} \left| \mathcal{P}_t(\rho) - \widehat{\mathcal{P}}_t(\rho) \right| \tag{38}$$

$$\leq \sum_{\rho \in \mathcal{S}^\omega} \mathcal{P}_{t-1}(\rho) \left| \mathcal{P}(s_t \mid s_{t-1}) - \widehat{\mathcal{P}}(s_t \mid s_{t-1}) \right|$$
$$+ \sum_{\rho \in \mathcal{S}^\omega} \widehat{\mathcal{P}}(s_t \mid s_{t-1}) \left| \mathcal{P}_{t-1}(\rho) - \widehat{\mathcal{P}}_{t-1}(\rho) \right| \tag{39}$$

$$\leq \sum_{\rho \in \mathcal{S}^\omega} \mathcal{P}_{t-1}(\rho) \cdot (2\alpha) + \sum_{\rho \in \mathcal{S}^\omega} \left| \mathcal{P}_{t-1}(\rho) - \widehat{\mathcal{P}}_{t-1}(\rho) \right| \tag{40}$$

$$= 2\alpha + 2 D_{TV}(\mathcal{P}_{t-1}(\cdot), \widehat{\mathcal{P}}_{t-1}(\cdot)) \tag{41}$$

$$\leq 2\alpha t \tag{42}$$

The final result is obtained by an induction on $t$ where the base case comes from $\mathcal{P}_0(\cdot) = \widehat{\mathcal{P}}_0(\cdot)$. $\qquad \square$

**Proposition 5.5 (restated).** *Let $\epsilon > 0$, $\delta > 0$, $s \in \mathcal{S}$ and horizon $H \geq 1$ be given. Under Assumption 5.3 we can make the following two statements:*

*(1) We can obtain an $\epsilon$-approximate estimate for $\Pr(\langle s, q \rangle \models \Diamond^{\leq H} accept)$ with probability 1 by exact model checking with the transition probabilities of $\widehat{\mathcal{P}}_\pi$ in time $\mathcal{O}(poly(size(\mathcal{M}_\pi \otimes \mathcal{D})) \cdot H)$.*

*(2) We can obtain an $\epsilon$-approximate estimate for $\Pr(\langle s, q \rangle \models \Diamond^{\leq H} accept)$ with probability at least $1 - \delta$, by sampling $m \geq \frac{2}{\epsilon^2} \log \left( \frac{2}{\delta} \right)$ paths from the 'approximate' dynamics model $\widehat{\mathcal{P}}_\pi$.*

*Proof.* We start by proving statement (1) and then statement (2) will follow quickly. First let $\Pr(\langle s, q \rangle \models \Diamond^{\leq H} accept)$ and $\widehat{\Pr}(\langle s, q \rangle \models \Diamond^{\leq H} accept)$ denote the acceptance probabilities for the two transition probabilities $\mathcal{P}$ and $\widehat{\mathcal{P}}$ respectively. We also let $g(\cdot)$ and $\widehat{g}(\cdot)$ denote the average trace distribution (over the next $H$ timesteps) for the two transition probabilities $\mathcal{P}$ and $\widehat{\mathcal{P}}$ respectively, where,

$$g(\rho) = \frac{1}{H} \sum_{t=1}^{H} \mathcal{P}_t(\rho) \tag{43}$$

$$\widehat{g}(\rho) = \frac{1}{H} \sum_{t=1}^{H} \widehat{\mathcal{P}}_t(\rho) \tag{44}$$

Before we continue with the proof of (1) we make the following observations,

- $\displaystyle \max_{\langle s,q\rangle} \left| \Pr(\langle s,q\rangle \models \Diamond^{\leq H} accept) - \widehat{\Pr}(\langle s,q\rangle \models \Diamond^{\leq H} accept) \right| \leq 1$

- Let $f(x) : x \in \mathcal{X} \to [0,1]$ be a real-valued function. Let $\mathcal{P}_1(\cdot)$ and $\mathcal{P}_2(\cdot)$ be probability distributions over the space $\mathcal{X}$, then.

$$\left| \mathbb{E}_{x\sim\mathcal{P}_1(\cdot)}[f(x)] - \mathbb{E}_{x\sim\mathcal{P}_2(\cdot)}[f(x)] \right| \leq D_{TV}(\mathcal{P}_1(\cdot), \mathcal{P}_2(\cdot))$$

We continue by showing the following,

$$\left| \Pr(\langle s,q\rangle \models \Diamond^{\leq H} accept) - \widehat{\Pr}(\langle s,q\rangle \models \Diamond^{\leq H} accept) \right| \tag{45}$$

$$= \left| \mathbb{E}_{\rho\sim g} \left[ 1\left[\langle s,q\rangle \models \Diamond^{\leq H} accept\right]\right] - \mathbb{E}_{\rho\sim\widehat{g}} \left[ 1\left[\langle s,q\rangle \models \Diamond^{\leq H} accept\right]\right] \right| \tag{46}$$

$$\leq D_{TV}(g(\cdot), \widehat{g}(\cdot)) \tag{47}$$

$$= \frac{1}{2} \sum_{\rho\in\mathcal{S}^\omega} |g(\rho) - \widehat{g}(\rho)| \tag{48}$$

$$= \frac{1}{2H} \sum_{\rho\in\mathcal{S}^\omega} \left| \sum_{t=1}^{H} \mathcal{P}_t(\rho) - \widehat{\mathcal{P}}_t(\rho) \right| \tag{49}$$

$$\leq \frac{1}{2H} \sum_{t=1}^{H} \left| \sum_{\rho\in\mathcal{S}^\omega} \mathcal{P}_t(\rho) - \widehat{\mathcal{P}}_t(\rho) \right| \tag{50}$$

$$\leq \frac{1}{2H} \sum_{t=1}^{H} H(\epsilon/H) \tag{51}$$

$$= \epsilon/2 \tag{52}$$

$$\tag{53}$$

The first inequality (Eq. 47) comes from our earlier observations. The second inequality (Eq. 50) is straightforward and the final inequality (Eq. 51) is obtained by applying Lemma C.1 and Assumption 5.3. We note that this result is similar to the *simulation lemma* [48], which has been proved many times for several different settings [1, 16, 47, 57].

This concludes the proof of statement (1), since we have shown that $\widehat{\Pr}(\langle s,q\rangle \models \Diamond^{\leq H} accept)$ is an $\epsilon/2$-approximate estimate of $\Pr(\langle s,q\rangle \models \Diamond^{\leq H} accept)$, under the Assumption 5.3.

The proof of statement (2) follows quickly. We have established that,

$$\left| \Pr(\langle s,q\rangle \models \Diamond^{\leq H} accept) - \widehat{\Pr}(\langle s,q\rangle \models \Diamond^{\leq H} accept) \right| \leq \epsilon/2 \tag{54}$$

It remains to obtain an $\epsilon/2$-approximate estimate of $\widehat{\Pr}(\langle s,q\rangle \models \Diamond^{\leq H} accept)$. By using the same reasoning as in the proof of Proposition 5.4. We can obtain an $\epsilon/2$-approximate estimate of $\widehat{\Pr}(\langle s,q\rangle \models \Diamond^{\leq H} accept)$ by sampling $m$ paths, $\rho_1, \ldots \rho_m$, from the approximate dynamics model $\widehat{\mathcal{P}}$. Then provided,

$$m \geq \frac{2}{\epsilon^2} \log\left(\frac{2}{\delta}\right) \tag{55}$$

with probability $1 - \delta$ we can obtain $\epsilon/2$-approximate estimate of $\widehat{\Pr}(\langle s,q\rangle \models \Diamond^{\leq H} accept)$ and by extension an $\epsilon$-approximate estimate of $\Pr(\langle s,q\rangle \models \Diamond^{\leq H} accept)$. This concludes the proof. $\qquad\square$

## C.5 Proof of Theorem 6.5

**Theorem 6.5 (restated).** *Under Assumption 6.3 and 6.4, and provided that every state action pair $(s,a) \in \mathcal{S} \times \mathcal{A}$ has been visited at least $\mathcal{O}\left(\frac{H^2|\mathcal{S}|^2}{\epsilon^2} \log\left(\frac{|\mathcal{A}||\mathcal{S}|^2}{\delta}\right)\right)$ times. Then with probability $1 - \delta$ the system satisfies the constraints of Problem 4.1, independent of the 'task policy'.*

*Proof.* We split the proof up in to three parts, **(1)**, **(2)** and **(3)**. In part **(1)** we show that the given sample complexity bound gives us an approximate model of the environment dynamics with high

probability. In part **(2)** we use our assumptions to reason about the probabilistic recoverability of the system when it enters a critical state. In part **(3)** we put everything together and deal with approximation error $\epsilon$ the remaining failure probability that are both unavoidable for the statistical model checking procedures used to shield the system.

**(1)** We show that the following holds with probability $1 - \delta/2$,

$$D_{TV}\left(\mathcal{P}_\pi(\cdot \mid s), \widehat{\mathcal{P}}_\pi(\cdot \mid s)\right) \leq \epsilon/H \; \forall s \in \mathcal{S} \tag{56}$$

when every state action pair $(s, a) \in \mathcal{S} \times \mathcal{A}$ has been visited at least,

$$\mathcal{O}\left(\frac{H^2|\mathcal{S}|^2}{\epsilon^2} \log\left(\frac{|\mathcal{A}||\mathcal{S}|^2}{\delta}\right)\right)$$

times. First we let $\#(s, a)$ denote the total number of times that $(s, a)$ has been observed, similarly we let $\#(s', s, a)$ denote the total number of times that $(s', s, a)$ has been observed. The maximum likelihood estimate for the unknown probability $\mathcal{P}(s' \mid, s, a)$ is $\widehat{\mathcal{P}}(s' \mid s, a) = \#(s', s, a)/\#(s, a)$. Let us fix some $(s, a) \in \mathcal{S} \times \mathcal{A}$, and $s' \in \mathcal{S}$, we let $p_{s'} = \mathcal{P}(s' \mid s, a)$ denote the true probability of transitioning to $s'$ from $(s, a)$ and we let $\hat{p}_{s'} = \#(s', s, a)/\#(s, a)$ denote our estimate. We note that $\mathbb{E}[\hat{p}_{s'}] = p_{s'}$, i.e. $\hat{p}_{s'}$ is an unbiased estimator for $p_{s'}$. Let $m = \#(s, a)$ also be the number of times that $(s, a)$ has been observed, then by Hoeffding's inequality [40] we have,

$$\mathbb{P}\left[|p_{s'} - \hat{p}_{s'}| \geq \frac{\epsilon}{H|\mathcal{S}|}\right] \leq 2\exp\left(-2m\frac{\epsilon^2}{H^2|\mathcal{S}|^2}\right) \tag{57}$$

Bounding the LHS from above by $1 - \delta/2(|\mathcal{A}||\mathcal{S}|^2)$ and rearranging gives the following lower bound for $m$,

$$m \geq \frac{H^2|\mathcal{S}|^2}{2\epsilon^2} \log\left(\frac{4|\mathcal{A}||\mathcal{S}|^2}{\delta}\right) \tag{58}$$

Taking a union bound over all $(s', s, a) \in \mathcal{S} \times \mathcal{S} \times \mathcal{A}$, then for all state action pairs $(s, a) \in \mathcal{S} \times \mathcal{A}$ we have the following with probability at least $1 - \delta$.

$$2D_{TV}\left(\mathcal{P}(\cdot \mid s, a), \widehat{\mathcal{P}}(\cdot \mid, s, a)\right) = \sum_{s' \in S}|p_{s'} - \hat{p}_{s'}| \leq \sum_{s' \in \mathcal{S}}\frac{\epsilon}{H|\mathcal{S}|} \leq \epsilon/H \tag{59}$$

Now fix some $s \in \mathcal{S}$ and we observe the following,

$$2D_{TV}\left(\mathcal{P}_\pi(\cdot \mid s), \widehat{\mathcal{P}}_\pi(\cdot \mid s)\right) = \sum_{s' \in \mathcal{S}}|\mathcal{P}_\pi(s' \mid s) - \widehat{\mathcal{P}}_\pi(s' \mid s)| \tag{60}$$

$$= \sum_{s' \in \mathcal{S}}\sum_{a \in \mathcal{A}}|\mathcal{P}(s' \mid s, a)\pi(a \mid s) - \widehat{\mathcal{P}}(s' \mid s, a)\pi(a \mid s)| \tag{61}$$

$$= \sum_{a \in \mathcal{A}}\pi(a \mid s)\sum_{s' \in \mathcal{S}}|\mathcal{P}(s' \mid s, a) - \widehat{\mathcal{P}}(s' \mid s, a)| \tag{62}$$

$$= \sum_{a \in \mathcal{A}}\pi(a \mid s)2D_{TV}\left(\mathcal{P}(\cdot \mid s, a), \widehat{\mathcal{P}}(\cdot \mid, s, a)\right) \tag{63}$$

$$\leq \epsilon/H \tag{64}$$

Thus with probability at least $1 - \delta/2$ we have for all $s \in \mathcal{S}$ that,

$$D_{TV}\left(\mathcal{P}_\pi(\cdot \mid s), \widehat{\mathcal{P}}_\pi(\cdot \mid s)\right) \leq \epsilon/H \tag{65}$$

**(2)** Using Assumption 6.3 and 6.4 we can argue about the safety of the system. Suppose firstly, that we can check the condition $\Pr(\langle s, q \rangle \models \Diamond^{\leq H} accept) \leq p_1$, precisely and without any failure probability (we will deal with statistical model checking in part **(3)**). From any non-critical state we can transition arbitrarily to a critical state, although under Assumption 6.3 this critical state is not irrecoverable with probability $\geq p_1$. We now consider the following two cases:

(i) $\Pr(\langle s, q \rangle \models \Diamond^{\leq H} accept) \leq p_1$ under the 'task' policy.

771    (ii) $\Pr(\langle s, q \rangle \models \Diamond^{\leq H} accept) > p_1$ under the 'task' policy.

772 For case (i) we can safely use the 'task' policy and return to a non-critical state within $H$ timesteps
773 with probability at least $1 - p_1$. For case (ii) we deploy the 'safe' policy and under Assumption 6.4
774 we can return to a non-critical state within $H$ timesteps with probability at least $1 - p_1$. We have now
775 established an invariant, since from every non-critical state we can return to a non-critical state with
776 probability $1 - p_1$ and thus satisfy $\Pr(\langle s, q \rangle \models \Diamond^{\leq H} accept) \leq p_1$ at every timestep $t \in [0, T]$.

777 **(3)**    We now make a similar argument but for the statistical model checking procedure where we
778 can only obtain an $\epsilon$-approximate estimate for the probability $\Pr(\langle s, q \rangle \models \Diamond^{\leq H} accept)$ with high
779 probability. Let us denote our $\epsilon$-approximate estimate $\widehat{\Pr}(\langle s, q \rangle \models \Diamond^{\leq H} accept)$, rather than check
780 the condition $\Pr(\langle s, q \rangle \models \Diamond^{\leq H} accept) \leq p_1$, we can check condition $\widehat{\Pr}(\langle s, q \rangle \models \Diamond^{\leq H} accept) \leq$
781 $p_1 - \epsilon$, and if $\widehat{\Pr}(\langle s, q \rangle \models \Diamond^{\leq H} accept)$ is indeed an $\epsilon$-approximate estimate then this guarantees
782 $\Pr(\langle s, q \rangle \models \Diamond^{\leq H} accept) \leq p_1$. Consider the following two cases:

783 (i) Our estimate $\widehat{\Pr}(\langle s, q \rangle \models \Diamond^{\leq H} accept) \leq p_1 - \epsilon$

784 (ii) Our estimate $\widehat{\Pr}(\langle s, q \rangle \models \Diamond^{\leq H} accept) > p_1 - \epsilon$

785 For case (i) we can safely use the 'task' policy and return to a non-critical state within $H$ timesteps
786 with probability at least $1 - p_1$. For case (ii) we deploy the 'safe' policy and under Assumption 6.4
787 we can return to a non-critical state within $H$ timesteps with probability at least $1 - p_1$. Again we
788 have established an invariant, since from every non-critical state we can return to a non-critical state
789 with probability $1 - p_1$ and thus satisfy $\Pr(\langle s, q \rangle \models \Diamond^{\leq H} accept) \leq p_1$ at every timestep $t \in [0, T]$.

790 We still need to deal with the failure probability of the statistical model checking procedure at
791 each timestep, by choosing failure probability $1 - \delta/2T$ we can guarantee (by a union bound) an
792 $\epsilon$-approximate estimate for each timestep with probability $1 - \delta/2$. Finally, taking a union bound
793 over part **(1)** and **(2)** gives the desired total failure probability $1 - \delta$.

794          $\square$

# D    Environment Details

## D.1    Colour Gridworld

797 The colour gridworld environment is a simple $9 \times 9$ grid, with
798 state space $|\mathcal{S}| = 81$ and action space $|\mathcal{A}| = 5$, where each action
799 corresponds to the following movements: *Left*,*Right*, *Up*, *Down*, *Stay*.
800 The objective is to navigate from the start state in one corner of the
801 grid, to the goal state in the other corner, after reaching the goal state
802 the agent is then sent back to the start state. The agent must navigate
803 to the goal state as many times as possible in a fixed episode length
804 of $T = 1000$. The reward function is a sparse reward that gives the
805 agent $+1$ reward for reaching the goal and $0$ otherwise. When the
806 environment is fully deterministic the maximum achievable reward
807 is 58.

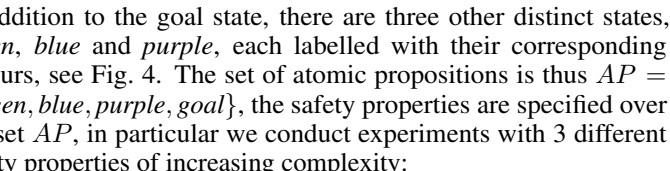

Figure 4: Colour gridworld environment. Top left hand corner (*agent*) is the start position. The agent must navigate to the *goal* position in the bottom right hand corner of gridworld. The coloured states labelled *blue*, *green* and *purple* correspondingly.

808 In addition to the goal state, there are three other distinct states,
809 *green*, *blue* and *purple*, each labelled with their corresponding
810 colours, see Fig. 4. The set of atomic propositions is thus $AP =$
811 $\{green, blue, purple, goal\}$, the safety properties are specified over
812 the set $AP$, in particular we conduct experiments with 3 different
813 safety properties of increasing complexity:

814      • (1) $\square \neg green$

815      • (2) $\square goal \rightarrow \Diamond^{\leq 10} blue$

816      • (3) $\square goal \rightarrow \Diamond^{\leq 10} \square^{\leq 5} purple$

817 Property (1) is a simple invariant property $P_{inv}(\neg green)$ that states the green state must always be
818 avoided. Property (2) and (3) are more complex safety properties that interfere with the goal state. In

particular, property (2) states that once the *goal* state is reached then the *blue* state must be reached within 10 steps, this actually has no direct consequences on the maximum reward achievable but may interfere with convergence as the goal state seemingly leads to a high penalty if the *blue* state is not reached.

Property (3) states that once the *goal* state is reached then the *purple* state must be reached within 10 steps and then *purple* must hold for the next 5 timesteps. In safety property both interferes with the goal and has direct consequences on the maximum achievable reward as staying in purple for 5 steps does not lead to progress towards the goal state. In terms of the size of the DFA $|\mathcal{Q}|$, property (1) is an invariant so the cost function is Markov and the size of the DFA is 2, for property (2) and (3) the size of the DFA is 12 and 62 respectively.

Each of the safety properties are tested with the corresponding $p$ value for the environment, detailed in Table 1, which is repeated here for reference. The $p$ value corresponds to the level of stochasticity in the environment. In particular, if $p = 0.25$ then there is a 25% chance of the agents action being overridden with another random action chosen uniformly. Given

Table 2: Safety properties and $p$ value

| property | rand. act. $p$ |
|---|---|
| (1) $\Box \neg green$ | 0.25 |
| (2) $\Box goal \rightarrow \Diamond^{\leq 10} blue$ | 0.25 |
| (3) $\Box goal \rightarrow \Diamond^{\leq 10} \Box^{\leq 5} purple$ | 0.1 |

the environment is stochastic then it is difficult to satisfy the safety properties with probability 1. Through preliminary statistical analysis we computed the maximum satisfaction probabilities for each property, to help inform an appropriate $p$ value to test with. With $p = 0.25$, property (1) can be satisfies with very high probability close to 1, while still achieving maximum reward. With $p = 0.25$ property (2) can be satisfied with probability $\approx 0.93$ while still achieving maximum reward. With $p = 0.1$ property (3) can be satisfied with probability $\approx 0.75$ while still achieving good reward.

**Hyperparameter settings.** We discuss some of the hyperparameter settings for our shielding approach that are not detailed in Table 5.

Property (1): we use a model checking horizon of $H = 3$, and probability threshold $p_1 = 1.0$, with the number of samples $m = 4096$, we can obtain a roughly $\epsilon = 0.05$ approximate estimate of the finite horizon satisfaction probability with failure probability $\delta = 0.01$.

Property (2): we use a model checking horizon of $H = 10$, and probability threshold $p_1 = 0.9$, with the number of samples $m = 8192$, we can obtain a roughly $\epsilon = 0.05$ approximate estimate of the finite horizon satisfaction probability with a smaller failure probability $\delta = 0.001$.

Property (3): again we use a model checking horizon of $H = 10$, and probability threshold $p_1 = 0.6$, with the number of samples $m = 1024$, we can obtain roughly a $\epsilon = 0.1$ approximate estimate of the finite horizon satisfaction probability with failure probability $\delta = 0.01$.

**Extended discussion of results.** First we provide slightly larger figures that than provided in the main paper, see Figure 5.

In general we observe that our shielding method is able to effectively trade-off reward and safety, in all cases converging to a system that obtains superior or comparable performance with the baseline. For property (1) we might expect our method to be able to recover the optimal policy that avoids the green state, it is clear in this case that the shielding procedure has harmed convergence and perhaps further investigation and hyperparameter tuning will encourage improvements. For property (2) and (3) the results are what we expect – we can recover the best policy that satisfies the step-wise bounded safety property with the desired probability $p_1$.

The intuitive reason for why simply penalising Q-learning doesn't work, is that tuning the cost coefficient $C$ is challenging for stochastic environments, where safety cannot be enforced 'almost surely' (with probability 1), and the precise value of $C$ offers little to no semantic meaning. For different levels of stochasticity $p$ values it is hard to know what desired level of safety we can achieve while still converging to a high reward policy, making tuning $C$ even harder without knowing more about the structure of the environment. In Appendix F we study more closely the effect of $C$ and $p$. Furthermore, we note te sensitivity of our method to the chosen model checking horizon $H$. In particular, if $H$ is too large we might expect the system to be overly conservative, we also address this in more detail in Appendix F.

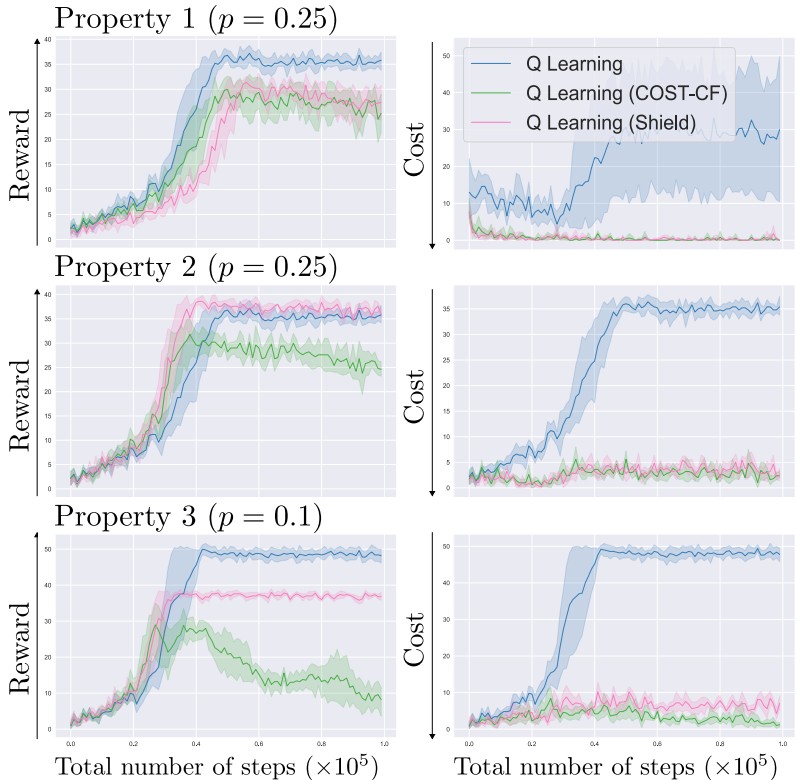

Figure 5: Episode reward and cost for tabular RL 'colour' gridworld environment.

## D.2 Atari Seaquest

Our DreamerV3 [34] based shielding procedure is tested on Atari Seaquest, provided as part of the Arcade Learning Environment (ALE)[10, 50]. Seaquest is a partially observable environment meaning we do not have direct access to the underlying state space $\mathcal{S}$, we are however provided with observations $o \in O$ as pixel images which correspond to $64 \times 64 \times 3$ tensors. Fortunately DreamerV3 is specifically designed to operate in visual settings and is able to effectively learn a predictive world model that closely approximate the environment dynamics. The action space of Seaquest is finite, specifically $|\mathcal{A}| = 18$,

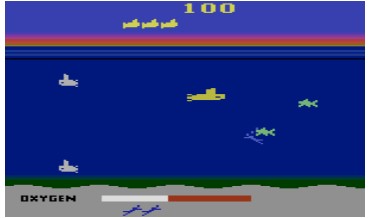

Figure 6: Atari Seaquest environment [10, 50]. The goal is to rescue divers (*small blue people*), while shooting enemy *sharks* and *submarines*.

where each action corresponds to a joystick movement and fire button interaction. Rewards are obtained by 'shooting' an enemy shark or submarine, or by rescuing divers and returning them to the surface. In addition, the agent must manage its oxygen resources and avoid being hit by sharks and the enemy submarines which fire back, see Fig. 6. The environment is also made stochastic by using 'sticky actions' [50], where the agents previous action is repeated with probability $p = 0.25$.

In terms of safety properties we experiment with the following two properties,

- (1) $(\Box \neg surface \rightarrow \Box(surface \rightarrow diver)) \wedge (\Box \neg out\text{-}of\text{-}oxygen) \wedge (\Box \neg hit)$

- (2) $\Box diver \wedge \neg surface \rightarrow \Diamond^{\leq 30} surface$

Property (1) states that after diving (i.e. not *surface*), the agent must only *surface* with a *diver* on board, and never run *out-of-oxygen* and never get *hit* by an enemy. The size of the DFA for this property is $|\mathcal{D}| = 4$. Property (2) states that once a *diver* is on board the agent must *surface* within 30 timesteps (i.e. rescue the diver).

**Hyperparameter settings.** For our shielding approach almost all the hyperparameters are specified in Appendix E. The only hyperparameter that varies is the model checking horizon $H$. For property (1) we use $H = 30$, empirically this seems adequate enough to avoid running *out-of-oxygen* and begin surfacing in enough time. For property (2) we use $H = 50$, this is to avoid picking up a *diver* at the bottom of the ocean where it may not be possible to return to the surface in 30 timesteps.

**Extended discussion of results.** First we provide slightly larger figures that than provided in the main paper, see Figure 7

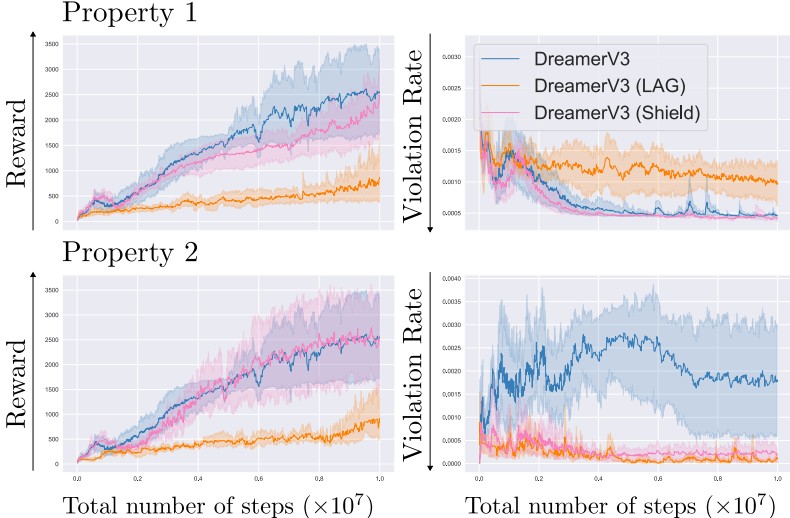

Figure 7: Episode reward and violation rate for deep RL Atari Seaquest.

For both safety properties DreamerV3 with shielding obtains comparative performance in terms of reward with the unmodified DreamerV3 baseline. Of course this baseline entirely ignores the safety properties and simply maximizes reward. We remark on the differences between the safety properties themselves, property (1) in particular specifies the natural safety properties of the environment, since violating property (1) results in a death, the agent only start with 4 lives (and can gain one more ever 10000 points) and so satisfying property (1) is beneficial for long term reward, short the behaviour satisfying property (1) is correlated with higher reward and we might expect the globally optimal policy in the environment to never violated property (1). Property (2) specifies that once a diver is recovered the submarine must return to the surface in 30 timesteps, we would not expect that the globally optimal policy satisfies this property (2) rather we would expect to converge to a locally optimal policy satisfying property (2) while still obtaining good reward.

With respect to the baseline DreamerV3 (LAG) which has access to the cost function, we see that in both cases it fails to reliable learn a safe policy that simultaneously maximizes reward. For property (2) DreamerV3 (LAG) appear to do slightly better in terms of safety, however when qualitatively inspecting the runs for property (2) we see the DreamerV3 (LAG) agent intentionally get hit by enemy submarines/sharks to re-spawn on the surface without actually having to navigate there. This may be a more effective way to satisfy the safety property with high probability but it clearly leads to worse long term reward.

# E Hyperparameters & Implementation Details

## E.1 Access to Code

To maintain a high standard of anonymity we provide code for the experiments run on 'colour' gridworld as supplementary material, rather than through GitHub. The colour gridworld environment is implemented with the Gym [14] interface. Tabular Q-learning is implemented with *numpy* in *Python*, the model checking procedures (both exact and Monte Carlo) are implemented with JAX [12] which supports vectorized computation on GPU and CPU. The code for the Atari Seaquest experiments

are not currently available, although our code base was heavily derived from the code base for *Approximate Model-based Shielding* (AMBS) [30], see `https://github.com/sacktock/AMBS` (MIT License).

**Training details.** For collecting both sets of experiments we has access to 2 Nvidia Tesla A30 (24GB RAM) GPU and a 24-core/48 thread Intel Xeon CPU each with 32GB RAM. For the 'colour' gridworld experiments each run can take several minutes up to a day depending on which property is being tested, for example one run for property (3) can take roughly 1.5 days as the product state space is fairly large. For the Atari Seaquest experiments each run can take 8 hours to 1 day depending on the precise configuration of DreamerV3, in general we see a slow down of $\times 2$ when using shielding compared to the unmodified DreamerV3 baseline. Memory requirements may differ depending on the DreamerV3 configuration used, for the *xlarge* DreamerV3 configuration 32GB of GPU memory should suffice.

**Statistical significance.** Error bars are provided for each of our experiments. In particular, we report 5 random initializations (seeds) for each experiment, the error bars are non-parametric (bootstrap) 95% confidence intervals, provided by `seaborn.lineplot` with default parameters: `errorbar=('ci', 95)`, `n_boot=1000`. The error bars capture the randomness in the initialization of the DreamerV3 world model and policy parameters, the randomness of the environment and any randomness in the batch sampling.

 **E.2 Colour Gridworld**

Table 3: Q-learning

| Name | Symbol | value |
|------|--------|-------|
| Learning rate | $\alpha$ | 0.1 |
| Discount factor | $\gamma$ | 0.95 |
| Exploration type | - | Boltzmann |
| Temperature | $\tau$ | 0.05 |

Table 4: Q-learning with counter factual experiences [43]

| Name | Symbol | value |
|------|--------|-------|
| Learning rate | $\alpha$ | 0.1 |
| Discount factor | $\gamma$ | 0.95 |
| Exploration type | - | Boltzmann |
| Temperature | $\tau$ | 0.05 |
| Cost coefficient | $C$ | 10.0 |

Table 5: Q-learning with shielding (Algorithm 1)

| Name | Symbol | value |
|------|--------|-------|
| Model checking type | - | *Monte-Carlo* |
| Approximate model | - | *True* |
| Shielding | - | *Task* |
| Number of samples | $m$ | varies |
| Approximation error | $\epsilon$ | varies |
| Failure probability | $\delta$ | varies |
| Model checking horizon | $H$ | varies |
| Satisfaction prob. | $p$ | varies |
| Prior | - | *uninformative* |

'Task policy' $\pi_{task}$

See Q-learning (Table 3)

...

'Backup policy' $\pi_{safe}$

See Q-learning with counter factual experiences (Table 4)

...

 **E.3 Atari Seaquest**

Table 6: DreamerV3 [34]

| Name | Symbol | value |
|---|---|---|
| General | | |
| Replay capacity | $|D|$ | $10^6$ |
| Batch size | $|B|$ | 16 |
| Batch length | - | 64 |
| Number of envs | - | 8 |
| Train ratio | - | 64 |
| Number of MLP layers | - | 5 |
| Number of MLP units | - | 1024 |
| Activation | - | LayerNorm + SiLU |
| World Model | | |
| Configuration size | - | medium |
| Number of latents | - | 32 |
| Classes per latent | - | 32 |
| Number of layers | - | 3 |
| Number of hidden units | - | 640 |
| Number of recurrent units | - | 1024 |
| CNN depth | - | 48 |
| RSSM loss scales | $\beta_{\text{pred}}, \beta_{\text{dyn}}, \beta_{\text{rep}}$ | 1.0, 0.5, 0.1 |
| Predictor loss scales | $\beta_o, \beta_r, \beta_c, \beta_\gamma$ | 1.0, 1.0, 1.0, 1.0 |
| Learning rate | - | $10^{-4}$ |
| Adam epsilon | $\epsilon_{\text{adam}}$ | $10^{-8}$ |
| Gradient clipping | - | 1000 |
| Actor Critic | | |
| Imagination horizon | $H$ | 15 |
| Discount factor | $\gamma$ | 0.997 |
| TD lambda | $\lambda$ | 0.95 |
| Critic EMA decay | - | 0.98 |
| Critic EMA regularizer | - | 1 |
| Return norm. scale | $S_{\text{reward}}$ | $\text{Per}(R, 95) - \text{Per}(R, 5)$ |
| Return norm. limit | $L_{\text{reward}}$ | 1 |
| Return norm. decay | - | 0.99 |
| Actor entropy scale | $\eta_{\text{actor}}$ | $3 \cdot 10^{-4}$ |
| Learning rate | - | $3 \cdot 10^{-5}$ |
| Adam epsilon | $\epsilon_{\text{adam}}$ | $10^{-5}$ |
| Gradient clipping | - | 100 |

Table 7: Augmented Lagrangian [7, 41, 72]

| Name | Symbol | value |
|------|--------|-------|
| Augmented Lagrangian | | |
| Penalty multiplier | $\mu_k$ | $5 \cdot 10^{-9}$ |
| Initial Lagrange multiplier | $\lambda^k$ | 0.01 |
| Penalty power | $\sigma$ | $10^{-6}$ |
| Cost coefficient | $C$ | 1.0 |
| Cost threshold | $d$ | 1.0 |
| Penalty Critic | | |
| See 'Actor Critic' in Table 6 | | |
| ... | | |

Table 8: DreamerV3 with Shielding (Algorithm 5)

| Name | Symbol | value |
|------|--------|-------|
| Shielding | | |
| Approximation error | $\epsilon$ | 0.09 |
| Number of samples | $m$ | 512 |
| Failure probability | $\delta$ | 0.01 |
| Look-ahead/shielding horizon | $H$ | varies |
| Satisfaction prob. | $p$ | 0.9 |
| Cost coefficient | $C$ | 10 |
| 'Task policy' | | |
| See 'Actor Critic' in Table 6 | | |
| ... | | |
| 'Backup policy' | | |
| See 'Actor Critic' in Table 6 | | |
| ... | | |

# F  Ablation Studies

In this section we provide several ablation studies for the 'colour' gridworld environment. We test the most significant hyperparameters and algorithmic components of our method including the baseline (Q-learning with penalties). In particular we demonstrate the counter factual experiences is crucial for learning the safety properties of the environment when the size of the corresponding DFA is non trivial. We also experiment with using exact model checking – demonstrating that we don't loose much by using statistical model checking procedures. Furthermore, we experiment with the cost coefficient $C$, the model checking horizon $H$ and the level of stochasticity $p$.

## F.1  Counter factual experiences

We run our method and the baseline (Q-learning with penalties) without counterfactual experiences to train the 'backup policy' or penalized task policy (baseline).

For property (2) and (3) we see a significant drop in safety performance, since learning to respect the safety property over the much larger product state space will require much more experience and without exploiting the structure of the DFA (using counter factual experiences) to generate synthetic data the task behaviour will be much more quickly learnt. For property (1), the invariant property, we observe identical performance as the DFA is trivial (only 2 states), and so counter factual experiences is essentially redundant in this case.

## F.2  Exact model checking

We run our method (Shielding) with two different configurations: exact model checking with the 'approximate' transition probabilities (learning from experience) and exact model checking with the 'true' transition probabilities. We compare these two methods to the configuration used in the main paper: Monte Carlo (statistical) model checking with the learned transition probabilities.

In all cases we see that Shield (MC-Approx) obtains almost identical performance to Shield (Exact-True), which demonstrates that we don't loose much by statistical model checking with the learned probabilities, when for example we don't have access to the transition probabilities ahead of time, or the MDP is too large to exact model check. We see some variance with Shield (Exact-Approx), which can be explained by sub-optimal convergence in terms of reward, although note that the safety performance is consistent with the other configurations. Perhaps exact model checking with an inaccurate model of the transition probabilities restricts exploration to areas of the state space that are actually safe.

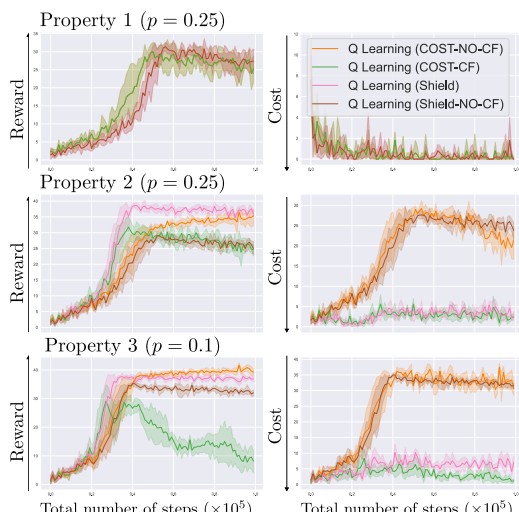

Figure 8: Episode reward and cost for Q-learning (Shield) and Q-learning (COST-CF) with and without counterfactual experiences (CF).

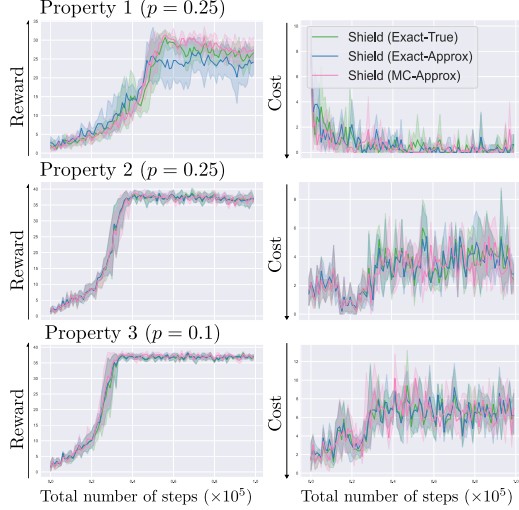

Figure 9: Episode reward and cost for Shield (Exact-True) – exact model checking with the 'true' probabilities, Shield (Exact-Approx) - exact model checking with the learning transition probabilities, and Shield (MC-Approx) – from the main paper.

### F.3 Cost coefficient $C$

We experiment with different values for the cost coefficient $C$ used for our baseline (Q-learning with penalties). In particular, we use $C \in \{0.1, 1.0, 10.0, 100.0\}$, we expect that a larger cost coefficient will penalize unsafe behaviour more harshly and result in 'safer' behaviour (i.e., fewer safety-property violations).

Unsurprisingly, across the board, by increasing the cost coefficient $C$ we obtain a policy that has fewer safety-property violations. The improved 'safety performance' is of course at the expense of reward or task performance, this is a trade-off we would expect. In particular for $C = 100.0$ we see that the learned policy essentially avoids the goal state (achieving zero reward) all but guaranteeing safety (no safety-violations). The purpose of this ablation study is to demonstrate that while we can achieve any desired level of safety by tuning the cost coefficient $C$, the actual value of $C$ offers little to no semantic meaning for the probability of violating the safety property.

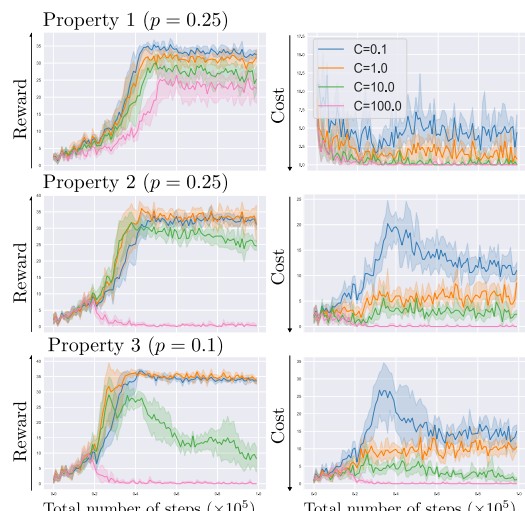

Figure 10: Episode reward and cost for Q-learning (COST-CF) – baseline from the main paper, with different cost coefficients $C$.

### F.4 Model checking horizon $H$

As was alluded to in the main paper, our method can be very sensitive to the model checking horizon (hyperparameter) $H$. In particular, if $H$ is too large then we might expect the system to exhibit overly conservative behaviour. As a rule of thumb we suggest that $H$ should be set to roughly the shortest path in the DFA from the initial state to an accepting state – this can easily be computed by using Dijkstra's (shortest-path) algorithm. In this ablation we experiment with much larger $H$ than recommended. This significantly impacts the performance of our proposed approach. However, we do propose a solution, Q-learning (Shield-Rec) which in short, checks that the action proposed by the 'task policy' is recoverable with the 'backup policy', or in other words by playing with the action $a \sim \pi_{task}$ proposed by the 'task policy' We can still satisfy $\Pr(\langle s, q \rangle \models \Diamond^{\leq H} accept) \leq p_1$ by using the 'backup policy' after playing $a$.

In general we observe that when $H$ is too large our original method (Shield) is overly conservative, sacrificing reward or task performance for safety guarantees. Our proposed solution (Shield-Rec) is alleviates this issue partly, providing reasonable safety performance and comparable task performance. We note that this solution is clearly not perfect as is it appears to be slightly more permissive allowing more safety-violations than necessary. More investigation into this framework would be interesting future work, and perhaps more hyperparameter tuning, specifically by tuning $p_1$, could improve this method. The goal would be to obtain an algorithm that is not overly sensitive to $H$, and as long as $H$ is sufficiently big to guarantee safety we don't see much performance degradation by further increasing $H$.

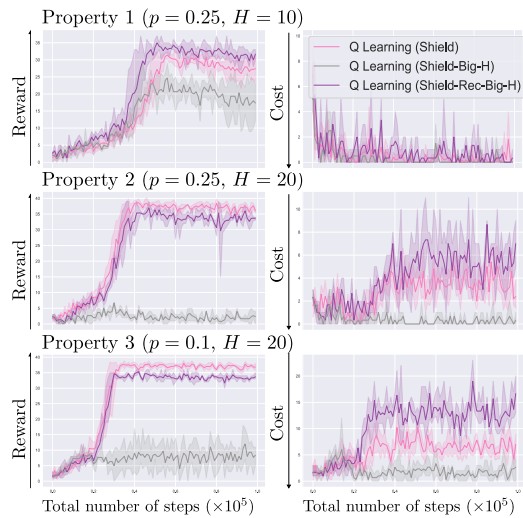

Figure 11: Episode reward and cost for Q-learning (Shield) - from the main paper, Q-learning (Shield) with bigger $H$ and Q-learning (Shield-Rec) with bigger $H$.

 **F.5   Level of stochasticity** $p$

1047 Finally we investigate the effect of the level of stochasticity of the environment. Specifically, the value
1048 $p$ corresponding the the probability that the agent's action is ignored and another action is chosen
1049 (uniformly at random) from the action space and played instead. For example, of $p = 0.25$ and the
1050 agent chooses the action *Right*, there is a $75\%$ chance that the agent goes right and a $25\%$ chance
1051 the agent goes a different direction. If $p = 0.0$ (deterministic environment) then achieving complete
1052 safety (zero-violations) becomes easier as the agent has complete control of the environment through
1053 their actions.

1054 We experiment with the following $p$ values: $p =$
1055 $0.1$ for property (1), $p = 0.1$ for property (2)
1056 and $p = 0.05$ for property (3). For these smaller
1057 $p$ values we would expect it to be easier for
1058 our methods including the baseline to achieve
1059 a higher-rate of safety and possibly complete
1060 safety in some cases.

1061 We see a similar situation as in the main paper,
1062 Q-learning (without penalties) simply finds the
1063 best policy ignoring costs. However, Q-learning
1064 (with penalties) is able to obtain the same perfor-
1065 mance now as our method Q-learning (Shield),
1066 both in terms of reward and cost. With a smaller
1067 $p$ value the safety-property can be satisfied with
1068 higher probability while still visiting the goal
1069 state frequently and obtaining high reward. In
1070 particular, these $p$ values are chosen such that
1071 each of the safety properties can be satisfies with
1072 probability at least $0.9$ from the goal state, thus
1073 penalizing safety-violations with $C = 10.0$ ap-
1074 pears to be enough to guarantee safety above
1075 $0.9$ at each timestep while still achieving high

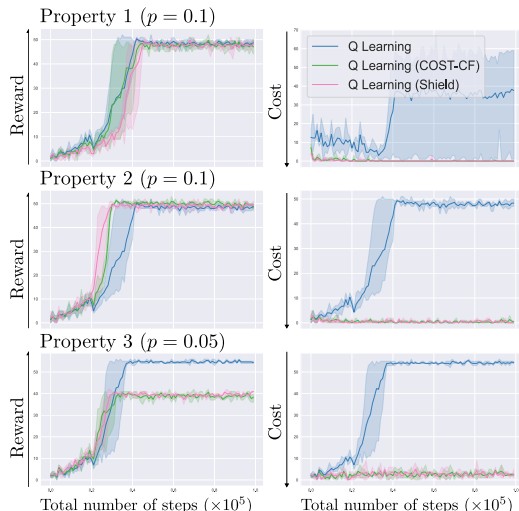

Figure 12: Episode reward and cost for Q-learning, Q-learning (COST-CF) and Q-learning (Shield) – all from the main paper. With smaller levels of stochasticity $p$

1076 reward. For different values of $C$ we might expect the baseline to have a different performance
1077 profile.

## G   Comparison to CMDP

1079 In this additional section we analyze the relationships between our problem setup and other common
1080 CMDP settings, for both the finite horizon and corresponding (discounted) infinite horizon problems.

### G.1   Finite Horizon

1082 For reference we restate Problem 4.1 here.

1083 **Problem 4.1 (restated)** (Step-wise bounded regular safety property constraint). *Let $P_{safe}$ be a regular*
1084 *safety property, $\mathcal{D}$ be the DFA such that $\mathcal{L}(\mathcal{D}) = BadPref(P_{safe})$ and $\mathcal{M}$ be the MDP;*

$$\max_\pi V_\pi \quad subject\ to \quad \Pr\left(\langle s_t, q_t \rangle \models \Diamond^{\leq H} accept\right) \leq p_1 \quad \forall t \in [0, T]$$

1085 *where all probability is taken under the product Markov Chain $\mathcal{M}_\pi \otimes \mathcal{D}$, $p_1 \in [0, 1]$ is a probability*
1086 *threshold, $H$ is the model checking horizon and $T$ is the fixed episode length.*

#### G.1.1   Expected Cumulative Constraint

1088 First we restate Problem 4.4.

**Problem 4.4 (restated)** (Expected cumulative constraint [4, 58]).

$$\max_\pi V_\pi \quad subject\ to \quad \mathbb{E}_{\langle s_t, q_t \rangle \sim \mathcal{M}_\pi \otimes \mathcal{D}}\left[\sum_{t=0}^T \mathcal{C}(\langle s_t, q_t \rangle)\right] \leq d_1$$

1089 *where $d_1 \in \mathbb{R}_+$ is the cost threshold and $T$ is the fixed episode length.*

**Proposition G.1.** *A feasible policy $\pi$ for Problem 4.1 with parameters $p_1 \in [0,1]$ is also a feasible policy for Problem 4.4 with parameter $d_1 \in \mathbb{R}_+$, provided that $d_1 \geq (T+1) \cdot p_1$.*

*Proof.* For $t \in [0,T]$ we define, the following random variables, $X_0, \ldots, X_T$, where

$$X_t = \mathcal{C}(\langle s_t, q_t \rangle) = 1\left[accept \in L'(\langle s_t, q_t \rangle)\right] \tag{66}$$

where,

$$\mathbb{E}[X_t] = \mathbb{E}\left[1\left[accept \in L'(\langle s_t, q_t \rangle)\right]\right] \tag{67}$$

$$= \Pr\left(accept \in L'(\langle s_t, q_t \rangle)\right) \tag{68}$$

$$\leq p_1 \tag{69}$$

The argument is straightforward if at every timestep $t \in [0,T]$ we have $\Pr(\langle s_t, q_t \rangle \models \Diamond^{\leq H} accept) \leq p_1$ then with probability $\leq p_1$ we have $accept \in L(\langle s_t, q_t \rangle)$. Then, under mild assumptions (i.e. $\mathcal{C}(\langle s_t, q_t \rangle) < \infty$) we consider the following decomposition of the expected cumulative cost,

$$\mathbb{E}_\pi\left[\sum_{t=0}^{T} \mathcal{C}(\langle s_t, q_t \rangle)\right] = \mathbb{E}_\pi\left[\sum_{t=0}^{T} X_t\right] \tag{70}$$

$$= \mathbb{E}_{s_0 \sim \mathcal{P}_0(\cdot)}[X_0] + \mathbb{E}_{s_1 \sim \mathcal{P}_1(\cdot)}[X_1] + \ldots + \mathbb{E}_{s_T \sim \mathcal{P}_T(\cdot)}[X_T] \tag{71}$$

$$= \mathbb{E}_\pi[X_0] + \mathbb{E}_\pi[X_1] + \ldots + \mathbb{E}_\pi[X_T] \tag{72}$$

We replace the subscript '$\langle s_t, q_t \rangle \sim \mathcal{M}_\pi \otimes \mathcal{D}$' here for brevity. Clearly by linearity of expectations this statement holds. Although it is worth noting that each expectation is taken under a different marginal state distribution (i.e. $\mathcal{P}_t(\cdot)$), which depends on $\pi$ (apart from the initial state distribution $\mathcal{P}_0(\cdot)$). From now on we will write this is implicitly (i.e. Eq. 72), rather than writing the marginal state distribution (at time $t$) for each expectation. Using our earlier observations we can now bound the expected cumulative cost from above as follows,

$$\mathbb{E}_\pi\left[\sum_{t=0}^{T} \mathcal{C}(\langle s_t, q_t \rangle)\right] = \mathbb{E}_\pi[X_0] + \mathbb{E}_\pi[X_1] + \ldots + \mathbb{E}_\pi[X_{T-1}] + \mathbb{E}_\pi[X_T] \tag{73}$$

$$\leq (T+1) \cdot p_1 \tag{74}$$

$\square$

**Proposition G.2.** *The converse is not strictly true, since there may be a feasible policy $\pi$ for Problem 4.4 with threshold $d_1 \leq (T+1) \cdot p_1$ which does not satisfy the constraints of Problem 4.1.*

*Proof.* We want to prove the following statement, a policy $\pi$ satisfying,

$$\mathbb{E}_\pi\left[\sum_{t=0}^{T} \mathcal{C}(\langle s_t, q_t \rangle)\right] \leq (T+1) \cdot p_1 \tag{75}$$

does not imply that,

$$\Pr\left(\langle s_t, q_t \rangle \models \Diamond^{\leq H} accept\right) \leq p_1 \quad \forall t \in [0,T] \tag{76}$$

To prove this we will show that there may be some policy $\pi$ that satisfies Eq. 75, but does not satisfy Eq. 76 at some timestep $t$. For simplicity we consider the first timestep (i.e. $t = 0$). First we assume $\pi$ is such that Eq. 75 holds, assuming $H \leq T$ then clearly we have,

$$\mathbb{E}_\pi\left[\sum_{t=0}^{H} \mathcal{C}(\langle s_t, q_t \rangle)\right] \leq \mathbb{E}_\pi\left[\sum_{t=0}^{T} \mathcal{C}(\langle s_t, q_t \rangle)\right] \leq (T+1) \cdot p_1 \tag{77}$$

Let $\Pr(\langle s_0, q_0 \rangle \models \Diamond^{\leq H} accept)$ denote the proportion of accepting paths from the initial state $s_0 \sim \mathcal{P}_0(\cdot)$ and automaton state $q_0 = \Delta(\mathcal{Q}_0, L(s_0))$. Suppose $\pi$ is such that $\Pr(\langle s_0, q_0 \rangle \models \Diamond^{\leq H} accept) > p_1$. We note that for each path $\rho \in \mathcal{S}^\omega$ and corresponding $trace(\rho) \in \Sigma^\omega$ such that $trace(\rho) \models \Diamond^{\leq H} accept$ the sum $\sum_{t=0}^{H} \mathcal{C}(\langle s_t, q_t \rangle) \geq 1$, and now we have,

$$(T+1) \cdot p_1 \geq \mathbb{E}_\pi\left[\sum_{t=0}^{T} \mathcal{C}(\langle s_t, q_t \rangle)\right] \geq \mathbb{E}_\pi\left[\sum_{t=0}^{H} \mathcal{C}(\langle s_t, q_t \rangle)\right] > p_1 \tag{78}$$

Now clearly for all $p_1 \in [0,1]$ and $T \in \mathbb{Z}_+$ the following holds,

$$p_1 < (T+1) \cdot p_1 \tag{79}$$

This implies that there may exist some $\pi$ satisfying Eq. 75 and such that $\Pr(\langle s_0, q_0 \rangle \models \Diamond^{\leq H} accept) > p_1$, i.e. does not satisfy Eq. 76 at timestep $t = 0$. $\qquad\square$

**Proposition G.3.** *A feasible policy $\pi$ for Problem 4.4 with threshold $d_1 \leq p_1$, satisfies $\Pr(\langle s_t, q_t \rangle \models \Diamond^{\leq H} accept) \leq p_1$ for all $t \in [0,T]$. This bound is tight.*

*Proof.* Firstly, a feasible policy $\pi$ for Problem 4.4 with threshold $d_1 \leq p_1$ clearly satisfies,

$$\mathbb{E}_\pi \left[ \sum_{t=0}^{T} \mathcal{C}(\langle s_t, q_t \rangle) \right] \leq p_1 \tag{80}$$

Assuming $H \leq T$, then this implies that for all $t' \in [0, T-H]$ we have,

$$\mathbb{E}_\pi \left[ \sum_{t=t'}^{t'+H} \mathcal{C}(\langle s_t, q_t \rangle) \right] \leq \mathbb{E}_\pi \left[ \sum_{t=0}^{T} \mathcal{C}(\langle s_t, q_t \rangle) \right] \leq p_1 \tag{81}$$

Let $\Pr(\langle s_{t'}, q_{t'} \rangle \models \Diamond^{\leq H} accept)$ denote the proportion of accepting paths at timestep $t'$, where $s_{t'} \sim \mathcal{P}_{t'}(\cdot)$. Here $\mathcal{P}_{t'}(\cdot)$ denotes the marginal state distribution at time $t'$. Recall that for each path $\rho \in \mathcal{S}^\omega$ and corresponding $trace(\rho) \in \Sigma^\omega$ such that $trace(\rho) \models \Diamond^{\leq H} accept$ the sum $\sum_{t=t'}^{t'+H} \mathcal{C}(\langle s_t, q_t \rangle) \geq 1$. Without loss of generality fix some $t' \in [0, T-H]$ and suppose that $\Pr(\langle s_{t'}, q_{t'} \rangle \models \Diamond^{\leq H} accept) > p_1$. This implies that,

$$\mathbb{E}_\pi \left[ \sum_{t=0}^{T} \mathcal{C}(\langle s_t, q_t \rangle) \right] \geq \mathbb{E}_\pi \left[ \sum_{t=t'}^{t'+H} \mathcal{C}(\langle s_t, q_t \rangle) \right] > p_1 \tag{82}$$

Which is a contradiction. Therefore, it must be the case that when Eq. 80 is satisfied then so is $\Pr(\langle s_t, q_t \rangle \models \Diamond^{\leq H} accept]) \leq p_1$ for all $t \in [0, T-H]$. For the remaining $t' \in [T-H, T]$ a similar argument can be made, the only detail is to ensure the sum in Eq. 81 is up to $T$ rather than $t' + H$. To prove that this bound is tight we can again show the possible existence of a counter example. In particular, we want to prove the following statement, a policy $\pi$ satisfying,

$$\mathbb{E}_\pi \left[ \sum_{t=0}^{T} \mathcal{C}(\langle s_t, q_t \rangle) \right] \leq p_1 + c \tag{83}$$

for some constant $c > 0$, does not imply that,

$$\Pr\left( \langle s_t, q_t \rangle \models \Diamond^{\leq H} accept \right) \leq p_1 \quad \forall t \in [0, T] \tag{84}$$

We will show that there may exist some policy $\pi$ that satisfies Eq. 83 but does not satisfy Eq. 84 at some timestep $t$. Firstly, we assume $\pi$ is such that Eq. 83 holds, this implies that for all $t' \in [0, T-H]$ we have,

$$\mathbb{E}_\pi \left[ \sum_{t=t'}^{t'+H} \mathcal{C}(\langle s_t, q_t \rangle) \right] \leq \mathbb{E}_\pi \left[ \sum_{t=0}^{T} \mathcal{C}(\langle s_t, q_t \rangle) \right] \leq p_1 + c \tag{85}$$

Fix some $t' \in [0, T-H]$ and once again let $\Pr(\langle s_{t'}, q_{t'} \rangle \models \Diamond^{\leq H} accept)$ denote the proportion of accepting paths at timestep $t'$. Suppose $\pi$ is such that $\Pr(\langle s_{t'}, q_{t'} \rangle \models \Diamond^{\leq H} accept) > p_1$. Again recall that for each path $\rho \in \mathcal{S}^\omega$ and corresponding trace $trace(\rho) \in \Sigma^\omega$ such that $trace(\rho) \models \Diamond^{\leq H} accept$ the sum $\sum_{t=t'}^{t'+H} \mathcal{C}(\langle s_t, q_t \rangle) \geq 1$, and so,

$$p_1 + c \geq \mathbb{E}_\pi \left[ \sum_{t=0}^{T} \mathcal{C}(\langle s_t, q_t \rangle) \right] \geq \mathbb{E}_\pi \left[ \sum_{t=t'}^{t'+H} \mathcal{C}(\langle s_t, q_t \rangle) \right] > p_1 \tag{86}$$

Now clearly for all $p_1 \in [0,1]$ and $c > 0$, the following holds,

$$p_1 < p_1 + c \tag{87}$$

This implies that there may exist some $\pi$ satisfying Eq. 83 and such that $\Pr(\langle s_{t'}, q_{t'} \rangle \models \Diamond^{\leq H} accept) > p_1$, i.e. does not satisfy Eq. 84 at timestep $t = t'$. $\qquad\square$

 **G.1.2   Probabilistic Cumulative Constraint**

 First we restate Problem 4.5.

**Problem 4.5 (restated)** (Probabilistic cumulative constraint [18, 56])**.**

$$\max_{\pi} V_{\pi} \quad \textit{subject to} \quad \mathbb{P}_{\langle s_t, q_t \rangle \sim \mathcal{M}_{\pi} \otimes \mathcal{D}} \left[ \sum_{t=0}^{T} \mathcal{C}(\langle s_t, q_t \rangle) \leq d_2 \right] \geq 1 - \delta_2$$

*where $d_2 \in \mathbb{R}_+$ is the cost threshold, $\delta_2$ is a tolerance parameter and $T$ is the fixed episode length.*

**Proposition G.4.** *A feasible policy $\pi$ for Problem 4.1 with parameters $p_1 \in [0, 1]$ is also a feasible policy for Problem 4.5 with parameters $d_2 \in \mathbb{R}_+$ and $\delta_2 \in (0, 1]$, provided that, $d_2 \geq \sqrt{(T+1)/2 \cdot \log(1/\delta_2)} + (T+1) \cdot p_1$.*

*Proof.* For $t \in [0, T]$ we define the following random variables, $X_0, \ldots, X_T$, where,

$$X_t = \mathcal{C}(\langle s_t, q_t \rangle) = 1 \left[ accept \in L'(\langle s_t, q_t \rangle) \right] \tag{88}$$

and we make the same following observation,

$$\mathbb{E}\left[X_t\right] = \mathbb{E}\left[ 1 \left[ accept \in L'(\langle s_t, q_t \rangle) \right] \right] \tag{89}$$

$$= \Pr\left( accept \in L'(\langle s_t, q_t \rangle) \right) \tag{90}$$

$$\leq p_1 \cdot \delta \tag{91}$$

See the proof of Prop. G.1 for details, the argument is identical. Once again, under mild assumptions (i.e. $\mathcal{C}(\langle s_t, q_t \rangle) < \infty$) we consider the following decomposition of the expected cumulative cost,

$$\mathbb{E}_{\pi}\left[ \sum_{t=0}^{T} \mathcal{C}(\langle s_t, q_t \rangle) \right] = \mathbb{E}_{\pi}\left[X_0\right] + \mathbb{E}_{\pi}\left[X_1\right] + \ldots + \mathbb{E}_{\pi}\left[X_T\right] \tag{92}$$

$$\leq (T+1) \cdot p_1 \tag{93}$$

Again we replace the subscript '$\langle s_t, q_t \rangle \sim \mathcal{M}_{\pi} \otimes \mathcal{D}$' here for brevity, see the proof of Prop. G.1 for the full details. Before we proceed we must first deal with the dependence between the random variables $X_0, \ldots, X_T$. Strictly speaking it is not the case that $\Pr(X_t = 1 \mid X_{t-1}, \ldots, X_0) = \Pr(X_t = 1)$. However, we have already established that $\Pr(X_t = 1) \leq p_1$, as such we can simulate $X_0, \ldots, X_T$ as a sequence of independent coin flips $Y_0, \ldots, Y_T$ with probability $p_1$, it is then the case that $\mathbb{P}[\sum_{t=0}^{T} X_t > d_2] \leq \mathbb{P}[\sum_{t=0}^{T} Y_t > d_2]$. We can now continue by bounding the probability we care about,

$$1 - \mathbb{P}\left[ \sum_{t=0}^{T} \mathcal{C}(\langle s_t, q_t \rangle) \leq d_2 \right] = \mathbb{P}\left[ \sum_{t=0}^{T} \mathcal{C}(\langle s_t, q_t \rangle) > d_2 \right] \tag{94}$$

$$= \mathbb{P}\left[ \sum_{t=0}^{T} X_t > d_2 \right] \tag{95}$$

$$\leq \mathbb{P}\left[ \sum_{t=0}^{T} Y_t > d_2 \right] \tag{96}$$

$$= \mathbb{P}\left[ \sum_{t=0}^{T} Y_t > (T+1) \cdot p_1 + d_2 - (T+1) \cdot p_1 \right] \tag{97}$$

$$= \mathbb{P}\left[ \sum_{t=0}^{T} Y_t > \mathbb{E}\left[ \sum_{t=0}^{T} Y_t \right] + d_2 - (T+1) \cdot p_1 \right] \tag{98}$$

$$\leq \exp\left( -\frac{2 \cdot (d_2 - (T+1) \cdot p_1)^2}{\sum_{t=0}^{T} (\max\{Y_i\} - \min\{Y_i\})^2} \right) \tag{99}$$

$$= \exp\left( -\frac{2 \cdot (d_2 - (T+1) \cdot p_1)^2}{(T+1)} \right) \tag{100}$$

The first inequality (Eq. 96) comes from our earlier construction and the second (Eq. 99) is obtained from Hoeffding's inequality [40] for bounded random variables. Finally, bounding the final expression from above by $\delta_2$ and rearranging gives the desired result. $\qquad \square$

**Proposition G.5.** *A feasible policy $\pi$ for Problem 4.5 with parameters $\delta_2 \leq p_1$ and $d_2 < 1$, satisfies* $\Pr(\langle s_t, q_t \rangle \models \Diamond^{\leq H} accept) \leq p_1$ *for all $t \in [0, T]$. This bound is tight.*

*Proof.* A feasible policy $\pi$ for Problem 4.5 with parameters $\delta_2 \leq p_1$ and $d_2 < 1$ clearly implies that,

$$\mathbb{P}\left[\sum_{t=0}^{T} \mathcal{C}(\langle s_t, q_t \rangle) < 1\right] \geq 1 - p_1 \tag{101}$$

Assuming $H \leq T$, then this implies that for all $t' \in [0, T - H]$ we have,

$$\mathbb{P}\left[\sum_{t=t'}^{t'+H} \mathcal{C}(\langle s_t, q_t \rangle) < 1\right] \geq \mathbb{P}\left[\sum_{t=0}^{T} \mathcal{C}(\langle s_t, q_t \rangle) < 1\right] \geq 1 - p_1 \tag{102}$$

Let $\Pr(\langle s_{t'}, q_{t'} \rangle \models \Diamond^{\leq H} accept)$ denote the proportion of accepting paths at timestep $t'$, where $s_{t'} \sim \mathcal{P}_{t'}(\cdot)$. Again $\mathcal{P}_{t'}(\cdot)$ denotes the marginal state distribution at time $t'$. Recall that for each path $\rho \in \mathcal{S}^\omega$ and corresponding $trace(\rho) \in \Sigma^\omega$ such that $trace(\rho) \models \Diamond^{\leq H} accept$ the sum $\sum_{t=t'}^{t'+H} \mathcal{C}(\langle s_t, q_t \rangle) \geq 1$. Without loss of generality fix some $t' \in [0, T - H]$ and suppose that $\Pr(\langle s_{t'}, q_{t'} \rangle \models \Diamond^{\leq H} accept) > p_1$. This implies that,

$$\mathbb{P}\left[\sum_{t=0}^{T} \mathcal{C}(\langle s_t, q_t \rangle) \geq 1\right] \geq \mathbb{P}\left[\sum_{t=t'}^{t'+H} \mathcal{C}(\langle s_t, q_t \rangle) \geq 1\right] > p_1 \tag{103}$$

Which is a contradiction. Therefore, it must be the case that when Eq. 101 is satisfied then so is $\Pr(\langle s_t, q_t \rangle \models \Diamond^{\leq H} accept]) \leq p_1$ for all $t \in [0, T - H]$. For the remaining $t' \in [T - H, T]$ a similar argument can be made, the only detail is to ensure the sum in Eq. 102 is up to $T$ rather than $t' + H$. To prove that this bound is tight we can show the possible existence of a counter example. In particular, we want to prove the following statement, a policy $\pi$ satisfying,

$$\mathbb{P}\left[\sum_{t=0}^{T} \mathcal{C}(\langle s_t, q_t \rangle) < 1\right] \geq 1 - (p_1 + c) \tag{104}$$

for some constant $c > 0$ does not imply that,

$$\Pr\left(\langle s_t, q_t \rangle \models \Diamond^{\leq H} accept\right) \leq p_1 \quad \forall t \in [0, T] \tag{105}$$

We will show that there may exist some policy $\pi$ that satisfies Eq. 104 but does not satisfy Eq. 105 at some timestep $t$. Firstly, we assume $\pi$ is such that Eq. 104 holds, this implies that for all $t' \in [0, T - H]$ we have,

$$\mathbb{P}\left[\sum_{t=t'}^{t'+H} \mathcal{C}(\langle s_t, q_t \rangle) < 1\right] \geq \mathbb{P}\left[\sum_{t=0}^{T} \mathcal{C}(\langle s_t, q_t \rangle) < 1\right] \geq 1 - (p_1 + c) \tag{106}$$

Fix some $t' \in [0, T - H]$ and let $\Pr(\langle s_{t'}, q_{t'} \rangle \models \Diamond^{\leq H} accept)$ denote the proportion of accepting paths at timestep $t'$. Suppose that $\pi$ is such that $\Pr(\langle s_{t'}, q_{t'} \rangle \models \Diamond^{\leq H} accept) > p_1$. Again recall that for each path $\rho \in \mathcal{S}^\omega$ and corresponding $trace(\rho) \in \Sigma^\omega$ such that $trace(\rho) \models \Diamond^{\leq H} accept$ the sum $\sum_{t=t'}^{t'+H} \mathcal{C}(\langle s_t, q_t \rangle) \geq 1$, and so,

$$p_1 + c \geq \mathbb{P}\left[\sum_{t=0}^{T} \mathcal{C}(\langle s_t, q_t \rangle) \geq 1\right] \geq \mathbb{P}\left[\sum_{t=t'}^{t'+H} \mathcal{C}(\langle s_t, q_t \rangle) \geq 1\right] > p_1 \tag{107}$$

Now clearly for all $p_1 \in [0, 1]$ and $c > 0$, the following holds,

$$p_1 < p_1 + c \tag{108}$$

This implies that there may exist some $\pi$ satisfying Eq. 104 and such that $\Pr(\langle s_{t'}, q_{t'} \rangle \models \Diamond^{\leq H} accept) > p_1$, i.e. does not satisfy Eq. 105 at timestep $t = t'$. $\qquad\square$

 ### G.1.3   Instantaneous constraint

 First we restate Problem 4.6.

**Problem 4.6 (restated)** (Instantaneous constraint [23, 60, 69]).

$$\max_{\pi} V_{\pi} \quad subject\ to \quad \mathbb{P}_{\langle s_t, q_t \rangle \sim \mathcal{M}_{\pi} \otimes \mathcal{D}}\big[ \mathcal{C}(\langle s_t, q_t \rangle) \leq d_3 \big] = 1 \quad \forall t \in [0, T]$$

 **Proposition G.6.** *A feasible policy $\pi$ for Problem 4.6 with threshold $d_3 < 1$ (otherwise the problem*  *is trivial) is a feasible policy for Problem 4.1 if and only if $p_1 = 0$.*

 *Proof.* We start by proving the $4.6 \Rightarrow 4.1$ direction. A feasible policy $\pi$ for Problem 4.6 with $d_3 < 1$  satisfies,

$$\Pr\left( \mathcal{C}(\langle s_t, q_t \rangle) < 1 \right) = 1 \quad \forall t \in [0, T] \tag{109}$$

 which implies that,

$$\Pr\left( \mathcal{C}(\langle s_t, q_t \rangle) = 0 \right) = 1 \quad \forall t \in [0, T] \tag{110}$$

 and by Defn. 4.3,

$$\Pr\left( accept \notin L'(\langle s_t, q_t \rangle) \right) = 1 \quad \forall t \in [0, T] \tag{111}$$

 Then if for all $t \in [0, T]$, $accept \notin L'(\langle s_t, q_t \rangle)$ then we have $\Pr(\langle s_0, q_0 \rangle \not\models \Diamond accept) = 1$, where  $q_0 = \Delta(\mathcal{Q}_0, L(s_0))$ and by extension we have $\Pr(\langle s_t, q_t \rangle \not\models \Diamond accept^{\leq H}) = 1$ for all $t \in [0, T]$.  This completes the proof of this direction.

 Now we prove the $4.1 \Rightarrow 4.6$ direction. A policy $\pi$ satisfying $\Pr(\langle s_t, q_t \rangle \models \Diamond accept^{\leq H})) = 0$ for all  $t \in [0, T]$ implies that $\Pr(\langle s_t, q_t \rangle \not\models \Diamond accept^{\leq H}) = 1$ for all $t \in [0, T]$ which implies the following,

$$\Pr\left( accept \notin L'(\langle s_t, q_t \rangle) \right) = 1 \quad \forall t \in [0, T] \tag{112}$$

 and by Defn. 4.3,

$$\Pr\left[ \mathcal{C}(\langle s_t, q_t \rangle) = 0 \right] = 1 \quad \forall t \in [0, T] \tag{113}$$

 which implies that,

$$\Pr\left[ \mathcal{C}(\langle s_t, q_t \rangle) < 1 \right] = 1 \quad \forall t \in [0, T] \tag{114}$$

 which concludes the proof. $\qquad\square$

 ## G.2   Infinite Horizon

 While in this paper we only consider finite horizon problems with a fixed episode length $T$, we note  that we can also make a set of similar statements for the infinite horizon (discounted) setting. In this  section we provide the corresponding statements and proofs for the infinite horizon setting. Firstly,  we consider the following infinite horizon problem.

 **Problem G.7** (Step-wise bounded regular safety property constraint). *Let $P_{safe}$ be a regular safety*  *property, $\mathcal{D}$ be the DFA such that $\mathcal{L}(\mathcal{D}) = BadPref(P_{safe})$ and $\mathcal{M}$ be the MDP;*

$$\max_{\pi} V_{\pi} \quad subject\ to \quad \Pr\left( \langle s_t, q_t \rangle \models \Diamond^{\leq H} accept \right) \leq p_1 \quad \forall t = 0, 1, 2, \ldots$$

 *where all probability is taken under the product Markov chain $\mathcal{M}_{\pi} \otimes \mathcal{D}$, $p_1 \in [0, 1]$ is a probability*  *threshold $H$ is the model checking horizon .*

 ### G.2.1   Expected Cumulative Constraint

**Problem G.8** (Expected cumulative constraint).

$$\max_{\pi} V_{\pi} \quad subject\ to \quad \mathbb{E}_{\langle s_t, q_t \rangle \sim \mathcal{M}_{\pi} \otimes \mathcal{D}}\left[ \sum_{t=0}^{\infty} \gamma^t \mathcal{C}(\langle s_t, q_t \rangle) \right] \leq d_1$$

 *where $d_1 \in \mathbb{R}_+$ is the cost threshold and $\gamma \in [0, 1)$ is the discount factor.*

 **Proposition G.9.** *A feasible policy $\pi$ for Problem G.7 with parameters $p_1 \in [0, 1]$, is also a feasible*  *policy for Problem G.8 with parameter $d_1 \in \mathbb{R}_+$, provided that $d_1 \geq T \cdot p_1$, where $T = 1/(1 - \gamma)$ is*  *the effective horizon.*

*Proof.* For $t = 0, 1, 2, \ldots$ we define, the following random variables, $X_0, X_1, X_2, \ldots$, where,

$$X_t = \mathcal{C}(\langle s_t, q_t \rangle) = 1\left[accept \in L'(\langle s_t, q_t \rangle)\right] \tag{115}$$

where,

$$\mathbb{E}\left[X_t\right] = \mathbb{E}\left[1\left[accept \in L'(\langle s_t, q_t \rangle)\right]\right] \tag{116}$$

$$= \Pr\left(accept \in L'(\langle s_t, q_t \rangle)\right) \tag{117}$$

$$\leq p_1 \tag{118}$$

The argument for this is straightforward. If at every timestep $t = 0, 1, 2, \ldots$ we have $\Pr(\langle s_t, q_t \rangle \models \Diamond^{\leq H} accept) \leq p_1$ then with probability $\leq p_1$ we have $accept \in L(\langle s_t, q_t \rangle)$. Let $T = 1/(1 - \gamma)$ be the effective horizon, then under mild assumptions (i.e. $\mathcal{C}(\langle s_t, q_t \rangle) < \infty$) we can consider the following decomposition of the expected cumulative cost,

$$\mathbb{E}_\pi\left[\sum_{t=0}^{\infty} \gamma^t \mathcal{C}(\langle s_t, q_t \rangle)\right] = \mathbb{E}_\pi\left[\sum_{t=0}^{\infty} \gamma^t X_t\right] \tag{119}$$

$$= \mathbb{E}_{s_0 \sim \mathcal{P}_0(\cdot)}\left[X_0\right] + \gamma \cdot \mathbb{E}_{s_1 \sim \mathcal{P}_1(\cdot)}\left[X_1\right] + \ldots$$
$$+ \gamma^T \cdot \mathbb{E}_{s_T \sim \mathcal{P}_T(\cdot)}\left[X_T\right] + \ldots \tag{120}$$

$$= \mathbb{E}_\pi\left[X_0\right] + \gamma \cdot \mathbb{E}_\pi\left[X_1\right] + \ldots + \gamma^T \cdot \mathbb{E}_\pi\left[X_T\right] + \ldots \tag{121}$$

We replace the subscript '$\langle s_t, q_t \rangle \sim \mathcal{M}_\pi \otimes \mathcal{D}$' here for brevity. Clearly by linearty of expectations this statement holds. Although it is worth noting that each expectation is taken under a different marginal state distribution (i.e. $\mathcal{P}_t(\cdot)$), which depends on $\pi$ (apart from the initial state distribution $\mathcal{P}_0(\cdot)$). From now on we will write this is implicitly (i.e. Eq. 121), rather than writing the marginal state distribution (at time $t$) for each expectation. Using our earlier observations we can now bound the expected cumulative cost from above as follows,

$$\mathbb{E}_\pi\left[\sum_{t=0}^{\infty} \gamma^t \mathcal{C}(\langle s_t, q_t \rangle)\right] = \mathbb{E}_\pi\left[X_0\right] + \gamma \cdot \mathbb{E}_\pi\left[X_1\right] + \ldots + \gamma^T \cdot \mathbb{E}_\pi\left[X_T\right] + \ldots \tag{122}$$

$$\leq p_1 + \gamma \cdot p_1 + \ldots \qquad + \gamma^{T-1} \cdot p_1 + \gamma^T \cdot p_1 + \ldots \tag{123}$$

$$= p_1 \cdot \sum_{t=0}^{\infty} \gamma^t = p_1 \cdot (1/(1-\gamma)) = T \cdot p_1 \tag{124}$$

$\square$

**Proposition G.10.** *The converse is not strictly true, since there may be a feasible policy $\pi$ for Problem G.8 with threshold $d_1 \leq T \cdot p_1$ which does not satisfy the constraints of Problem G.7*

We want to prove the following statement, a policy $\pi$ satisfying,

$$\mathbb{E}_\pi\left[\sum_{t=0}^{\infty} \gamma^t \mathcal{C}(\langle s_t, q_t \rangle)\right] \leq T \cdot p_1 \tag{125}$$

does not imply that,

$$\Pr\left(\langle s_t, q_t \rangle \models \Diamond^{\leq H} accept\right) \leq p_1 \quad \forall t = 0, 1, 2, \ldots \tag{126}$$

*Proof.* To prove this we will show that there may be some policy $\pi$ that satisfies Eq. 125, but does not satisfy Eq. 126 at some timestep $t$. For simplicity we consider the first timestep (i.e. $t = 0$). First we assume $\pi$ is such that Eq. 125 holds, then clearly we have,

$$\mathbb{E}_\pi\left[\sum_{t=0}^{H} \gamma^t \mathcal{C}(\langle s_t, q_t \rangle)\right] \leq \mathbb{E}_\pi\left[\sum_{t=0}^{\infty} \gamma^t \mathcal{C}(\langle s_t, q_t \rangle)\right] \leq T \cdot p_1 \tag{127}$$

Let $\Pr(\langle s_0, q_0 \rangle \models \Diamond^{\leq H} accept)$ denote the proportion of accepting paths from the initial state $s_0 \sim \mathcal{P}_0(\cdot)$. Suppose $\pi$ is such that $\Pr(\langle s_0, q_0 \rangle \models \Diamond^{\leq H} accept) > p_1$. We note that for each path $\rho \in \mathcal{S}^\omega$

and corresponding $trace(\rho) \in \Sigma^\omega$ such that $trace(\rho) \models \Diamond^{\leq H} accept$ the sum $\sum_{t=0}^{H} \gamma^t \mathcal{C}(\langle s_t, q_t \rangle) \geq \gamma^H$, and so,

$$T \cdot p_1 \geq \mathbb{E}_\pi \left[ \sum_{t=0}^{\infty} \gamma^t \mathcal{C}(\langle s_t, q_t \rangle) \right] \geq \mathbb{E}_\pi \left[ \sum_{t=0}^{H} \gamma^t \mathcal{C}(\langle s_t, q_t \rangle) \right] > p_1 \cdot \gamma^H \tag{128}$$

Now clearly for all $p_1 \in [0,1]$, $\gamma \in [0,1)$, $H \in \mathbb{Z}_+$ and $T = 1/(1-\gamma)$ the following holds,

$$p_1 \cdot \gamma^H < T \cdot p_1 \tag{129}$$

This implies that there may exist some $\pi$ satisfying Eq. 125 and such that $\Pr(\langle s_0, q_0 \rangle \models \Diamond^{\leq H} accept) > p_1$, i.e. does not satisfy Eq. 126 at timestep $t = 0$. $\qquad\square$

**Proposition G.11.** *A feasible policy $\pi$ for Problem 4.4 with threshold $d_1 \leq p_1 \cdot \gamma^{T+H}$ satisfies $\Pr(\langle s_t, q_t \rangle \models \Diamond^{\leq H} accept) \leq p_1$ up to the effective horizon $T = 1/(1-\gamma)$. This bound is tight.*

*Proof.* Let $T = 1/(1-\gamma)$ be the effective horizon. A feasible policy $\pi$ for Problem 4.4 with threshold $d_1 \leq p_1 \cdot \gamma^{T+H}$ clearly satisfies,

$$\mathbb{E}_\pi \left[ \sum_{t=0}^{\infty} \gamma^t \mathcal{C}(\langle s_t, q_t \rangle) \right] \leq p_1 \cdot \gamma^{T+H} \tag{130}$$

which implies that for all $t' \in [0, T]$ we have,

$$p_1 \cdot \gamma^{T+H} \geq \mathbb{E}_\pi \left[ \sum_{t=0}^{\infty} \gamma^t \mathcal{C}(\langle s_t, q_t \rangle) \right] \geq \mathbb{E}_\pi \left[ \sum_{t=t'}^{t'+H} \gamma^t \mathcal{C}(\langle s_t, q_t \rangle) \right] \tag{131}$$

$$= \mathbb{E}_\pi \left[ \gamma^{t'} \sum_{t=t'}^{t'+H} \gamma^{t-t'} \mathcal{C}(\langle s_t, q_t \rangle) \right] \tag{132}$$

$$= \gamma^{t'} \cdot \mathbb{E}_\pi \left[ \sum_{t=t'}^{t'+H} \gamma^{t-t'} \mathcal{C}(\langle s_t, q_t \rangle) \right] \tag{133}$$

Let $\Pr(\langle s_{t'}, q_{t'} \rangle \models \Diamond^{\leq H} accept)$ denote the proportion of accepting paths at timestep $t'$, where $s_{t'} \sim \mathcal{P}_{t'}(\cdot)$. Here $\mathcal{P}_{t'}(\cdot)$ denotes the marginal state distribution at time $t'$. Recall that for each path $\rho \in \mathcal{S}^\omega$ and corresponding $trace(\rho) \in \Sigma^\omega$ such that $trace(\rho) \models \Diamond^{\leq H} accept$ the sum $\sum_{t=t'}^{t'+H} \gamma^{t-t'} \mathcal{C}(\langle s_t, q_t \rangle) \geq \gamma^H$. Without loss of generality fix some $t' \in [0, T]$ and suppose that $\Pr(\langle s_{t'}, q_{t'} \rangle \models \Diamond^{\leq H} accept) > p_1$. This implies that,

$$\mathbb{E}_\pi \left[ \sum_{t=0}^{\infty} \gamma^t \mathcal{C}(\langle s_t, q_t \rangle) \right] \geq \gamma^{t'} \cdot \mathbb{E}_\pi \left[ \sum_{t=t'}^{t'+H} \gamma^{t-t'} \mathcal{C}(\langle s_t, q_t \rangle) \right] \tag{134}$$

$$> p_1 \cdot \gamma^H \cdot \gamma^{t'} \geq p_1 \cdot \gamma^{T+H} \tag{135}$$

Which is a contradiction. Therefore, it must be the case that when Eq. 130 is satisfied then so is $\Pr(\langle s_t, q_t \rangle \models \Diamond^{\leq H} accept]) \leq p_1$ for all $t \in [0, T]$. To prove that this bound is tight we can again show the possible existence of a counter example. In particular, we want to prove the following statement, a policy $\pi$ satisfying,

$$\mathbb{E}_\pi \left[ \sum_{t=0}^{\infty} \gamma^t \mathcal{C}(\langle s_t, q_t \rangle) \right] \leq p_1 \cdot \gamma^{T+H} + c \tag{136}$$

for some constant $c > 0$, does not imply that,

$$\Pr\left( \langle s_t, q_t \rangle \models \Diamond^{\leq H} accept \right) \leq p_1 \quad \forall t \in [0, T] \tag{137}$$

We will show that there may exist some policy $\pi$ that satisfies Eq. 136 but does not satisfy Eq. 137 at some timestep $t$. For simplicity we consider timestep $t = T$, although we note that with a little extra

work we could come up with a proof for any $t \in [0, T]$. Firstly, we assume $\pi$ is such that Eq. 136 holds, then we have,

$$p_1 \cdot \gamma^{T+H} + c \geq \mathbb{E}_\pi \left[ \sum_{t=0}^{\infty} \gamma^t \mathcal{C}(\langle s_t, q_t \rangle) \right] \geq \mathbb{E}_\pi \left[ \sum_{t=T}^{T+H} \gamma^t \mathcal{C}(\langle s_t, q_t \rangle) \right] \quad (138)$$

Let $\Pr(\langle s_T, q_T \rangle \models \Diamond^{\leq H} accept)$ denote the proportion of accepting paths at timestep $T$. Suppose $\pi$ is such that $\Pr(\langle s_T, q_T \rangle \models \Diamond^{\leq H} accept) > p_1$. We note that for each path $\rho \in \mathcal{S}^\omega$ and corresponding $trace(\rho) \in \Sigma^\omega$ such that $trace(\rho) \models \Diamond^{\leq H} accept$ the sum $\sum_{t=T}^{T+H} \gamma^t \mathcal{C}(\langle s_t, q_t \rangle) \geq \gamma^{T+H}$, and so,

$$p_1 \cdot \gamma^{T+H} + c \geq \mathbb{E}_\pi \left[ \sum_{t=0}^{\infty} \gamma^t \mathcal{C}(\langle s_t, q_t \rangle) \right] \quad (139)$$

$$\geq \mathbb{E}_\pi \left[ \sum_{t=T}^{T+H} \gamma^t \mathcal{C}(\langle s_t, q_t \rangle) \right] \quad (140)$$

$$> p_1 \cdot \gamma^{T+H} \quad (141)$$

Now clearly for all $p_1 \in [0, 1]$, $\gamma \in [0, 1)$, $c > 0$, $H \in \mathbb{Z}_+$ and $T = 1/(1 - \gamma)$, the following holds,

$$p_1 \cdot \gamma^{T+H} < p_1 \cdot \gamma^{T+H} + c \quad (142)$$

This implies that there may exist some $\pi$ satisfying Eq. 136 and such that $\Pr(\langle s_T, q_T \rangle \models \Diamond^{\leq H} accept) > p_1$, i.e. does not satisfy Eq. 137 at timestep $t = T$. $\qquad \square$

### G.3 Probabilistic Cumulative Constraint

**Problem G.12** (Probabilistic cumulative constraint).

$$\max_\pi V_\pi \quad subject\ to \quad \mathbb{P}_{\langle s_t, q_t \rangle \sim \mathcal{M}_\pi \otimes \mathcal{D}} \left[ \sum_{t=0}^{\infty} \gamma^t \mathcal{C}(\langle s_t, q_t \rangle) \leq d_2 \right] \geq 1 - \delta_2$$

where $d_2 \in \mathbb{R}_+$ is the cost threshold, $\delta_2$ is a tolerance parameter and $\gamma \in [0, 1)$ is the discount factor.

**Proposition G.13.** *A feasible policy $\pi$ for Problem G.7 with parameters $p_1 \in [0, 1]$, is also a feasible policy for Problem G.12 with parameters $d_2 \in \mathbb{R}_+$ and $\delta_2 \in (0, 1]$, provided that, $d_2 \geq \sqrt{(\lceil \log(T) \rceil \cdot T)/2 \cdot \log(1/\delta_2)} + \lceil \log(T) \rceil \cdot T \cdot p_1 + 1$, where $T = 1/(1 - \gamma)$ is the effective horizon.*

*Proof.* Again $t = 0, 1, 2, \ldots$ we define the following random variables, $X_0, X_1, X_2, \ldots$, where,

$$X_t = \mathcal{C}(\langle s_t, q_t \rangle) = \mathbb{1}\left[ accept \in L'(\langle s_t, q_t \rangle) \right] \quad (143)$$

and we make the following observation,

$$\mathbb{E}[X_t] = \mathbb{E}\left[ \mathbb{1}\left[ accept \in L'(\langle s_t, q_t \rangle) \right] \right] \quad (144)$$

$$= \Pr\left( accept \in L'(\langle s_t, q_t \rangle) \right) \quad (145)$$

$$\leq p_1 \quad (146)$$

See the proof of Prop. G.9, the argument is identical. Under mild assumptions (i.e. $\mathcal{C}(\langle s_t, q_t \rangle) < \infty$) we consider the following decomposition of the (undiscounted) expected cumulative cost up to timestep $\lceil \log(T) \rceil \cdot T - 1$,

$$\mathbb{E}_\pi \left[ \sum_{t=0}^{\lceil \log(T) \rceil \cdot T - 1} \mathcal{C}(\langle s_t, q_t \rangle) \right] = \mathbb{E}_\pi [X_0] + \mathbb{E}_\pi [X_1] + \ldots + \mathbb{E}_\pi \left[ X_{\lceil \log(T) \rceil \cdot T - 1} \right] \quad (147)$$

$$\leq \lceil \log(T) \rceil \cdot T \cdot p_1 \quad (148)$$

Again we replace the subscript '$\langle s_t, q_t \rangle \sim \mathcal{M}_\pi \otimes \mathcal{D}$' here for brevity, see the proof of Prop. G.9 for more details. Before we proceed we must first deal with the dependence between the random variables $X_0, \ldots X_{\lceil \log(T) \rceil \cdot T - 1}$. Strictly speaking it is not the case that $\Pr(X_t = 1 \mid X_{t-1}, \ldots, X_0) =$

$\Pr(X_t = 1)$. However, we have already established that $\Pr(X_t = 1) \leq p_1$, as such we can simulate $X_0, \ldots, X_{\lceil \log(T) \rceil \cdot T - 1}$ as a sequence of independent coin flips $Y_0, \ldots, Y_{\lceil \log(T) \rceil \cdot T - 1}$ with probability $p_1$, it is then the case that $\mathbb{P}[\sum_{t=0}^{\lceil \log(T) \rceil \cdot T - 1} X_t > d_2] \leq \mathbb{P}[\sum_{t=0}^{\lceil \log(T) \rceil \cdot T - 1} Y_t > d_2]$. Now we can bound the probability that we care about,

$$1 - \mathbb{P}\left[\sum_{t=0}^{\infty} \gamma^t \mathcal{C}(\langle s_t, q_t \rangle) \leq d_2\right] = \mathbb{P}\left[\sum_{t=0}^{\infty} \gamma^t \mathcal{C}(\langle s_t, q_t \rangle) > d_2\right] \tag{149}$$

$$= \mathbb{P}\left[\sum_{t=0}^{\infty} \gamma^t X_t > d_2\right] \tag{150}$$

$$= \mathbb{P}\left[\sum_{t=0}^{\lceil \log(T) \rceil \cdot T - 1} \gamma^t X_t + \sum_{t=\lceil \log(T) \rceil \cdot T}^{\infty} \gamma^t X_t > d_2\right] \tag{151}$$

$$\leq \mathbb{P}\left[\sum_{t=0}^{\lceil \log(T) \rceil \cdot T - 1} X_t + 1 > d_2\right] \tag{152}$$

$$\leq \mathbb{P}\left[\sum_{t=0}^{\lceil \log(T) \rceil \cdot T - 1} Y_t + 1 > d_2\right] \tag{153}$$

$$= \mathbb{P}\left[\sum_{t=0}^{\lceil \log(T) \rceil \cdot T - 1} Y_t > \lceil \log(T) \rceil \cdot T \cdot p_1 + d_2 - \lceil \log(T) \rceil \cdot T \cdot p_1 - 1\right] \tag{154}$$

$$= \mathbb{P}\left[\sum_{t=0}^{\lceil \log(T) \rceil \cdot T - 1} Y_t > \mathbb{E}\left[\sum_{t=0}^{\lceil \log(T) \rceil \cdot T - 1} Y_t\right] + d_2 - \lceil \log(T) \rceil \cdot T \cdot p_1 - 1\right] \tag{155}$$

$$\leq \exp\left(-\frac{2 \cdot (d_2 - \lceil \log(T) \rceil \cdot T \cdot p_1 - 1)^2}{\sum_{t=0}^{\lceil \log(T) \rceil \cdot T - 1} (\max\{Y_i\} - \min\{Y_i\})^2}\right) \tag{156}$$

$$= \exp\left(-\frac{2 \cdot (d_2 - \lceil \log(T) \rceil \cdot T \cdot p_1 - 1)^2}{\lceil \log(T) \rceil \cdot T}\right) \tag{157}$$

Here the first inequality (Eq. 152) comes from the following two facts, certainly $\sum_{t=0}^{\lceil \log(T) \rceil \cdot T - 1} \gamma^t X_t \leq \sum_{t=0}^{\lceil \log(T) \rceil \cdot T - 1} X_t$ and we have that $\sum_{t=\lceil \log(T) \rceil \cdot T}^{\infty} \gamma^t X_t \leq 1$. The second fact is a little harder to see, first we note that $\lim_{\gamma \to 1} \gamma^T = 1/e$, where $T = 1/(1 - \gamma)$ is the effective horizon. Then we can rewrite,

$$\sum_{t=\lceil \log(T) \rceil \cdot T}^{\infty} \gamma^t X_t = \left(\gamma^{\lceil \log(T) \rceil \cdot T}\right) \cdot \left(\sum_{t=\lceil \log(T) \rceil \cdot T}^{\infty} \gamma^{t - \lceil \log(T) \rceil \cdot T} X_t\right) \tag{158}$$

$$= \left((\gamma^T)^{\lceil \log(T) \rceil}\right) \cdot \left(\sum_{t=\lceil \log(T) \rceil \cdot T}^{\infty} \gamma^{t - \lceil \log(T) \rceil \cdot T} X_t\right) \tag{159}$$

$$\leq \left(\frac{1}{e}^{\lceil \log(T) \rceil}\right) \cdot \left(\frac{1}{1 - \gamma}\right) \leq \left(\frac{1}{e}^{\log(T)}\right) \cdot T = \frac{1}{T} \cdot T = 1 \tag{160}$$

The second inequality (Eq. 153) comes from our earlier construction. The final inequality (Eq. 156) is obtained from Hoeffding's inequality [40] for bounded random variables. Finally, by bounding the final expression (Eq. 157) from above by $\delta_2$ and rearranging gives the desired result. $\qquad \square$

**Proposition G.14.** *A feasible policy $\pi$ for Problem G.12 with parameters $\delta_2 \leq p_1$ and $d_2 < \gamma^{T+H}$, satisfies $\Pr(\langle s_t, q_t \rangle \models \Diamond^{\leq H} accept) \leq p_1$ up to the effective horizon $T = 1/(1 - \gamma)$. This bound is tight.*

*Proof.* A feasible policy $\pi$ for Problem G.12 with parameters $\delta_2 \leq p_1$ and $d_2 < \gamma^{T+H}$ clearly implies that,

$$\mathbb{P}\left[\sum_{t=0}^{\infty} \gamma^t \mathcal{C}(\langle s_t, q_t \rangle) < \gamma^{T+H}\right] \geq 1 - p_1 \tag{161}$$

and certainly for all $t' \in [0, T]$ we have that,

$$1 - p_1 \leq \mathbb{P}\left[\sum_{t=0}^{\infty} \gamma^t \mathcal{C}(\langle s_t, q_t \rangle) < \gamma^{T+H}\right] \tag{162}$$

$$\leq \mathbb{P}\left[\sum_{t=t'}^{t'+H} \gamma^t \mathcal{C}(\langle s_t, q_t \rangle) < \gamma^{T+H}\right] \tag{163}$$

$$= \mathbb{P}\left[\gamma^{t'} \sum_{t=t'}^{t'+H} \gamma^{t-t'} \mathcal{C}(\langle s_t, q_t \rangle) < \gamma^{T+H}\right] \tag{164}$$

$$= \mathbb{P}\left[\sum_{t=t'}^{t'+H} \gamma^{t-t'} \mathcal{C}(\langle s_t, q_t \rangle) < (\gamma^{T+H}/\gamma^{t'})\right] \tag{165}$$

$$\leq \mathbb{P}\left[\sum_{t=t'}^{t'+H} \gamma^{t-t'} \mathcal{C}(\langle s_t, q_t \rangle) < \gamma^H\right] \tag{166}$$

Let $\Pr(\langle s_{t'}, q_{t'} \rangle \models \Diamond^{\leq H} accept)$ denote the proportion of accepting paths at timestep $t'$, where $s_{t'} \sim \mathcal{P}_{t'}(\cdot)$. Here $\mathcal{P}_{t'}(\cdot)$ denotes the marginal state distribution at time $t'$. Recall that for each path $\rho \in \mathcal{S}^\omega$ and corresponding $trace(\rho) \in \Sigma^\omega$ such that $trace(\rho) \models \Diamond^{\leq H} accept$ the sum $\sum_{t=t'}^{t'+H} \gamma^{t-t'} \mathcal{C}(\langle s_t, q_t \rangle) \geq \gamma^H$. Without loss of generality fix some $t' \in [0, T]$ and suppose that $\Pr(\langle s_{t'}, q_{t'} \rangle \models \Diamond^{\leq H} accept) > p_1$. This implies that,

$$\mathbb{P}\left[\sum_{t=0}^{\infty} \gamma^t \mathcal{C}(\langle s_t, q_t \rangle) \geq \gamma^{T+H}\right] \geq \mathbb{P}\left[\sum_{t=t'}^{t'+H} \gamma^{t-t'} \mathcal{C}(\langle s_t, q_t \rangle) \geq \gamma^H\right] > p_1 \tag{167}$$

Which is a contradiction. Therefore, it must be the case that when Eq. 161 is satisfied then so is $\Pr(\langle s_t, q_t \rangle \models \Diamond^{\leq H} accept]) \leq p_1$ for all $t \in [0, T]$. To prove that this bound is tight we can show the possible existence of a counter example. In particular, we want to prove the following statement, a policy $\pi$ satisfying,

$$\mathbb{P}\left[\sum_{t=0}^{\infty} \gamma^t \mathcal{C}(\langle s_t, q_t \rangle) < \gamma^{T+H}\right] \geq 1 - (p_1 + c) \tag{168}$$

for some constant $c > 0$ does not imply that,

$$\Pr\left(\langle s_t, q_t \rangle \models \Diamond^{\leq H} accept\right) \leq p_1 \quad \forall t \in [0, T] \tag{169}$$

We will show that there may exist some policy $\pi$ that satisfies Eq. 168 but does not satisfy Eq. 169 at some timestep $t$. Firstly, we assume $\pi$ is such that Eq. 168 holds, this implies that for all $t' \in [0, T]$ we have,

$$1 - (p_1 + c) \leq \mathbb{P}\left[\sum_{t=0}^{\infty} \gamma^t \mathcal{C}(\langle s_t, q_t \rangle) < \gamma^{T+H}\right] \tag{170}$$

$$\leq \mathbb{P}\left[\sum_{t=t'}^{t'+H} \gamma^t \mathcal{C}(\langle s_t, q_t \rangle) < \gamma^{T+H}\right] \tag{171}$$

$$\leq \mathbb{P}\left[\sum_{t=t'}^{t'+H} \gamma^{t-t'} \mathcal{C}(\langle s_t, q_t \rangle) < \gamma^H\right] \tag{172}$$

Fix some $t' \in [0, T]$ and let $\Pr(\langle s_{t'}, q_{t'} \rangle \models \Diamond^{\leq H} accept)$ denote the proportion of accepting paths at timestep $t'$. Suppose that $\pi$ is such that $\Pr(\langle s_{t'}, q_{t'} \rangle \models \Diamond^{\leq H} accept) > p_1$. Again recall that for each path $\rho \in \mathcal{S}^\omega$ and corresponding $trace(\rho) \in \Sigma^\omega$ such that $trace(\rho) \models \Diamond^{\leq H} accept$ the sum $\sum_{t=t'}^{t'+H} \gamma^{t-t'} \mathcal{C}(\langle s_t, q_t \rangle) \geq \gamma^H$, and so,

$$p_1 + c \geq \mathbb{P}\left[ \sum_{t=0}^{\infty} \gamma^t \mathcal{C}(\langle s_t, q_t \rangle) \geq \gamma^{T+H} \right] \tag{173}$$

$$\geq \mathbb{P}\left[ \sum_{t=t'}^{t'+b} \gamma^{t-t'} \mathcal{C}(\langle s_t, q_t \rangle) \geq \gamma^H \right] > p_1 \tag{174}$$

Now clearly for all $p_1 \in [0, 1]$, and $c > 0$, the following holds,

$$p_1 < p_1 + c \tag{175}$$

This implies that there may exist some $\pi$ satisfying Eq. 168 such that $\Pr(\langle s_{t'}, q_{t'} \rangle \models \Diamond^{\leq H} accept) > p_1$, i.e. does not satisfy Eq. 169 at timestep $t = t'$. $\qquad \square$

