# OpenReview forum: "Shielding Regular Safety Properties in Reinforcement Learning"
_NeurIPS.cc/2024/Conference — Submitted to NeurIPS 2024_

### Official Review · Reviewer_5bwE · 2024-07-07

**Soundness:** 4
**Presentation:** 4
**Contribution:** 3
**Rating:** 7
**Confidence:** 4

**Summary:**

The paper proposes an online shielding approach for safe RL that does not assume prior knowledge of environment dynamics and utilizes finite-horizon model checking with learned approximations of the environment dynamics. It specifically focuses on RL with regular safety properties provided as a PCTL formula. The authors present a framework that dynamically identifies unsafe actions and deploys a safe backup policy when necessary. The main technical contributions of the paper are:

- Definition of a constrained RL problem based on regular safety properties.
- Presentation of model checking algorithms to verify finite-horizon satisfaction probability.
- Development of sample complexity results for statistical model checking procedures.

The novelty of the paper lies in its approach to reinforcement learning with regular safety properties without requiring prior knowledge of environment dynamics. Unlike traditional shielding approaches that need full environment models or simulators, this framework uses learned approximations and finite-horizon model checking. The authors represent the synthesis problem as finding an optimal policy under a constraint that the resultant product Markov chain (from the policy) and the DFA (from the safety property) has probability $\leq p_1$ of violating the safety property in finite horizon $H$. The authors make use of the CMDP formulation to separate the reward and safety considerations by treating the safety property as a cumulative constraint in the CMDP.

**Strengths:**

Overall, this is a very strong submission and very clearly written. The authors present their problem statement precisely and also compare against many other related settings. This is challenging to do in safe RL since it is such a wide field with many parallel approaches, but I feel the authors did a commendable job here, especially in demonstrating how related and alternative formulations can be represented in their setting.

This is a very interesting combination of shielding from temporal logic specifications with a CMDP formulation. Especially, as the framework allows for regular safety properties instead of just invariant properties, it allows for quite general specifications.

The authors also do a good job in showing the generality of the work as it pertains to levels of model knowledge.

**Weaknesses:**

The main weakness I can see in this paper is perhaps a lack of experiments in settings with more complex models and safety specifications. Especially settings in which safety and optimality of the reward are in conflict, it would be interesting to see how the agent is able to achieve a trade off.

Another consideration is perhaps motivation of using a CMDP as a framework for safe RL. I don't see this as well motivated in the paper. There are several approaches to safe RL that use MDPs and are able to use probabilistic model checking type techniques to give guarantees for cost thresholds.

**Questions:**

Is there any similarities in the approach from the literature in shielding for POMDPs? E.g.,"Safe Reinforcement Learning via Shielding for POMDPs" by Carr et. al. Especially as one of the contributions in this paper is that the authors assume no knowledge of the dynamics. While the setting is not quite the same, I think there may be overlaps in the concept on how Carr et al, are able to shielding guarantees on a partial model

Typos
line 188 - " The hyperparameter p1 is be direct" grammar issue

lines 370, 408, 952, 983 - "loose" -> "lose"

[45,46] are the same reference.

**Limitations:**

There is some discussion of limitations in the section before the conclusion. I agree with the authors on the downsides of separating reward and safety.

---

> ### Author Rebuttal · Authors · 2024-08-05
>
> We thank the reviewer for their time in constructing their review.
>
> Re weaknesses: we point the reviewer to the experiments involving property 3 on our colour grid-world environment, in this instance there is a direct conflict between optimal reward and constraint satisfaction, this is reflected by the fact that normal Q-learning obtains the same optimal reward whereas our shielding approach obtains good reward within the constraints of the problem (i.e. Problem 4.1). Some more extended discussion of this is given in Appendix D.1. Similarly for the deep RL experiments, property 2 conflicts with the optimal reward policy and so blindly maximising reward will result in poor safety performance, this is reflected by the results in Figure 3, we refer the reviewer to Appendix D.2 for more details.
>
> With regards to the CMDP formulation, we agree that CMDPs have limitations, particularly when one considers expected cumulative cost constraints these offer little semantic meaning in terms of safety (see lines 24-26). We consider a CMDP formulation analogous to chance constraints but for a finite horizon H, we believe that this is more appropriate in the context of safety. We cannot provide absolute guarantees (100% safety) due to the statistical error that arises from approximating the dynamics of environment and sampling from the model. We would be grateful if you could share some of the approaches that you mentioned in your review to see if we can draw some parallels to our approach.
>
> Re questions: indeed, there are some similarities between our approach and Carr et. al., in particular, for the deep RL experiment the environment is partially observable. Our approach here is built upon approximate model-based shielding (AMBS) [1], which operates in the most permissive setting (no partial-knowledge of the POMDP transitions) and learns the dynamics of the environment with dreamer-v3. In Carr et. al. they assume access to a graph preserving approximation of the POMDP (known transitions but unknown probabilities) or a partial model, which also overapproximates the transition model of the POMDP, meaning that if there is a (non-zero probability) transition in the POMDP then it will also be in the partial model. This assumption is quite restrictive and as a result the experiments in Carr et. al. are constrained to grid-world environments where it is easy to specify a partial model. In our case, we can deploy our approach in high-dimensional visual-input settings for arbitrary regular safety properties. I would imagine coming up with a partial model for Seaquest would be near impossible given the observation space is very large. Furthermore, in Carr et. al. they consider reach avoid properties whereas we consider a more general class of safety properties (regular), although in principle these can be reduced to avoid properties on the product MDP/MC. That being said, we greatly thank the reviewer for bringing this paper to our attention. We also appreciate the reviewer for flagging some grammatical errors.
>
> [1] Goodall, Alexander W., and Francesco Belardinelli. "Approximate Model-Based Shielding for Safe Reinforcement Learning." ECAI 2023. IOS Press, 2023. 883-890.

---

> > ### Comment · Reviewer_5bwE · 2024-08-11
> >
> > Thanks for your detailed answers.
> >
> > I don't have any further comments

---

### Official Review · Reviewer_6qag · 2024-07-10

**Soundness:** 4
**Presentation:** 4
**Contribution:** 3
**Rating:** 7
**Confidence:** 4

**Summary:**

The authors present a new safe RL approach, building on safety shields.
The idea is to leverage model-checking techniques during the RL training to block actions that are identified as unsafe in the shield and use a learned backup policy if this is the case. In contrast to previous approaches, the "meta-algorithm" presented does not require an a-priori known model of the safety aspects of the environment. The approach comes with theoretical guarantees on the satisfaction of finite-horizon PCTL specifications and the evaluation assesses the potential of the proposed algorithm.

**Strengths:**

I'd first like to pinpoint that the paper is quite clear and very well written.

The proposed approach has several key advantages. First, it doesn't necessarily require providing a shield beforehand, in contrast to previous work. Second, the properties the agent needs to enforce during training are specified via PCTL, a well-established specification formalism that is not prone to exponential blow-up in the size of the automaton for translating the formula (as is the case for LTL). Third, the approach comes with guarantees on the shield: (i) when the hyper-parameters of the optimization procedure are well-chose, one can bound the probability of failure of the system from the initial state, (ii) the optimal policy found under the PCTL constraint is ensured to be a feasible policy for standard constrained MDP objectives. Finally, the authors discuss and provide guarantees under different assumptions, namely, the access to a model of the safety-relevant aspect of the environment (as in previous work), under a black-box model, and when one has access to an approximate model such that the total variation between the true and approximate transition probabilities is bounded. Additional statistical guarantees are provided for these last two assumptions.

Experiments successfully highlight the potential of the approach.

**Weaknesses:**

**Beyond tabular settings.**
The main concern I have with this paper is the fact that the guarantees seem to solely hold in the tabular setting, i.e., when the state-action space is finite and tractable. Notably, in the second round of experiments, the authors use Dreamer-v3 to learn an approximate model of the environment. Although the resulting method seems to outperform constrained RL methods, the theoretical guarantees do not hold in practice.

**On the assumptions.**
Assumption 5.2 is confusing. Indeed, when reading it for the first time, I thought one needs to have access to a *generative model*, i.e., a black box model of $\mathcal{P}$ that can be requested at any time, under any state and action, akin to the setting of [1, 2]. Specifically, this would mean that one doesn't necessarily need to sequentially execute the environment to obtain samples (as this is the case in RL), but, at any time, for any given state-action pair $(s, a)$, one could request the model to obtain a finite number of samples from $\mathcal{P}(\cdot \mid s, a)$. Note that this is not compliant with RL. When reading further, it seems that Monte-Carlo model checking only needs to produce episodes, which is fully compliant with RL. Thus, the distinction is really important here.

On another point, Assumption 5.3 looks rather restrictive. How to ensure that the approximate model learned through Algorithm 1 (line 300) yields a bounded total variation? This should be discussed in the main text. Moreover, the linked guarantees (Proposition 5.5) are not evaluated in the experiments. It could be interesting to have an example of how the statistical guarantees can be applied in practice.

[1] Michael J. Kearns, Yishay Mansour, Andrew Y. Ng: A Sparse Sampling Algorithm for Near-Optimal Planning in Large Markov Decision Processes. Mach. Learn. 49(2-3): 193-208 (2002)\
[2] Yujia Jin, Aaron Sidford: Towards Tight Bounds on the Sample Complexity of Average-reward MDPs. ICML 2021: 5055-5064

**Questions:**

- How do your guarantees apply when using dreamer as an approximate model of the environment? I guess the guarantees vanish here, and if it is the case, under which assumption Proposition 5.5 can be extended to still provide some guarantees?
- Could you clarify Assumption 5.2 (see the related point in the weaknesses).
- About Assumption 5.3, the total variation distance seems difficult (or even impossible) to formally check. Could you discuss this further?
- Line 110: you require the reward function to be bounded in $[0, 1]$. What is the impact of this assumption?

**Other minor remarks.**
- I think there is a slight bug in Definition 3.3; $trace(\rho)$ yields a word of infinite length while $\mathcal{L}(\mathcal{D})$ consists of words of finite length. So $trace(\rho)$ should never belong to  $\mathcal{L}(\mathcal{D})$.
- line 169: $\mathbb{Z}_{+}$: why not simply write $\mathbb{N}$?
- line 204: aren't you this way defining an $\omega$-automaton?
- line 249: I don't see why $H$ should be small. Most model-checkers rely on value-iteration-based engines with very large horizon, so I think $H$ shouldn't be a problem here, but rather the size of the formula and the size of the state space.
- line 289 in Algorithm 1: I guess the probability in the **if** statement is computed through $\hat{\mathcal{P}}$. This should be explicitly indicated.
- line 326: you first use $\text{Pr}^{min}$ and then $\inf_{\pi} \text{Pr}$. Doesn't the $\inf$ coincide with the $\min$ here? I found that formulation a bit confusing.
- In Table 1, invariant properties are unbounded. Isn't that disallowed in your approach?
- line 360: to me, the usage of PCTL* is not a slight detail; it embeds LTL that is quite different from PCTL; notably model checking algorithms are not polynomial in the size of the formula (but in the worst case doubly exponential).

**Limitations:**

Apart from the points raised above, the limitations of the work have been successfully addressed in the paper.

---

> ### Author Rebuttal · Authors · 2024-08-05
>
> We would like to thank the reviewer for taking the time to construct such an insightful review.
>
> We will clear up the questions you have, and I think this will deal with some of the weaknesses you have highlighted.
>
> Re questions:
> - When we start using neural networks or deep learning architectures like dreamer-v3 to model the environment dynamics, then most of our guarantees go out the window. However, not all is lost, as Proposition 5.5 still holds so long as the total-variation distance between the belief state representation of dreamer-v3 and the true belief state in the POMDP is upper bounded by the same quantity $\epsilon/H$, this is a bit handwavy, but essentially minimising TV distance between the belief state representation minimises an upper bound on the TV distance between the underlying state of the POMDP and this objective is directly encoded in the dreamer-v3 loss function, some of these details are described in AMBS [1] upon which our experiments are based and [2] provides a more formal proof of this upper bound.
>
> [1] Goodall, Alexander W., and Francesco Belardinelli. "Approximate Model-Based Shielding for Safe Reinforcement Learning." ECAI 2023. IOS Press, 2023. 883-890.
>
> [2] Gangwani, Tanmay, et al. "Learning belief representations for imitation learning in pomdps." uncertainty in artificial intelligence. PMLR, 2020.
>
> - If we have a generative model of the environment that we can query at any time from any state then this does break the standard RL setting and could have several implications, e.g. we could just construct an approximate MDP by sampling many times from each state and do value iteration (or similar) to get an approximate optimal policy without having to interact with the environment. To clarify, Assumption 5.2 means exactly this, we have a black box model of the true environment that we can query at any point from any state. We note that this is a weaker assumption than 5.1 (which also breaks the standard RL setting), although it is still strong. We stress that in our experiments we operate in the most permissive setting, where we don’t have the true probabilities $\mathcal{P}$ or access to a black box model of the environment. Assumption 5.2 is more of a ‘what if’ scenario, how could we still do model checking without the true probabilities $\mathcal{P}$ but with a black box model instead. We want to make it clear that in our experiments we do not have access to a black box model, or the true probabilities and we do Monte Carlo/statistical model checking with the approximate model (which is learned by experience generated from episodes of environment interaction). I think the confusion can be lifted here by understanding that we can do Monte Carlo model checking with either a black box model or with an approximate model it is not limited to either scenario. In Appendix F.2 we investigate the effect of given the shield access to the true probabilities $\mathcal{P}$ and access to sampling from $\mathcal{P}$.
> - It would not be possible to check that the TV distance for the next state distribution for some $(s, a) \in S \times A$ is bounded by $\epsilon$ without knowing the actual probability of interest ($\mathcal{P}(\cdot | s, a)$), i.e. we want to know if $D_{TV}(\mathcal{P}(\cdot | s, a), \widehat{\mathcal{P}}(\cdot | s, a)) \leq \epsilon$. We can have an arbitrary high-statistical confidence $(1- \delta)$ that $D_{TV}(\mathcal{P}(\cdot | s, a), \widehat{\mathcal{P}}(\cdot | s, a)) \leq \epsilon$ holds for any given $\epsilon$, provided that the number of samples from $\mathcal{P}(\cdot | s, a)$ is large enough. This constitutes the first half of the proof of Theorem 6.5. In particular, if any state-action pair $(s, a) \in S \times A$ has been visited at least $\mathcal{O}\left(\frac{H^2|\mathcal{S}|^2}{\epsilon^2} \log\left( \frac{|\mathcal{A}||\mathcal{S}|^2}{\delta}\right)\right)$ many times then $D_{TV}(\mathcal{P}(\cdot | s, a), \widehat{\mathcal{P}}(\cdot | s, a)) \leq \epsilon$ holds with probability roughly $(1- \delta)$ up to some constants.
> - I don’t think there is any significant impact of assuming that the rewards are in $[0, 1]$, most of our technical statements and proofs are only concerned with safety.
>
> Re minor remarks:
> - At a first glance there does seem to be a slight bug in Definition 3.3 but I will double check in (Baier & Katoen 2008).
> - I think we tried to exclude $H=0$ here but I don’t think it matters if we use $H \in \mathbb{N}$.
> - Indeed, this would be an $\omega$-automaton it might be nice to mention this, thanks for your suggestion.
> - Indeed, when we are exact model checking $H$ doesn’t matter that much.
> - Yes, that is correct we will make this clearer.
> - Yes, I agree this is slightly confusing we will change this and use min.
> - No we allow for any arbitrary regular safety properties not only bounded ones. Rather, we simply check the invariant property for a bounded lookahead horizon $H$. Under the assumption (Assumption 6.3) that $H$ is big enough to avoid irrecoverable states (Definition 6.2), then we can guarantee the satisfaction of the unbounded regular safety property for the entire episode length $T$, provided that $p_1$ is chosen appropriately, see Proposition 4.2. Note that we can’t include unbounded eventually as this would not be a safety property (rather liveness).
> - We are not claiming that we can check arbitrary PCTL* formula as indeed this suffers from similar problems as LTL. Rather we are checking regular safety properties, we just use PCTL/PCTL* notation to compactly define the regular safety properties in Table 1, rather than provide the full DFA. All the properties in Table 1 are regular safety properties as we can come up with a corresponding DFA for them, we thought that the keen-eyed reader might realise that the formulas in Table 1 are in fact PCTL* formula and not PCTL and thus we tried to add some clarification here. More precise semantics and details for these properties are given in Appendix D.1.

---

> > ### Comment · Reviewer_6qag · 2024-08-09
> > **Thank you for the answers**
> >
> > Thank you for having answered my questions.
> >
> > I think every point made in the authors' response should be made clear and mentioned in the paper, especially the part related to the use of neural networks. The fact that guarantees are limited when using dreamer should also be mentioned in the experimental section.

---

> ### Author Response · Authors · 2024-08-09
> **Re comment:**
>
> Thank you for your remarks, we will certainly make these points clear in the updated version of our paper.

---

### Official Review · Reviewer_amjX · 2024-07-10

**Soundness:** 3
**Presentation:** 2
**Contribution:** 1
**Rating:** 3
**Confidence:** 3

**Summary:**

This paper studies RL with 'regular' safety properties. The constraint of safe RL is based on the satisfaction of a logic formula in probability. The action from the 'backup' policy will proactively override the potentially unsafe action from RL to ensure/optimize safety, a typical shielding mechanism in formal safe control methods. The authors demonstrated the effectiveness of their approach against CMDP and regular RL (Q-learning) in two examples.

**Strengths:**

The approach is sound, as a typical shielding method, should work well in a safe RL setting, in 2 examples shown in the paper.
The problem studied in this paper is important.

**Weaknesses:**

1. Novelty. Novelty is my biggest concern for this paper. First, Problem 4.1 has already been discussed and solved in [1][2]. Second, the shielding approach is nothing new, a very standard way in formal methods. You can even trace back to simplex architecture with an advanced controller with safety back control. Third, the model-checking approach of this paper is not novel.
2. Significance. The experiments are weak with only 2 baselines (1 as original RL, 1 as CMDP) on 2 simple examples.

[1] Wang, Yixuan, et al. "Enforcing hard constraints with soft barriers: Safe reinforcement learning in unknown stochastic environments." International Conference on Machine Learning. PMLR, 2023.
[2] Wachi, Akifumi, et al. "Safe exploration in reinforcement learning: A generalized formulation and algorithms." Advances in Neural Information Processing Systems 36 (2024).

You may want to extend the related works part including two papers above and more.

**Questions:**

Please check the weakness part above.

**Limitations:**

The reviewer would like to know the limitations discussion by the authors in the rebuttal phase.

---

> ### Author Rebuttal · Authors · 2024-08-05
>
> We thank the reviewer for their time in constructing their review.
>
> Re weakness 1.: Indeed, similar chance constraints have been introduced and studied in several prior works including [1] and [2]. We thank the reviewer for bringing these papers to our attention and I believe paper [2] is cited in our related work section. With regards to [1] our setting is different, we consider discrete state-action MDPs with arbitrary size and transition dynamics, in [1] they consider continuous state-action MDPs with smooth and continuous transition dynamics and the rely on empirical estimation of a CBF to guarantee safety. I could imagine coming up with a CBF for non-smooth and discrete transition dynamics might be problematic. We also consider regular safety properties rather than box-constraints on the state space. Our framework is also flexible enough to be used to effectively shield high-dimensional visual-input settings such as Atari, and thus scales well. [2] can be considered as a shielding approach and is more closely related to our work, however in [2] they assume access to an emergency reset button, most algorithms do not have this privilege. Furthermore, they also consider a problem setting that constrains the cumulative cost below a pre-defined threshold with high-probability (this is formalised in our paper as Problem 4.5), this is distinctly different to our approach that constrains the probability of just one safety-violation (or cost) in the H-step horizon. We provide an exploratory comparison between our problem setup and Problem 4.5 in Appendix G.
>
> Re weakness 2.: To test the flexibility of our framework we first run experiments in a simple tabular setting where we could test each of the model checking paradigms (Appendix F.2), the results in the main paper corresponding to the most permissive setting (no prior model and no prior backup policy). We test with regular safety properties of increasing complexity and demonstrate the importance of using counter-factual experiences to train the backup policy (Appendix F.1). We also run experiments on Seaquest with two different regular safety properties, in total this is really 5 different problems over two different environments.  In terms of baselines, we used a Lagrangian version of dreamer-v3 similar to  [3],  to try and solve the CMDP. Evaluating against other CMDP RL algorithms like PPO-Lag would be disingenuous as PPO-Lag is a model free algorithm (rather than model-based) and would not have the capacity of the dreamer-v3 model and would exhibit significantly slower convergence and worse performance, this is demonstrated in works [3] and [4], where the model-based algorithms dramatically outperform model-free algorithms in terms of safety.
>
> [3]  Weidong Huang, Jiaming Ji, Borong Zhang, Chunhe Xia, and Yaodong Yang. 2023. Safe DreamerV3: Safe Reinforcement Learning with World Models. arXiv preprint arXiv:2307.07176524 (2023).
>
> [4] Yarden As, Ilnura Usmanova, Sebastian Curi, and Andreas Krause. 2022. Constrained policy optimization via bayesian world models. arXiv preprint arXiv:2201.09802 (2022).
>
> Re Limitations: Since we consider the most permissive setting where the transition probabilities are unknown and the backup policy is not provided a priori, then we cannot obtain the same strict safety guarantees that classical shielding approaches can. This is a trade-off of course, as classical shielding approaches are restricted to small grid world problems and require a significant amount of engineering effort from one environment to the next, whereas our approach can be used mostly off-the-shelf and operate in high dimensional environments. We can only obtain safety guarantees when the model of the environment is sufficiently accurate, and the backup policy has learned a good safe strategy. However, our guarantees are only probabilistic/statistical, so we can’t make formal statements about the entire system, but we can have well placed confidence in it. Further limitations include: lack of RL convergence guarantees – this is a key open problem with the shielding paradigm in general; reliance on the backup policy to operate safely, although for most settings low reward safe behaviour can be easily learnt (think breaking to a halt in a car); finally, Assumption 6.3 which requires the model checking horizon to be sufficiently large to avoid irrecoverable states, this Assumption is similar to the one made in [5] (think breaking to a halt in a car – it only takes a finite amount of time to stop to a halt).
>
> [5] Thomas, Garrett, Yuping Luo, and Tengyu Ma. "Safe reinforcement learning by imagining the near future." Advances in Neural Information Processing Systems 34 (2021): 13859-13869.

---

> ### Comment · Reviewer_amjX · 2024-08-10
>
> I appreciate the feedback from the authors. However, I still have some disagreements.
>
> _With regards to [1] our setting is different, we consider discrete state-action MDPs with arbitrary size and transition dynamics, in [1] they consider continuous state-action MDPs with smooth and continuous transition dynamics and the rely on empirical estimation of a CBF to guarantee safety. I could imagine coming up with a CBF for non-smooth and discrete transition dynamics might be problematic._
>
> This is not true for the reviewer.  CBF tries to solve a QP problem (optimization) or reduce loss function (learning-based) with a given dynamic model and can pick an action from the action sets to ensure forward invariance and safety. One can always learn this dynamic model with ML/neural network to do so in the unknown environment settings. And there is no fundamental difference between continuous and discrete settings. Even, a discrete setting is simpler as your optimization variable is reduced to a finite set.
>
> As for the experiments, the ML paper is expected to provide comprehensive experimental studies to show the effectiveness, especially for (Safe) RL community, as training RL is fundamentally unstable and varying.
>
> Given these, I would keep my current score as final.

---

> ### Author Response · Authors · 2024-08-12
> **Re comment:**
>
> Thanks for taking the time to construct a response.
>
> I will try to add some clarity as I think the part of the rebuttal you highlighted is not perfectly worded.
>
> I agree that in general discrete settings are easier to deal with than continuous settings. However, there are still some fundamental differences between the scope of our setup and the scope of paper [1]. In [1] the authors assume that the transition dynamics are smooth and continuous, and able to be modelled as an SDE. As a result the authors of [1] also explicitly state that their approach cannot handle hybrid dynamics (joint discrete and continuous dynamics) or discontinuous jump dynamics present in mujoco and safety gym (a popular safe RL benchmark). This has the following consequences:
> - The approach in [1] cannot be used to check arbitrary regular safety properties (only invariant properties), if we were to construct the product MDP between the approximated SDE and automaton, this would be a hybrid system (continuous dynamics + discrete transitions in the automaton) which [1] cannot deal with.
> - In the deep RL experiments we use Approximate Model-based Shielding (AMBS) [2] (and dreamer-v3) as the backbone of our approach, AMBS has been shown to work effectively for safety gym (see [3]) which the approach in [1] cannot.
> - In the deep RL experiments the environment is partially observable and the observation space is discrete but very high-dimensional (approximately 100,000 observations and 32,000 states), in our original statement/rebuttal we claimed that it would be very challenging to learn a smooth and continuous SDE from these dynamics as they are discrete and discontinuous and it is not clear how the approach of [1] might deal with partial observability.
>
> [1] Wang, Yixuan, et al. "Enforcing hard constraints with soft barriers: Safe reinforcement learning in unknown stochastic environments." International Conference on Machine Learning. PMLR, 2023.
>
> [2] Goodall, Alexander W., and Francesco Belardinelli. "Approximate Model-Based Shielding for Safe Reinforcement Learning." ECAI 2023. IOS Press, 2023. 883-890.
>
> [3] Goodall, Alexander W., and Francesco Belardinelli. "Leveraging Approximate Model-based Shielding for Probabilistic Safety Guarantees in Continuous Environments." Proceedings of the 23rd International Conference on Autonomous Agents and Multiagent Systems. 2024.
>
> With regards to our experiments, our paper is a hybrid theoretical and experimental paper, thus we dedicated an equal amount of time in rigorously proving the technical details of our paper and conceiving adequate experimental settings to test our approach.

---

### Official Review · Reviewer_sSLq · 2024-07-11

**Soundness:** 1
**Presentation:** 1
**Contribution:** 2
**Rating:** 3
**Confidence:** 4

**Summary:**

This paper presents an approach to online shielding for reinforcement learning agents. Namely, safety is formulated in probabilistic temporal logic with a parametric threshold as an indicator for reachability of the goal state. The proposed algorithm checks the reachability probability threshold in each state of the environment and raises a warning when the threshold is violated. A pre-trained backup policy is then proposed to be deployed which overrides the action of the agent. The approach is evaluated on tabular and visual RL benchmarks.

**Strengths:**

The paper addresses an interesting and valuable problem.

Evaluation includes visual RL benchmarks, which are interesting to provide online safety for.

**Weaknesses:**

The paper gives a lot of choice to the reader to compose a problem setting of their interest. It does not, however, provide precise enough approach description for each of them. This makes the contributions blurred. Described problem settings have been extensively studied before and the proposed approach does not significantly improve on them. It is also claimed that the approach can be used both during training and deployment, it is, however, not clear if the authors formulate these as two distinct settings and evaluate separately or not. In the latter case, it would be a dangerous simplification. Figures in the evaluation section are not readable.

Presentation: The presentation suffers from imprecise narrative leaving multiple questions until the evaluation section. Assumptions are introduced twice and it is not clear what exact problem the authors propose to address, or to what exact problem setting it generalizes. The use of "etc." and "some other" give the impression that more problems can be addressed than presented in the evaluation.

Minor:
- p.7: "if need be we"
- "don't" --> do not

**Questions:**

1. What problem settings and RL controllers does the meta-algorithm apply to precisely?
2. How exactly is the approximate model updated in case of the breach of probability threshold?
3. Hoeffding's inequality and Bernstein bounds for Monte-Carlo simulations are known to explode for rare events. How do the authors propose to tackle the challenge of guaranteeing safety for rare violations which are potentially of most interest when it comes to online safety?
4. When sampling from "approximate" model, what are the expected guarantees to be achieved,  i.e., how much is approximation error expected to contribute to the final statistical guarantee?
5. In evaluation: how does hyperparameter tuning effect the probability threshold in the probabilistic reachability specification?

**Limitations:**

The approach is of limited novelty and employs existing components. Technical details in terms of the exact problem setting and guarantees are insufficient to judge the contribution. The approach is not yet placed in the larger context of online safety for RL agents, which would require more precise problem formulation and discussion of limitations.

---

> ### Author Rebuttal · Authors · 2024-08-05
>
> We thank the reviewer for their time in constructing their review.
>
> Re summary: one minor inconsistency, the backup policy is not necessarily pre-trained, in our experiments the backup policy is trained online with RL to minimise cost, although our framework is flexible enough to allow for a pre-trained/handcrafted backup policy.
>
> Re weaknesses: The goal of our paper is to present a flexible framework for shielding RL policies w.r.t regular safety properties. A precise description of the problem setting is provided in Section 3. In particular, we consider discrete state and action MDPs augmented with a set of atomic propositions and corresponding labelling function – in line with standard model checking formalisms. The problem setup is then precisely described as a CMDP, see Problem 4.1, which introduces a new type of constraint, the (probabilistic) step-wise bounded regular safety property constraint. If you have examples of how our exact problem setup has been studied before we would very grateful if you could share them with us, so that we can improve our paper.
>
> The shield can be used during training and deployment, and we make no distinction between the two settings, this is justified as follows: a key property of shielding is that even after training with a shield the shield still needs to be put in place during deployment of the agent, see [1]. This is unavoidable for us and so the performance of the agent at the end of training will reflect the performance of the agent during deployment (assuming the agent is deployed in the same environment as in training). Solving the problem of mismatch between the training environment and the deployment environment is beyond the scope of our paper.
>
> [1] Alshiekh, Mohammed, et al. "Safe reinforcement learning via shielding." Proceedings of the AAAI conference on artificial intelligence. Vol. 32. No. 1. 2018.
>
> More readable figures are provided in Appendix D.1 and D.2.
>
> Re presentation: We very clearly state the assumptions of the different model-checking paradigms at the beginning of Section 5, each paradigm corresponds to different levels of prior knowledge available, see the global Author Rebuttal for more details. Rather than hone in on the specifics we demonstrate the flexibility of our framework in the tabular setting (where Q-learning is used) and the deep RL setting where Dreamer-v3 is used for both RL and dynamics learning. We thank the reviewer for the spelling and grammatical errors they picked up on.
>
> Re question 1.: The aim of our paper is to present a flexible framework, as such the choice of RL algorithm and dynamics learning depends on the actual environment. In our experiments, we considered tabular RL the obvious choice here is to use tabular Q-learning for task and backup policy optimisation and maximum likelihood to construct the approximate transition probabilities. For the deep RL setting we considered Seaquest (a high-dimensional visual-input Atari game), we opted for dreamer-v3 for both policy optimisation and dynamics learning. For continuous control settings one might consider, for example, TD3 for RL and Gaussian process to model the system dynamics.
>
> Re question 2.: The approximate model is updated with transition tuples $(s’, s, a)$ collected during interaction with the environment. If the agent violates the regular property (enters an accepting state), the experience from this interaction is incorporated in the approximate model, so that the next time the agent is in a similar situation they can more accurately estimate the probability of a safety-violation. In the case of our deep RL approach based on dreamer-v3 we refer the reviewer to AMBS [2] upon which our solution is built.
>
> [2] Goodall, Alexander W., and Francesco Belardinelli. "Approximate Model-Based Shielding for Safe Reinforcement Learning." ECAI 2023. IOS Press, 2023. 883-890.
>
> Re question 3.: Indeed, the types of statistical bounds that we use suffer from the problem of rare events. This issue is exacerbated by the fact that our framework constructs an approximate model of the environment dynamics from experience, if there is a very rare and catastrophic event that we might want to avoid we might never see it during training and be unaware of it and unable to accurately predict the probability of it happening.  I believe that such events require special consideration depending on the intended domain. For self-driving cars, we might have to augment our dataset with rare catastrophic experiences. We could then use sampling strategies like importance sampling to try and better understand the risk of these rare events.
>
> Re question 4.: We simply require that the TV distance between the true MC $P(\cdot | s)$ and the approximate MC $\widehat{P}(\cdot | s)$ is bounded by $\epsilon/H$ for all states $s \in S$. This is stated in assumption 5.3. This is all we need to guarantee that we can model check with the approximate model and obtain the same statistical guarantees.  So, an arbitrary approximation error of $\alpha$ contributes an error of $\alpha * H$ to the approximation error of the reachability probability, proof details are outlined in Appendix C.4.
>
> Re question 5.: We discuss this in some detail in Appendix D. The main hyperparameter of the colour gridworld environment is the $p$ hyperparameter which corresponds to the level of stochasticity of the environment, for larger $p$ the probability threshold $p_1$ may need to be more permissive to learn a policy with high reward, although in principle any $p_1$ could be used (although you might get a low reward policy). Further experimentation is also detailed in Appendix F.
>
> Re limitations: We believe that our work is transformative at the very least and presents a novel framework for shielding and guaranteeing the safety of RL agents w.r.t an interesting class of safety properties. We provide an extensive literature review which should make it clear where our work lies in the safe RL literature.

---

### Official Review · Reviewer_7jyh · 2024-07-19

**Soundness:** 3
**Presentation:** 1
**Contribution:** 1
**Rating:** 3
**Confidence:** 5

**Summary:**

This paper addresses safety and constraint compliance in deploying reinforcement learning (RL) systems. Such issues have triggered a vast body of research in the area of safe RL over the last few years. The paper introduces a safe RL framework for so-called regular safety properties, focusing on satisfying these properties with high probability. The paper compares and places this setup to common constrained Markov decision processes (CMDP) settings and presents a meta-algorithm with provable safety guarantees to prevent violations of regular safety properties during training and deployment. The approach is evaluated in both tabular and deep RL settings.

**Strengths:**

Safe RL is of utmost importance in order to enable RL agents to be successfully deployed in the real world. This paper tackles an important problem: How to ensure safety if the safety criteria go beyond standard reward or simple reachability features. The motivation of the paper is well done, and the placement in the literature is mostly complete and extensive.

**Weaknesses:**

While this paper addresses an important research question, I feel it is not ready for publication at a major ML conference. I will first summarize the issues that I see and then elaborate in more detail.

1. The claims made in the intro are not properly met. This concerns particularly the claim that the approaches are most permissive in terms of prior knowledge.

2. The evaluation is not sufficient. The experiments are very sparse and miss important information to assess the quality of the research.

3. The paper is packed with formal definitions, most of them being standard in the formal methods community. Conversely, little space is spent on describing the methods in detail and providing a proper evaluation.

I overall feel that the paper aims too high and wants to solve all kinds of aspects in model-based safe RL with shields. It would have been better to pick certain key aspects more clearly. As an example, the paper claims to
- not need a model
- to learn a model
- to use most of the available model checking paradigms (and even introduce them)
- evaluate with tabular and deep RL

Many of these aspects are then not properly discussed.

I will now comment on the previously listed weaknesses.

1. In the introduction, the authors claim to operate in a most permissive setting where environment dynamics are not known. Yet, then, they introduce three model-checking paradigms, which are essentially standard numerical model checking, Monte Carlo-based statistical model checking, or model checking with approximate models. In particular, the last version is shady. What is a guarantee for a learned model? How can a claim be made for a most permissive setting but still obtain hard guarantees when, as a consequence, all guarantees rely effectively on statistics or a learned model? Other approaches are simply very clear about their assumptions, such as knowing a model. The model learning procedure is standard, and if the whole 'permissiveness' of the setting depends on the fact that a model can learned, I do not see a novel approach here.

Very importantly: While a model is learned, no safety guarantees can be given, defying the notion of shielded RL. This aspect is not discussed in the paper. A step further, even the safe fallback policy that is explained in Section 6 relies on learning.

2. The evaluation only considers a few environments and compares a simple tabular and deel RL agent. What I would have liked to see is how the different assumptions and model checking paradigms affect the learning, the safety, the performance. I believe, with a thorough evaluation in this direction, the paper would be much stronger. If we can believe that it is feasible to learn a model, then I would like to see a comparison between the strong assumption of knowing the model, and having learned a model, and what the effects on safety guarantees are.

3. The contributions only start at page 5, and are then still interleaved with standard definitions. It is important to have a paper safe-contained, but, as an example, why is it necessary to introduce both LTL, PCTL, DFA in detail? And then, the evaluation even uses PCTL^*. To me this seems as if the space had been filled up with long definitions, while it should have been used more on the contributions of the paper.

Generally, I feel the paper follows a great direction, especially investigating Shielded RL with a learned model. With a stronger evaluation and emphasis of this part, the research and contribution could be much improved in my opinion.

Minor comments:

- proposition 4.2 is obvious and seems like an overformal statement of a simple fact.

- Def 4.3. Make clear that this is reward engineering

- Non-Markovian cost: Compare to reward machines

- The tradeoff between safety and exploration has (for instance) been investigated in

Carr et al.: Safe Reinforcement Learning via Shielding under Partial Observability. AAAI 2023.

- Learning the model, proposition 5.5: I think the topology (graph) of the model needs to be known to estimate the probabilities.

**Questions:**

Beyond the points I raise above, I have the following questions.

1. Please comment on the evaluation and effects of different assumptions on prior knowledge.

2. Do you need to know the topology of the MDP (Markov chain) to effectively learn a model?

3. Can you comment on the limitations of your work and thereby sharpen the contribution?

**Limitations:**

The limitations of the work in terms of assumptions and their real-world relation are not properly explained in the paper.

---

> ### Author Rebuttal · Authors · 2024-08-05
>
> We would like to thank the reviewer for their extensive review.
>
> Re weakness 1.: In our paper we consider the most permissive setting where no prior knowledge about the transition probabilities is required. We only require that there is a labelling function defined over the state space of the MDP and that the safety property is a regular safety property given as a DFA. In Section 5 we introduce 3 model checking paradigms, namely exact model checking that assumes the transition probabilities are known (as in previous work), Monte Carlo/statistical model checking with a black box model and model checking with an approximate model of the transition probabilities. As is stated in Proposition 5.5 we can obtain an epsilon-approximate estimate for the corresponding reachability probability in the product MC, either by exact model checking or by statistical model checking with the approximate model, to verify the validity of this statement we refer the reviewer to Appendix C.5 which contains the full proof. The goal of our paper is to develop a framework (or meta-algorithm) that is flexible enough so that it can operate in each paradigm, including the most permissive setting where the approximate model is learned from experience (i.e. no prior knowledge). We stress that these guarantees are probabilistic or analogously PAC as we rely on Hoeffding/Bernstein type bounds for both statistical model checking (Proposition 5.4) and learning the transition probabilities (Theorem 6.5).
>
> Re safety guarantees during learning: Before the model is sufficiently accurate and the backup policy has converged then there are no safety guarantees, this is semi-implied by Theorem 6.5 and the preceding assumption statements. However, as a consequence we can operate in with no prior knowledge.
>
> Re weakness 2.: The results provided in the main paper are obtained under the most permissive setting (no prior model and backup policy), you can find a comparison with each of the model checking paradigms in Appendix F.2, we also clearly state on line 369: “For example, we show that we don't lose much by using Monte Carlo model checking as opposed to exact model checking with the `true' probabilities”. This is reflected in the results in Appendix F.2. We might expect to lose some safety performance when operating in the most permissive setting, however since the environment is small, we are able to quickly learn an accurate model of the transition probabilities, and we can learn a safe backup policy quickly with counterfactual experiences and the cost function in Definition 4.3. For the deep RL experiments we were unable to provide a similar comparison as the observation/state space is too large and transition probabilities can’t be modelled in a compact way.
>
> Re weakness 3.: This was one of the main difficulties of writing this paper, we wanted the paper to be self-contained and accessible to a general audience that is not necessarily familiar with model checking. In particular, we introduce labelled MDP and MC which are a necessary formalisms for model checking, we introduce PCTL to provide standard notation for defining our safety properties later in the paper and to provide probabilistic semantics, we then introduce regular safety properties, DFA and the product MC which we also feel as necessary to fully grasp what is going on. In total this part of the paper spans little over 1.5 pages which we do not find as unreasonable, the fact our main contributions only start on page 5 is perhaps skewed by the fact that we provide the related work earlier in the paper. The definitions interleaved throughout the rest of the paper are necessary and non-standard (Problem 4.1 and Definition 4.3).
>
> Re minor comments: Proposition 4.2 is not too difficult to see, although the proof requires a union bound and we thought it necessary to be precise and provide this proof in Appendix C. We thank you for your other comments and we will take these into consideration.
>
> Re question 1.: As stated in (Re weakness 2.) the evaluation of different assumption on prior knowledge is presented in appendix F.2. See also (Re weakness 2.) for my comments on this.
>
> Re question 2.: No, unlike Carr et al. we don’t explicitly need the topology of the MDP to learn a model, see my response to reviewer 5beW where I highlight the differences between our approach and Carr et al.  In the most general setting, we can assume every state is reachable from every other state and we can simply learn the probability vector for each state-action pair using maximum likelihood, this is reflected by the strong dependence on $|S|$ in Theorem 6.5. If we know something about the topology of the MDP, e.g. how many successor states there are or the graphical structure of the MDP then the MDP can certainly be more quickly learnt. This is briefly discussed in the main body of the paper (lines 336-337).
>
> Re question3.: We try and break the mould of classical shielding approaches that assume knowledge of the transition dynamics or safety dynamics (or a partial model Carr et al.), this means we lose strict safety guarantees. This is a trade-off of course, by operating in the most permissive setting which assumes no prior knowledge we are not constrained to small grid world environments - we can operate in high dimensional environments, and we minimise the amount of prior engineering effort required when deploying in a new environment. We can only obtain safety guarantees when the model of the environment is sufficiently accurate, and the backup policy has learned a good safe strategy. Further limitations include lack of convergence guarantees (an open problem with the shielding paradigm), reliance on the backup policy, etc. see more limitations provided in the author rebuttal above.
>
> Re limitations: the relation between our problem setting and real-world considerations could be discussed in the opening paragraphs.

---

> > ### Comment · Reviewer_7jyh · 2024-08-11
> > **Thank you for the response, I still have questions.**
> >
> > I thank the authors for their detailed response and for clarifying the correctness of the approach. However, I am still unconvinced that learning an MDP model from data is a good approach in a safe RL (shielded RL!) setting. I quote: 'we are able to quickly learn an accurate model of the transition probabilities, and we can learn a safe backup policy quickly with counterfactual experiences and the cost function in Definition 4.3'. What does it mean to quickly learn an accurate model, or to learn a backup policy quickly? Are these statements backed up by numbers in the paper or the appendix? This would be a crucial point for me in the evaluation of the paper.
> >
> > Regarding the other replies, I still think that there are too many standard definitions whose space would be better spent explaining and evaluating the method. Moreover, I would like some more (technical) detail on how it is possible to loose the assumption of knowing the topology of the MDP.

---

> ### Author Response · Authors · 2024-08-13
> **Re more questions:**
>
> Thanks for your response.
>
> Re learning an MDP model: We note that there are several approaches to safe RL that aim to learn a dynamics model/model of the MDP or similar. Some of these have been highlighted by other reviewers and include [1-2], shielding approaches that learn a model of the MDP inlcude [3-4], other approaches based on formal methods/LTL constraints include [5-6], methods adapted to high-dimensional visual-input environments like Atari and safety gym include [7-8]. Many of these approaches can be classified as model-based which have been shown to have demonstrably better performance in terms of cumulative safety violations when compared with model-free algorithms (e.g. PPO-LAG, TRPO-LAG, CPO), see [7-8]
>
> [1] Wang, Yixuan, et al. "Enforcing hard constraints with soft barriers: Safe reinforcement learning in unknown stochastic environments." International Conference on Machine Learning. PMLR, 2023.
>
> [2] Wachi, Akifumi, et al. "Safe exploration in reinforcement learning: A generalized formulation and algorithms." Advances in Neural Information Processing Systems 36 (2024).
>
> [3] Goodall, Alexander W., and Francesco Belardinelli. "Approximate Model-Based Shielding for Safe Reinforcement Learning." ECAI 2023. IOS Press, 2023. 883-890.
>
> [4] He, Chloe, Borja G. León, and Francesco Belardinelli. "Do androids dream of electric fences? safety-aware reinforcement learning with latent shielding." arXiv preprint arXiv:2112.11490 (2021).
>
> [5] Hasanbeig, Mohammadhosein, Alessandro Abate, and Daniel Kroening. "Cautious Reinforcement Learning with Logical Constraints." Proceedings of the 19th International Conference on Autonomous Agents and MultiAgent Systems. 2020.
>
> [6] Hammond, Lewis, et al. "Multi-Agent Reinforcement Learning with Temporal Logic Specifications." Proceedings of the 20th International Conference on Autonomous Agents and MultiAgent Systems. 2021.
>
> [7] As, Yarden, et al. "Constrained policy optimization via bayesian world models." arXiv preprint arXiv:2201.09802 (2022).
>
> [8] Huang, Weidong, et al. "Safe dreamerv3: Safe reinforcement learning with world models." arXiv preprint arXiv:2307.07176 (2023).
>
> Re quickly learning an accurate model and backup policy: we quickly ran our colour gridworld experiments again and plotted the total number of state-action pairs that satisfy Assumption 5.3 throughout training. We also plotted the total number of $p_1$-recoverable states w.r.t the backup policy (during training), in this plot we provide the maximum number of $p_1$-recoverable states verified by the PRISM model checker. Unfortunately there is not a way to share these plots with you at this time, however we observed the following general trends:
> - For property 1 the total number of $p_1$-recoverable states (backup policy) converges to the maximum idenitfied by PRISM within 2 epsiodes (\~2000 environment steps). For property 2 and 3 the total number of $p_1$-recoverable states does not reach the maximum identified by PRISM and converges to a number slightly less after around 40 epsiodes (\~40000 environment steps), this is because the exploration strategy is limited, if we don't explore every state-action pair infinitely often then Q learning does not necessarily guarantee us the optimal policy, in practice we really just require that the states with high probability under the task policy state distribution are $p_1$-recoverable with the backup policy.
> - For the total number of accurate state-action pairs, we see that this steadily increases and once the task policy has converged it plateus as the state distribution induced by the task policy remains the same. Ideally we would like all state action pairs to satisfy Proposition 5.3, however this is again limited by the exploration strategy - once the task policy has converged there are many state-action pairs we will not see, in practice we really just require the safety critical state-action pairs to be accurate in our model.
> We will include these details in a revised version of our paper.
>
> Re dropping the assumption of knowing the topology of the MDP: in general if the topology of the MDP is unknown then we can assume that every state is reachable from every other state. Let $v(s, a)$ denote the total number of times that $(s,a)$ had been observed and let $v(s', s, a)$  denote the total number of times that $(s,a) \to s'$ has been observed. We start by initializing the visit counts $v(s, a)= |S|$ and $v(s', s, a)=1$. The maximum likelihood probability of $s'$ given $(s, a)$ is simply $\widehat{\mathcal{P}}(s' | s, a) = v(s', s, a)/v(s,a)$ giving us the intial probability vector $\widehat{\mathcal{P}}(\cdot | s, a) = [1/|S|, ..., 1/|S|]$. Every time we see a transition $(s,a) \to s'$ we update the corresponding visit counts $v(s, a) += 1$ and $v(s', s, a) += 1$, we then use Hoeffding's to bound the difference between our maximum likelihood estimate and the true probability $\mathcal{P}(s' | s, a)$. Exact details are given in Appendix C.5.

---

### Author Rebuttal · Authors · 2024-08-05

There is some apparent confusion between the claims made in our paper and the different model checking paradigms that we present. The main goal of our paper is to present a flexible framework for shielding RL agents w.r.t arbitrary regular safety properties (i.e. those properties that can be expressed by the acceptance of a DFA, where accepting path => unsafe). We wanted our framework to be able to operate under various scenarios/levels of prior knowledge about the transition dynamics of the environment, including the most permissive setting where we have no knowledge of the transition dynamics (or topology of the MDP) and no access to a backup policy. We list the following paradigms:
1. We have full knowledge of the transition probabilities of the environment (this is the typical assumption made in line with prior work on shielding). In this instance we use standard PCTL model checking techniques which operate in time O(H* size(formula)*size(MC)), these techniques are standard and presented in (Baier & Katoen 2008), we feel it unnecessary to include the details for this in the main paper, although for reference the procedure is detailed in Appendix A.
2. We have access to a black box model of the environment that can be queried at any time from any state and used to simulate possible future roll-outs. In this scenario we can use Monte Carlo/statistical model checking to compute the reachability probability $\Pr(\langle s, q \rangle \models \lozenge^{\leq H} accept)$ up to some $\epsilon$-error and with high probability $(1 - \delta)$.
3. We do not have full knowledge of the transition probabilities but we have access to an approximate dynamics model where the TV distance $D_{TV}(P(\cdot | s, a), \widehat{P}(\cdot | s, a)) \leq \epsilon/H$ is bounded. In this instance we can use either exact PCTL model checking or Monte Carlo model checking (using \widehat{P}) to get analogous results. NOTE: in the tabular case we can construct an approximate dynamics model that satisfies $D_{TV}(P(\cdot | s, a), \widehat{P}(\cdot | s, a)) \leq \epsilon/H$ by interacting with the environment and using experience tuples $(s’, s, a)$ to update a simple maximum likelihood model of the transition probabilities, the proof of this is provided in the first half of the proof of Theorem 6.5.

(1.) and (2.) clearly break the standard assumptions of RL. Therefore, in the experiments presented in the main body of the paper we operate in the most permissive setting, where an approximate model is constructed with experience generated by environment interaction and the approximate model (with Monte Carlo model checking) is used to check that $\Pr(\langle s, q \rangle \models \lozenge^{\leq H} accept) \leq p_1$ at each timestep. The backup policy is also constructed online with RL by minimizing the cost function (Definition 4.3), so no prior knowledge is needed.

We also noticed that some of the reviewers found similarities between our approach and “Safe Reinforcement Learning via Shielding for POMDPs" by Carr et. al. We highlight some of the differences here:
- Carr et. al. consider POMDPs we consider both fully observable MDPs in our first set of experiments and we study POMDPs in our deep RL Atari experiments with Seaquest.
- Carr et. al. Assume access fo a graph-preserving partial-model of the POMDP, which means that all state -> observation and state -> state transitions are known but the probabilities are unknown. They also assume that the partial-model overapproximates the true POMDP transitions which means that if there is a non-zero transition in the POMDP then it also exists in the partial-model. On the other hand, we do not assume access to this partial model or anything similar, we simply learn a latent dynamics model from experience (leveraging dreamer-v3).
- Carr et. al.’s assumptions mean that their experiments are restricted to small grid-world examples where it is easy to come up with a partial model of the POMDP. In our case we can operate on high-dimensional Atari games.
- Finally, Carr et. al. Consider reach avoid properties, while we consider a more general class of safety properties (regular), although in principle these can be reduced to avoid properties on the product MDP/MC.

We noticed that some reviewers wished us to elaborate more on the limitations of our approach. We summarise the limitations here:
- We consider the most permissive setting where the transition probabilities are unknown, and the backup policy is not necessarily provided beforehand. As such everything needs to be learnt from experience generated by interaction with the environment. Thus, we cannot provide strict safety guarantees at the beginning of training as we would need to experience safety-violations before we can accurately estimate their risk or probability of happening.
- Once enough experience is gathered and the backup policy has learnt a good safe strategy then we can provide safety guarantees in the tabular setting (Theorem 6.5). However, these guarantees are only probabilistic/statistical (or PAC) as they are limited by the estimation of the transition probabilities and the statistical error and failure probability of Monte Carlo model checking.
- We also lose RL asymptotic optimality/convergence guarantees – this is a key open problem with shielding in general that has not been massively explored.
- We train the backup policy online; this requires us to experience safety violations and this also relies on good hyperparameter choice for the backup policy to converge.
- Assumption 6.3 which requires the lookahead horizon to be sufficiently large to avoid any irrecoverable states. Similar assumptions have been made in [1] (think breaking to a halt in a car – it only takes a finite amount of time to stop to a halt).

[1] Thomas, Garrett, Yuping Luo, and Tengyu Ma. "Safe reinforcement learning by imagining the near future." Advances in Neural Information Processing Systems 34 (2021): 13859-13869.

---

### Decision · Program_Chairs · 2024-09-25

**Decision:**

Reject

**Comment:**

Considering this paper has split reviews, I took the time to also read the paper and after a careful discussion with the reviewers we agreed the paper could be greatly strengthened with more focus on contributions.

On the one hand, the reviews indicate the paper claims a series of contributions that could have high impact:
- Reviewer 5bwE indicates that the problem statement is well-defined.
- Reviewer 6qag mentioned that the proposed method doesn't necessarily require providing a shield beforehand.
On the other hand, some claims still seem to be insufficiently investigated and do not have sufficient support:
- Reviewer sSLq mentions that while the paper provides a framework that encompasses many settings, it does not show in sufficient detail how to treat each possible instantiation.
- Reviewer 7jyh found multiple claims in the introduction are not supported, and the evaluation is insufficient.
- Reviewer amjX has concerns regarding the novelty of the approach.

In summary, the paper deals with an important problem, investigating shielding in reinforcement learning with a learned model. However, the paper needs a significant review to clarify its core contribution and corresponding empirical evaluation before it is ready for publication.

After reading the manuscript, I suggest the following points for consideration in a new submission:
- The paper could compare its sample complexity with the sample complexity results from the RL for CMDPs literature.
- The empirical evaluation could show the feasibility of this approach in generalizing several CMDP settings. That is, besides problems with regular safety properties, the empirical evaluation could compare this approach with the SOTA constrained RL approaches for CMDP.